# Whole-genome sequencing reveals host factors underlying critical COVID-19

Athanasios Kousathanas[1,556], Erola Pairo-Castineira[2,3,556], Konrad Rawlik[2], Alex Stuckey[1], Christopher A. Odhams[1], Susan Walker[1], Clark D. Russell[2,4], Tomas Malinauskas[5], Yang Wu[6], Jonathan Millar[2], Xia Shen[7,8], Katherine S. Elliott[5], Fiona Griffiths[2], Wilna Oosthuyzen[2], Kirstie Morrice[9], Sean Keating[10], Bo Wang[2], Daniel Rhodes[1], Lucija Klaric[3], Marie Zechner[2], Nick Parkinson[2], Afshan Siddiq[1], Peter Goddard[1], Sally Donovan[1], David Maslove[11], Alistair Nichol[12], Malcolm G. Semple[13,14], Tala Zainy[1], Fiona Maleady-Crowe[1], Linda Todd[1], Shahla Salehi[1], Julian Knight[5], Greg Elgar[1], Georgia Chan[1], Prabhu Arumugam[1], Christine Patch[1], Augusto Rendon[1], David Bentley[15], Clare Kingsley[15], Jack A. Kosmicki[16], Julie E. Horowitz[16], Aris Baras[16], Goncalo R. Abecasis[16], Manuel A. R. Ferreira[16], Anne Justice[17], Tooraj Mirshahi[17], Matthew Oetjens[17], Daniel J. Rader[18], Marylyn D. Ritchie[18], Anurag Verma[18], Tom A. Fowler[1,19], Manu Shankar-Hari[20], Charlotte Summers[21], Charles Hinds[22], Peter Horby[23], Lowell Ling[24], Danny McAuley[25,26], Hugh Montgomery[27], Peter J. M. Openshaw[28,29], Paul Elliott[30], Timothy Walsh[10], Albert Tenesa[2,3,8], GenOMICC investigators*, 23andMe investigators*, COVID-19 Human Genetics Initiative*, Angie Fawkes[9], Lee Murphy[9], Kathy Rowan[31], Chris P. Ponting[3], Veronique Vitart[3], James F. Wilson[3,8], Jian Yang[32,33], Andrew D. Bretherick[3], Richard H. Scott[1,34], Sara Clohisey Hendry[2,557], Loukas Moutsianas[1,557], Andy Law[2,557], Mark J. Caulfield[1,35,557]✉ & J. Kenneth Baillie[2,3,4,10,557]✉

Critical COVID-19 is caused by immune-mediated inflammatory lung injury. Host genetic variation influences the development of illness requiring critical care[1] or hospitalization[2–4] after infection with SARS-CoV-2. The GenOMICC (Genetics of Mortality in Critical Care) study enables the comparison of genomes from individuals who are critically ill with those of population controls to find underlying disease mechanisms. Here we use whole-genome sequencing in 7,491 critically ill individuals compared with 48,400 controls to discover and replicate 23 independent variants that significantly predispose to critical COVID-19. We identify 16 new independent associations, including variants within genes that are involved in interferon signalling (*IL10RB* and *PLSCR1*), leucocyte differentiation (*BCL11A*) and blood-type antigen secretor status (*FUT2*). Using transcriptome-wide association and colocalization to infer the effect of gene expression on disease severity, we find evidence that implicates multiple genes—including reduced expression of a membrane flippase (*ATP11A*), and increased expression of a mucin (*MUC1*)—in critical disease. Mendelian randomization provides evidence in support of causal roles for myeloid cell adhesion molecules (*SELE*, *ICAM5* and *CD209*) and the coagulation factor *F8*, all of which are potentially druggable targets. Our results are broadly consistent with a multi-component model of COVID-19 pathophysiology, in which at least two distinct mechanisms can predispose to life-threatening disease: failure to control viral replication; or an enhanced tendency towards pulmonary inflammation and intravascular coagulation. We show that comparison between cases of critical illness and population controls is highly efficient for the detection of therapeutically relevant mechanisms of disease.

Critical illness in COVID-19 is both an extreme disease phenotype and a relatively homogeneous clinical definition; it includes patients with hypoxaemic respiratory failure[5] with acute lung injury[6], and excludes many patients with non-pulmonary clinical presentations[7], who are known to have divergent responses to therapy[8]. In the UK, individuals in the critically ill group are younger, less likely to have significant comorbidity and more severely affected than a general hospitalized cohort[5], characteristics which may amplify observed genetic effects. In addition, as development of critical illness is in itself a key clinical end-point for therapeutic trials[8], using critical illness as a phenotype in genetic studies enables the detection of directly therapeutically relevant genetic effects[1].

**Table 1 | Lead variants from independent association signals in the per-population GWAS and multi-ancestry meta-analysis**

| chr:pos (hg38) | rsID | REF | ALT | RAF | OR | OR$_{CI}$ | P | P$_{hgib2.23m}$ | P$_{reg}$ | Consequence | Gene | Cit. |
|---|---|---|---|---|---|---|---|---|---|---|---|---|
| 1:155066988 | rs114301457 | C | T* | 0.0058 | 2.4 | 1.82–3.16 | $6.8×10^{-10}$ | 0.00011* | – | Synonymous | EFNA4 | – |
| 1:155175305† | rs7528026 | G | A* | 0.032 | 1.4 | 1.24–1.55 | $7.16×10^{-9}$ | 0.00012* | – | Intron | TRIM46 | – |
| 1:155197995 | rs41264915 | A* | G | 0.89 | 1.3 | 1.19–1.37 | $1.02×10^{-12}$ | $1.51×10^{-9}$* | – | Intron | THBS3 | 3 |
| 2:60480453† | rs1123573 | A* | G | 0.61 | 1.1 | 1.09–1.18 | $9.85×10^{-10}$ | 0.000018* | – | Intron | BCL11A | – |
| 3:45796521 | rs2271616 | G | T* | 0.14 | 1.3 | 1.21–1.37 | $9.9×10^{-17}$ | $4.95×10^{-9}$* | – | 5′ UTR | SLC6A20 | 3 |
| 3:45859597 | rs73064425 | C | T* | 0.077 | 2.7 | 2.51–2.94 | $1.97×10^{-133}$ | $1.02×10^{-77}$* | – | Intron | LZTFL1 | 2 |
| 3:146517122 | rs343320 | G | A* | 0.081 | 1.2 | 1.16–1.35 | $4.94×10^{-9}$ | 0.00028* | – | Missense | PLSCR1 | – |
| 5:131995059 | rs56162149 | C | T* | 0.17 | 1.2 | 1.13–1.26 | $7.65×10^{-11}$ | 0.00074* | – | Intron | ACSL6 | – |
| 6:32623820 | rs9271609 | T* | C | 0.65 | 1.1 | 1.09–1.19 | $3.26×10^{-9}$ | 0.89 | – | – | HLA-DRB1 | – |
| 6:41515007† | rs2496644 | A* | C | 0.015 | 1.4 | 1.32–1.60 | $7.59×10^{-15}$ | $3.17×10^{-7}$* | – | Intron | LINC01276 | 3 |
| 9:21206606 | rs28368148 | C | G* | 0.013 | 1.7 | 1.45–2.09 | $1.93×10^{-9}$ | 0.0024 | 0.00089 | Missense | IFNA10 | – |
| 11:34482745 | rs61882275 | G* | A | 0.62 | 1.1 | 1.10–1.20 | $1.61×10^{-10}$ | $1.9×10^{-10}$* | – | Intron | ELF5 | – |
| 12:132489230 | rs56106917 | GC | G* | 0.49 | 1.1 | 1.09–1.18 | $2.08×10^{-9}$ | 0.00047* | – | Upstream | FBRSL1 | – |
| 13:112889041 | rs9577175 | C | T* | 0.23 | 1.2 | 1.12–1.24 | $3.71×10^{-11}$ | $1.29×10^{-6}$* | – | Downstream | ATP11A | – |
| 15:93046840† | rs4424872 | T* | A | 0.0079 | 2.4 | 1.87–3.01 | $8.61×10^{-13}$ | – | 0.29 | Intron | RGMA | – |
| 16:89196249 | rs117169628 | G | A* | 0.15 | 1.2 | 1.12–1.26 | $4.4×10^{-9}$ | $6.57×10^{-9}$* | – | Missense | SLC22A31 | |
| 17:46152620 | rs2532300 | T* | C | 0.77 | 1.2 | 1.10–1.22 | $4.19×10^{-9}$ | $2.49×10^{-9}$* | – | Intron | KANSL1 | 9 |
| 17:49863260 | rs3848456 | C | A* | 0.029 | 1.5 | 1.33–1.70 | $4.19×10^{-11}$ | $1.34×10^{-7}$* | – | Regulatory | . | 3 |
| 19:4717660 | rs12610495 | A | G* | 0.31 | 1.3 | 1.27–1.38 | $3.91×10^{-36}$ | $5.74×10^{-19}$* | – | Intron | DPP9 | 1 |
| 19:10305768 | rs73510898 | G | A* | 0.093 | 1.3 | 1.19–1.37 | $1.57×10^{-11}$ | 0.00016* | – | Intron | ZGLP1 | – |
| 19:10352442 | rs34536443 | G | C* | 0.05 | 1.5 | 1.36–1.65 | $6.98×10^{-17}$ | $4.06×10^{-11}$* | – | Missense | TYK2 | 1 |
| 19:48697960 | rs368565 | C | T* | 0.44 | 1.1 | 1.1–1.2 | $3.55×10^{-11}$ | 0.00087* | – | Intron | FUT2 | – |
| 21:33230000 | rs17860115 | C | A* | 0.32 | 1.2 | 1.19–1.3 | $9.69×10^{-22}$ | $1.77×10^{-18}$* | – | 5′ UTR | IFNAR2 | 1 |
| 21:33287378 | rs8178521 | C | T* | 0.27 | 1.2 | 1.12–1.23 | $3.53×10^{-12}$ | $8.02×10^{-6}$* | – | Intron | IL10RB | – |
| 21:33959662 | rs35370143 | T | TAC* | 0.083 | 1.3 | 1.17–1.36 | $1.24×10^{-9}$ | $2.33×10^{-7}$* | – | Intron | LINC00649 | – |

Variants and the reference and alternative allele are reported according to GRCh38. The three variants discovered in multi-ancestry meta-analysis but not in the European ancestry GWAS are labelled with ‡, and † indicates genome-wide significant heterogeneity. REF and ALT columns indicate the reference and alternative alleles; an asterisk (*) indicates the risk allele. For each variant, we report the risk allele frequency in Europeans (RAF), the odds ratio and 95% confidence interval (OR and OR$_{CI}$), and the association P value. 'Consequence' indicates the predicted worst consequence type across GENCODE basic transcripts predicted by VEP (v.104), and 'Gene' indicates the VEP-predicted gene, but not necessarily the causal mediator. For the HLA locus, the gene that was identified by HLA allele analysis is displayed. An asterisk (*) next to the replication P value (P$_{hgib2.23m}$ - HGI B2 and 23andMe; or P$_{reg}$- Regeneron) indicates that the lead signal (from multi-ancestry meta-analysis) is replicated with a Bonferroni-corrected $P < 0.002$ (0.05/25) with a concordant direction of effect. The 'Cit.' column lists citation numbers for the first publication of confirmed genome-wide associations with critical illness or (in brackets) any COVID-19 phenotype.

Using microarray genotyping in 2,244 cases, we previously discovered that critical COVID-19 is associated with genetic variation in the host immune response to viral infection (OAS1, IFNAR2 and TYK2) and the inflammasome regulator DPP9[1]. In collaboration with international groups, we extended these findings to include a variant near TAC4 (rs77534576)[3]. Several variants have been associated with milder phenotypes, including the ABO blood-type locus[2], a pleiotropic inversion in chr17q21.31[9] and associations in five additional loci, including the T lymphocyte-associated transcription factor, FOXP4[3]. An enrichment of rare loss-of-function variants in candidate interferon signalling genes has been reported[4], but this has yet to be replicated at genome-wide significance thresholds[10,11].

In partnership with Genomics England, we performed whole-genome sequencing (WGS) to improve the resolution and deepen the fine-mapping of significant signals and thereby provide further biological insight into critical COVID-19. Here we present results from a cohort of 7,491 critically ill patients from 224 intensive care units, compared with 48,400 control individuals, describing the discovery and validation of 23 gene loci for susceptibility to critical COVID-19 (Extended Data Fig. 1).

## Genome-wide association study analysis

After quality control procedures, we used a logistic mixed model regression, implemented in SAIGE[12], to perform association analyses with unrelated individuals (critically ill cases, $n = 7,491$; controls, $n = 48,400$ (100,000 Genomes Project (100k) cohort, $n = 46,770$; mild COVID-19, $n = 1,630$) (Methods, Supplementary Table 2). A total of 1,339 of these cases were included in the primary analysis for our previous report[1]. Genome-wide association studies (GWASs) were performed separately for genetic ancestry groups ($n_{cases}/n_{controls}$: European (EUR) 5,989/42,891; South Asian (SAS) 788/3,793; African (AFR) 440/1,350; East Asian (EAS) 274/366), and combined by inverse-variance-weighted fixed effects meta-analysis using METAL (Methods). We established the independence of signals using GCTA-cojo, and we validated this with conditional analysis using individual-level data with SAIGE (Methods, Supplementary Table 6). To reduce the risk of spurious associations arising from genotyping or pipeline errors, we required supporting evidence from variants in linkage disequilibrium (LD) for all genome-wide-significant variants: observed z-scores for each variant were compared with imputed z-scores for the same variant, with discrepant values being excluded (see Methods, Supplementary Fig. 2).

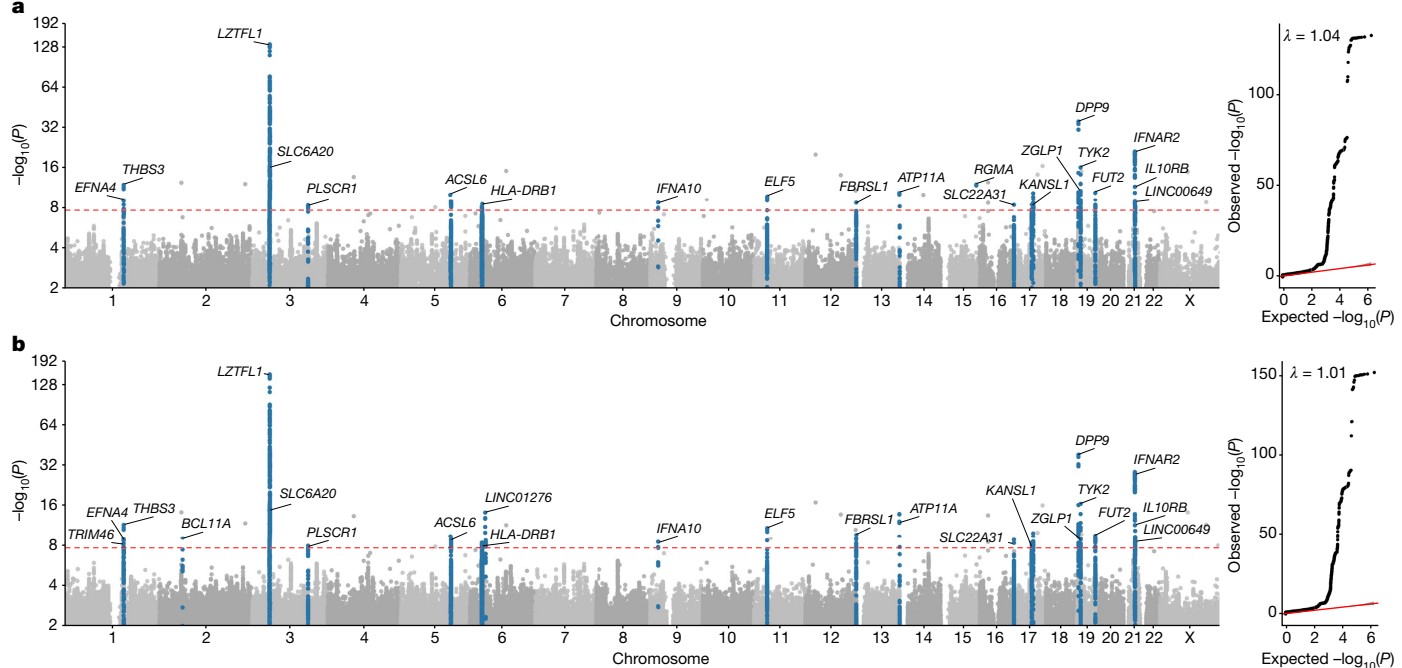

**Fig. 1 | GWAS results for the EUR ancestry group, and multi-ancestry meta-analysis.** Manhattan plots are shown on the left and quantile–quantile (QQ) plots of observed versus expected *P* values on the right, with genomic inflation (λ) displayed for each analysis. Highlighted results in blue in the Manhattan plots indicate variants that are LD-clumped ($r^2 = 0.1$, $P_2 = 0.01$, EUR LD) with the lead variants at each locus. Gene name annotation indicates genes that are affected by the predicted worst consequence type of each lead variant (annotation by Variant Effect Predictor (VEP)). For the HLA locus, the gene that was identified by HLA allele analysis is annotated. The GWAS was performed using logistic regression and meta-analysed by the inverse variant method. The red dashed line shows the Bonferroni-corrected *P* value: $P = 2.2 \times 10^{-8}$.

In population-specific analyses, we discovered 22 independent genome-wide-significant associations in the EUR ancestry group (Fig. 1, Supplementary Fig. 11, Table 1) at a *P* value threshold adjusted for multiple testing ($2.2 \times 10^{-08}$; Supplementary Table 5). In multi-ancestry meta-analysis, we identified an additional three independent genome-wide-significant association signals (Fig. 1, Table 1).

To assess the sensitivity of our results to mismatches of demographic characteristics between cases and controls (Supplementary Figs. 9, 10), we performed an age-, sex- and body mass index (BMI)-matched case–control analysis (Supplementary Figs. 18–21). As there is a theoretical risk of mismatch between cases and 100,000 Genomes Project participants in risk factors for exposure (for example, shielding behaviour) or susceptibility to critical COVID-19 (for example, immunosuppression), we performed a sensitivity analysis using only the cohort with mild COVID-19 (see above; Supplementary Table 10). In both of these analyses, allele frequencies and directions of effect were concordant for all lead signals.

We inferred credible sets of variants using Bayesian fine-mapping with susieR[13], by analysing the GWAS summaries of 17 regions of genomic length 3 Mb that were flanking groups of lead signals. We obtained 22 independent credible sets of variants for EUR and an additional 2 from the trans-ancestry meta-analysis with a posterior inclusion probability greater than 0.95 (Extended Data Table 1, Supplementary Information). Fine-mapping of the association signals revealed putative causal variants for both previously reported and novel association signals (see Supplementary Information, Extended Data Table 1). In 12 out of the 24 fine-mapped signals, the credible sets included 5 or fewer variants, and for 8 signals we detected variants with predicted missense or worse consequence across each credible set (Extended Data Table 1). We were able to fine-map multiple independent signals at previously identified loci (Fig. 3, Extended Data Figs. 2, 4). For example, the signal in the 3p21.31 region[2], was fine-mapped into two independent associations, with the credible set for the first refined to a single variant in the

5′ untranslated region (UTR) of *SLC6A20* (chr3:45796521:G:T, rs2271616, odds ratio (OR): 1.29, 95% confidence interval (CI):1.21–1.37), and the second credible set including multiple variants in downstream and intronic regions of *LZTFL1* (Fig. 3). Among the novel signals, at 3q24 and 9p21.3 we detected missense variants that affect *PLSCR1* and *IFNA10*, respectively (chr3:146517122:G:A, rs343320, p.His262Tyr, OR: 1.24, 95% CI:1.15–1.33, CADD: 22.6; chr9:21206606:C:G, rs28368148, p.Trp164Cys, OR:1.74, 95% CI: 1.45–2.09, CADD: 23.9). Both are predicted to be deleterious by the Combined Annotation Dependent Depletion (CADD) tool[14]. Structural predictions for these variants suggest functional effects (Extended Data Fig. 5). We assessed whether the main signals of this study were underlain by rarer variants with a lower minor allele frequency (MAF) (less than 0.02%) than our GWAS default threshold (less than 0.5%), by including rarer variant summaries when fine-mapping, but no additional variants were added to the main credible sets (Supplementary Table 9).

Consistent with our expectation that genetic susceptibility has a stronger role in younger individuals, age-stratified analysis (individuals of younger than 60 years old versus individuals of 60 years old or above) in the EUR group revealed a signal in the 3p21.31 region with a significantly stronger effect in the younger age group (chr3:45801750:G:A, rs13071258, OR: 3.34, 95% CI: 2.98–3.75 versus OR: 2.1, 95% CI 1.88–2.34), which is in strong LD ($r^2 = 0.947$) with the main GWAS signal indexed by rs73064425. Sex-specific analysis did not reveal significant effects (Supplementary Fig. 17).

## Replication

For replication, we performed a meta-analysis of summary statistics generously shared by 23andMe and the COVID-19 Host Genetics Initiative (HGI) data freeze 6 (B2). As a previous analysis of GenOMICC[1] contributes a substantial part of the signal at each locus in HGI v.6, and leave-one-out analyses were not available, we removed the signal

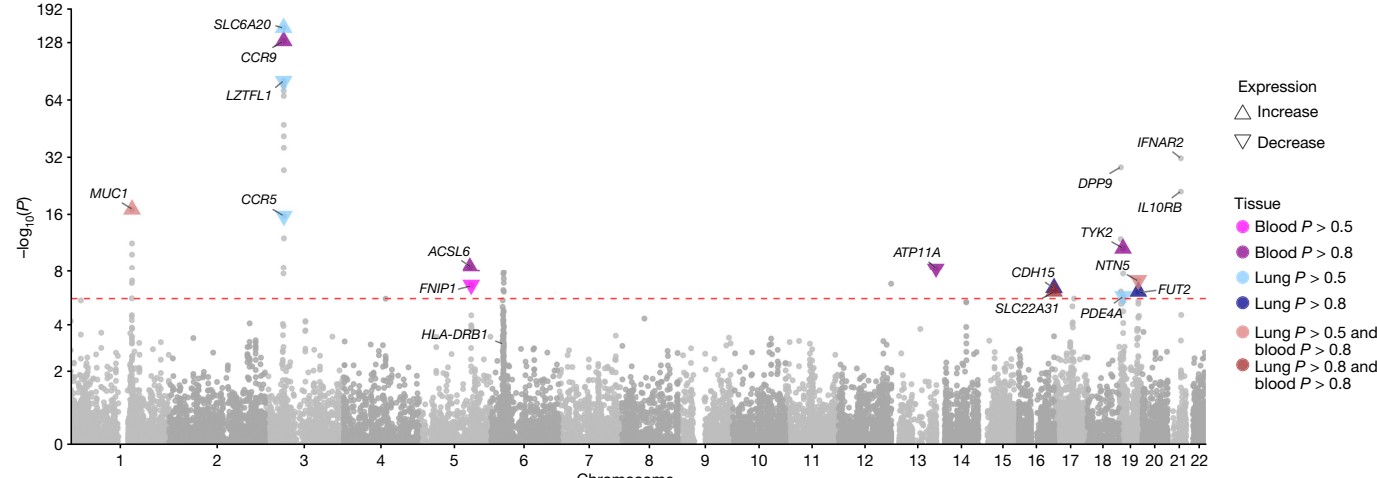

**Fig. 2 | Gene-level Manhattan plot showing results from the TWAS meta-analysis and highlighting genes that colocalize with GWAS signals or have strong metaTWAS associations.** The highlighting colour is different for the lung and blood tissue data that were used for colocalization, and we also distinguish loci that were significant in both. Results are grouped according to two classes for the posterior probability of colocalization (PP$_{H4}$): $P > 0.5$ and $P > 0.8$. If a variant is placed in both classes, then the colour that corresponds to the higher probability class is shown. Arrowheads indicate the direction of change in gene expression associated with an increased disease risk. The red dashed line shows the Bonferroni-corrected significance threshold for the metaTWAS analysis at $P = 2.3 \times 10^{-6}$.

from GenOMICC cases in HGI v.6 using mathematical subtraction to ensure independence (Methods). Using LD clumping to find variants genotyped in both the discovery and replication studies, we required $P < 0.002$ (0.05/25) and concordant direction of effect (Table 1, Supplementary Table 8) for replication. We interrogated two variants that failed replication in this set in a second GWAS meta-analysis of hospitalized patients with COVID-19 from UK Biobank, AncestryDNA, Penn Medicine Biobank and Geisinger Health Systems, which included a total of 9,937 individuals who were hospitalized with COVID-19 and 1,059,390 control individuals. This led to a further successful replicated finding, in *IFNA10* (Table 1).

We replicated 23 of the 25 significant associations that were identified in the population-specific and/or multi-ancestry GWASs. One of the non-replicated signals (rs4424872) corresponds to a rare variant that may not be well represented in the replication datasets—which are dominated by single-nucleotide polymorphism (SNP) genotyping data—but which also had significant heterogeneity among ancestries. The second non-replicated signal is within the human leukocyte antigen (HLA) locus, which has complex LD (see below).

## HLA region

The lead variant in the HLA region, rs9271609, lies upstream of the *HLA-DQA1* and *HLA-DRB1* genes. To investigate the contribution of specific HLA alleles to the observed association in the HLA region, we imputed HLA alleles at a four-digit (two-field) level using HIBAG[15]. The only allele that reached genome-wide significance was *HLA-DRB1*04:01* (OR: 0.80, 95% CI: 0.75–0.86, $P = 1.6 \times 10^{-10}$ in EUR), which has a stronger $P$ value than the lead SNP in the region (OR: 0.88, 95% CI: 0.84–0.92, $P = 3.3 \times 10^{-9}$ in EUR) and is a better fit to the data (Akaike information criterion (AIC): AIC$_{DRB1*04:01}$ = 30,241.34; AIC$_{leadSNP}$ = 30,252.93) (Extended Data Fig. 6). *HLA-DRB1*04:01* has been previously reported to confer protection against severe disease in a small cohort of European ancestry[16].

## Gene burden testing

To assess the contribution of rare variants to critical illness, we performed gene-based analysis using SKAT-O as implemented in

SAIGE-GENE[17] on a subset of 12,982 individuals from our cohort (7,491 individuals with critical COVID-19 and 5,391 control individuals), for which the genome-sequencing data were processed with the same alignment and variant calling pipeline. We tested the burden of rare (MAF < 0.5%) variants considering the predicted variant consequence type (tested variant counts provided in the Supplementary Information). We assessed burden using a strict definition for damaging variants (high-confidence putative loss-of-function (pLoF) variants as identified by LOFTEE[18]) and a lenient definition (pLoF plus missense variants with CADD ≥ 10)[14], but found no significant associations at a gene-wide-significance level. Moreover, all individual rare variants included in the tests had $P$ values greater than $10^{-5}$.

Consistent with other recent work[11], we did not find any significant gene burden test associations among the 13 genes previously reported from an interferon-pathway-focused study[4] (tests for all genes had $P > 0.05$; Supplementary Information), and we did not replicate the reported association[19–21] in *TLR7* (EUR $P = 0.30$ for pLoF and $P = 0.075$ for missense variants).

## Transcriptome-wide association study analysis

To infer the effect of genetically determined variation in gene expression on disease susceptibility, we performed a transcriptome-wide association study (TWAS) using gene expression data (GTEx v.8; ref. [22]) for two disease-relevant tissues: lung and whole blood. We found significant associations between critical COVID-19 and predicted expression in lung (14 genes) and blood (6 genes) (Supplementary Fig. 23) and in an all-tissue meta-analysis (GTEx v.8; 51 genes) (Supplementary Fig. 24). Expression signals for 16 genes significantly colocalized with susceptibility (Fig. 2). As the LD structure of the HLA is complex, we only assessed colocalization for the significant association, *HLA-DRB1*. Although it was not significant in our TWAS analysis, expression quantitative trait loci (eQTLs) in the proximity of the association significantly colocalize with the GWAS signal for both blood and lung (both PP$_{H4}$ > 0.8; Supplementary Information).

We repeated the TWAS analysis using models of intron excision rate from GTEx v.8 to obtain a splicing TWAS, which revealed significant signals in lung (16 genes) and whole blood (9 genes), and in an all-tissue meta-analysis (33 genes); 11 of these had strongly colocalizing splicing signals (Supplementary Information).

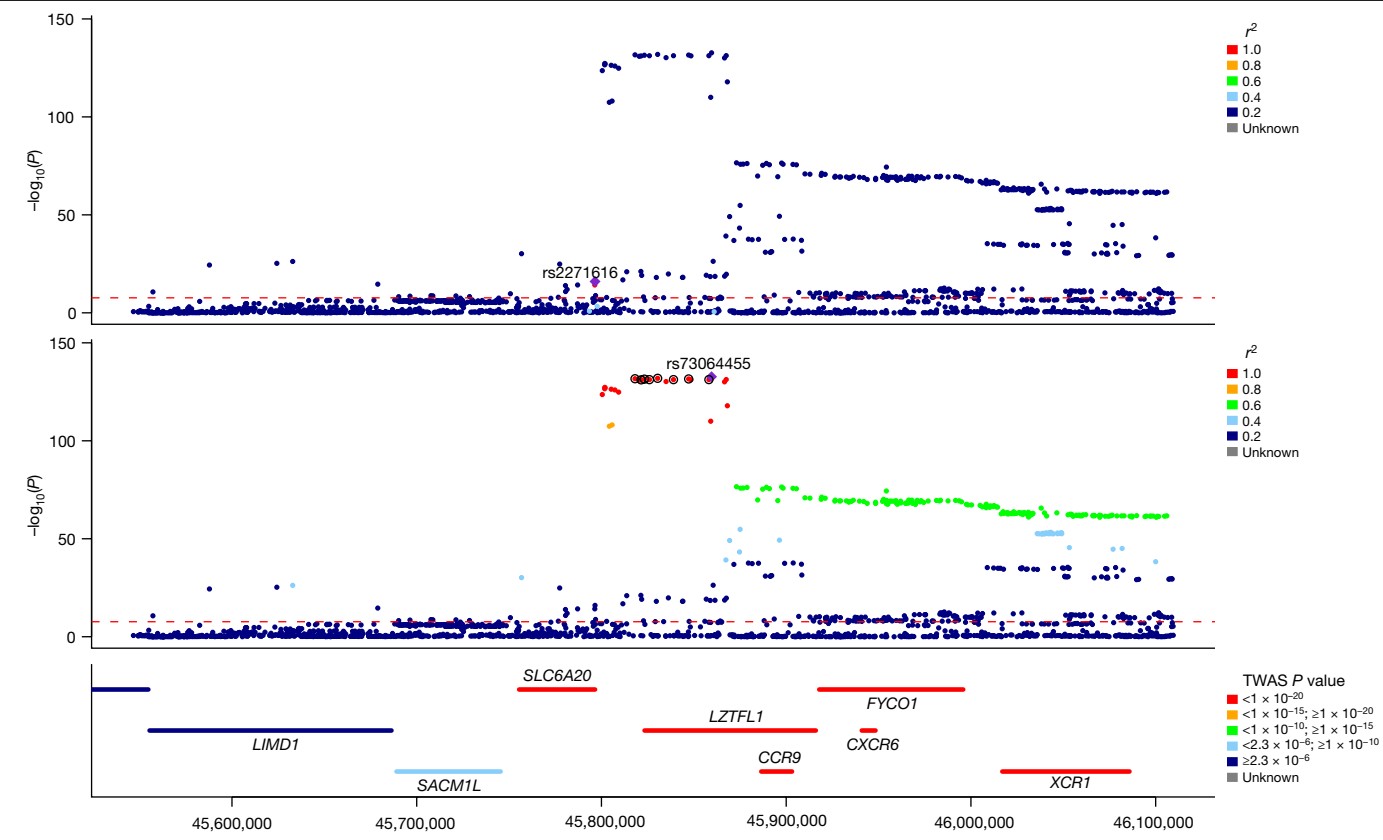

**Fig. 3 | Regional detail showing fine-mapping to identify two adjacent independent signals on chromosome 3.** Top two panels, variants in LD with the lead variants shown. The variants that are included in two independent credible sets are displayed with black outline circles. The $r^2$ values in the key denote upper limits; that is, 0.2 = [0, 0.2], 0.4 = [0.2, 0.4], 0.6 = [0.4, 0.6], 0.8 = [0.6, 0.8],1 = [0.8, 1]. Bottom, locations of protein-coding genes, coloured by TWAS $P$ value. The red dashed line shows the Bonferroni-corrected $P$ value: $P = 2.2 \times 10^{-8}$ for individuals of European ancestry.

## Mendelian randomization

We performed generalized summary-data-based Mendelian randomization (GSMR)[23] in a replicated outcome study design using the protein quantitative trait loci (pQTLs) from the INTERVAL study[24]. GSMR incorporates information from multiple independent SNPs and provides stronger evidence of a causal relationship than single-SNP-based approaches. Of 16 proteome-wide-significant associations in this study, 8 were replicated in an external dataset at a Bonferroni-corrected $P$ value threshold of $P < 0.0031$ ($P < 0.05/16$; Extended Data Table 2, Extended Data Fig. 7).

## Discussion

We report 23 replicated genetic associations with critical COVID-19, which were discovered in only 7,491 cases. This demonstrates the efficiency of the design of the GenOMICC study, an open-source[25] international research programme (https://genomicc.org) that focuses on extreme phenotypes: patients with life-threatening infectious disease, sepsis, pancreatitis and other critical illness phenotypes. GenOMICC detects greater heritability and stronger effect sizes than other study designs across all variants (Supplementary Figs. 22, 14). In COVID-19, critical illness is not only an extreme susceptibility phenotype, but also a more homogeneous one: we have shown previously that critically ill patients with COVID-19 are more likely to have the primary disease process—hypoxaemic respiratory failure[5]—and that patients in this group have a divergent response to immunosuppressive therapy compared to other hospitalized patients[8]. We detect distinct signals at several of the associated loci, in some cases implicating different biological mechanisms.

Five of the variants associated with critical COVID-19 have direct roles in interferon signalling and broadly concordant predicted biological effects. These include a probable destabilizing amino acid substitution in a ligand, *IFNA10* (Trp164Cys, Extended Data Fig. 5), and—as we reported previously[1]—reduced expression of a subunit of its receptor *IFNAR2* (Fig. 2). *IFNAR2* signals through a kinase that is encoded by *TYK2*[1]. Although the lead variant in *TYK2* in WGS is a protein-coding variant with reduced STAT1 phosphorylation activity[26], it is also associated with significantly increased expression of *TYK2* (Fig. 2, Methods). Fine-mapping reveals a significant association with an independent missense variant in *IL10RB*, a receptor for type III (lambda) interferons (rs8178521; Table 1). Finally, we detected a lead risk variant in phospholipid scramblase 1 (chr3:146517122:G:A, rs343320; *PLSCR1*) which disrupts a nuclear localization signal that is important for the antiviral effect of interferons[27] (Extended Data Fig. 5). PLSCR1 controls the replication of other RNA viruses, including vesicular stomatitis virus, encephalomyocarditis virus and influenza A virus[27,28].

Although our genome-wide gene-based association tests did not replicate any findings from a previous pathway-specific study of rare deleterious variants[4], our results provide robust evidence implicating reduced interferon signalling in susceptibility to critical COVID-19. Notably, systemic administration of interferon in two large clinical trials, albeit late in disease, did not reduce mortality[29,30].

We found significant associations in genes that are implicated in lymphopoesis and in the differentiation of myeloid cells. *BCL11A* is essential for B and T lymphopoiesis[31] and promotes the differentiation of plasmacytoid dendritic cells[32]. *TAC4*, reported previously[3], encodes a regulator of B cell lymphopoesis[33] and antibody production[34], and promotes the survival of dendritic cells[35]. Finally, although the strongest

fine-mapping signal at 5q31.1 (chr5:131995059:C:T, rs56162149) is in an intron of *ACSL6* with significant effects on expression (Supplementary Information), the credible set includes a missense variant in *CSF2* (encoding granulocyte–macrophage colony stimulating factor; GM-CSF) of uncertain significance (chr5:132075767:T:C; Extended Data Table 1). We have previously shown that GM-CSF is strongly up-regulated in critical COVID-19[36], and it is already under investigation as a target for therapy[37]. Mendelian randomization results are consistent with a direct link between the plasma levels of a closely related cytokine receptor subunit, IL3RA, and critical COVID-19 (Extended Data Table 2).

Fine-mapping, colocalization and TWAS analyses provide evidence for increased expression of *MUC1* as the mediator of the association with rs41264915 (Supplementary Table 12). This suggests that mucins could have a therapeutically important role in the development of critical illness in COVID-19.

Mendelian randomization provides genetic evidence in support of a causal role for coagulation factors (*F8*) and platelet activation (*PDGFRL*) in critical COVID-19 (Extended Data Table 2, Extended Data Fig. 7), consistent with autopsy[6], proteomic[38] and therapeutic[39] evidence. Perhaps more importantly, we identify specific and closely related intercellular adhesion molecules that have known roles in the recruitment of inflammatory cells to sites of inflammation, including E-selectin (*SELE*), intercellular adhesion molecule 5 (*ICAM5*) and DC-SIGN (dendritic-cell-specific ICAM3-grabbing non-integrin; *CD209*), which may provide additional therapeutic targets. DC-SIGN (*CD209*) mediates pathogen endocytosis and antigen presentation, and is known to be involved in multiple viral infections, including SARS-CoV and influenza A virus. It has affinity for SARS-CoV-2[40,41].

Our previous report of an association between the OAS gene cluster and severe disease was robustly replicated in an external cohort[1], but does not meet genome-wide significance in the present analysis (Supplementary Table 7). This may indicate a change in the observed effect size because any effect that is detected in GWASs is more likely to have been sampled from the larger end of the range of possible effect sizes —the 'winner's curse'. Alternatively, it may indicate either a change in the population of patients (cases or controls) or a change in the pathogen. For example it is possible that—as with the other coronaviruses that are known to infect humans[42]—more recent variants of SARS-CoV-2 have evolved to overcome this host antiviral defence mechanism.

## Limitations

In contrast to microarray genotyping, WGS is a rapidly evolving and relatively new technology for GWASs, with relatively few sources of population controls. We selected a control cohort from the 100,000 Genomes Project, which was sequenced and analysed using a different platform and bioinformatics pipeline compared with the case cohort (Extended Data Fig. 1). However, to minimize the risk of false-positive associations due to technical artifacts, extensive quality measures were used (Methods). In brief, we masked low-quality genotypes, filtered for genotype signal using a low threshold for missingness and performed a control–control relative allele frequency filter using a subset of samples processed with both bioinformatics pipelines. Finally, we required all significant associations to be supported by local variants in LD, which may be excessively stringent (Methods). Although this approach may remove some true associations, our priority is to maximize confidence in the reported signals. Of 25 variants that meet this requirement, 23 are externally replicated, and the remaining 2 may be true associations that are yet to be replicated owing to a lack of coverage or power in the replication datasets.

The design of our study incorporates genetic signals for every stage in the disease progression into a single phenotype. This includes establishment of infection, viral replication, inflammatory lung injury and hypoxaemic respiratory failure. Although we can have considerable confidence that the replicated associations with critical COVID-19 we report are robust, we cannot determine at which stage in the disease process, or in which tissue, the relevant biological mechanisms are active.

## Conclusions

These genetic associations identify biological mechanisms that may underlie the development of life-threatening COVID-19, several of which may be amenable to therapeutic targeting. Furthermore, we demonstrate the value of WGS for fine-mapping loci in a complex trait. In the context of the ongoing global pandemic, translation to clinical practice is an urgent priority. As with our previous work, biological and molecular studies—and, where appropriate, large-scale randomized trials—will be essential before our findings can be translated into clinical practice.

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

[1]Genomics England, London, UK. [2]Roslin Institute, University of Edinburgh, Edinburgh, UK. [3]MRC Human Genetics Unit, Institute of Genetics and Cancer, University of Edinburgh, Western General Hospital, Edinburgh, UK. [4]Centre for Inflammation Research, The Queen's Medical Research Institute, University of Edinburgh, Edinburgh, UK. [5]Wellcome Centre for Human Genetics, University of Oxford, Oxford, UK. [6]Institute for Molecular Bioscience, The University of Queensland, Brisbane, Queensland, Australia. [7]Biostatistics Group, Greater Bay Area Institute of Precision Medicine (Guangzhou), Fudan University, Guangzhou, China. [8]Centre for Global Health Research, Usher Institute of Population Health Sciences and Informatics, Edinburgh, UK. [9]Edinburgh Clinical Research Facility, Western General Hospital, University of Edinburgh, Edinburgh, UK. [10]Intensive Care Unit, Royal Infirmary of Edinburgh, Edinburgh, UK. [11]Department of Critical Care Medicine, Queen's University and Kingston Health Sciences Centre, Kingston, Ontario, Canada. [12]Clinical Research Centre at St Vincent's University Hospital, University College Dublin, Dublin, Ireland. [13]NIHR Health Protection Research Unit for Emerging and Zoonotic Infections, Institute of Infection, Veterinary and Ecological Sciences, University of Liverpool, Liverpool, UK. [14]Respiratory Medicine and Institute in the Park, Alder Hey Children's Hospital and University of Liverpool, Liverpool, UK. [15]Illumina Cambridge, Great Abington, UK. [16]Regeneron Genetics Center, Tarrytown, NY, USA. [17]Geisinger, Danville, PA, USA. [18]Department of Genetics, Perelman School of Medicine, University of Pennsylvania, Philadelphia, PA, USA. [19]Test and Trace, the Health Security Agency, Department of Health and Social Care, London, UK. [20]Department of Intensive Care Medicine, Guy's and St Thomas' NHS Foundation Trust, London, UK. [21]Department of Medicine, University of Cambridge, Cambridge, UK. [22]William Harvey Research Institute, Barts and the London School of Medicine and Dentistry, Queen Mary University of London, London, UK. [23]Centre for Tropical Medicine and Global Health, Nuffield Department of Medicine, University of Oxford, Oxford, UK. [24]Department of Anaesthesia and Intensive Care, The Chinese University of Hong Kong, Prince of Wales Hospital, Hong Kong, China. [25]Wellcome–Wolfson Institute for Experimental Medicine, Queen's University Belfast, Belfast, UK. [26]Department of Intensive Care Medicine, Royal Victoria Hospital, Belfast, UK. [27]UCL Centre for Human Health and Performance, London, UK. [28]National Heart and Lung Institute, Imperial College London, London, UK. [29]Imperial College Healthcare NHS Trust: London, London, UK. [30]Imperial College, London, UK. [31]Intensive Care National Audit and Research Centre, London, UK. [32]School of Life Sciences, Westlake University, Hangzhou, China. [33]Westlake Laboratory of Life Sciences and Biomedicine, Hangzhou, China. [34]Great Ormond Street Hospital, London, UK. [35]William Harvey Research Institute, Queen Mary University of London, London, UK. [556]These authors contributed equally: Athanasios Kousathanas, Erola Pairo-Castineira. [557]These authors jointly supervised this work: Sara Clohisey Hendry, Loukas Moutsianas, Andy Law, Mark J. Caulfield, J. Kenneth Baillie. *Lists of authors and their affiliations appear online. ✉e-mail: m.j.caulfield@qmul.ac.uk; j.k.baillie@ed.ac.uk

**GenOMICC investigators**

**GenOMICC co-investigators**

J. Kenneth Baillie[36,37], Colin Begg[38], Sara Clohisey Hendry[36], Charles Hinds[39], Peter Horby[40], Julian Knight[41], Lowell Ling[42], David Maslove[43], Danny McAuley[44,45], Johnny Millar[36], Hugh Montgomery[46], Alistair Nichol[47], Peter J. M. Openshaw[48,49], Alexandre C. Pereira[50], Chris P. Ponting[51], Kathy Rowan[52], Malcolm G. Semple[53,54], Manu Shankar-Hari[55], Charlotte Summers[56] & Timothy Walsh[37]

**Management, laboratory and data team**

Latha Aravindan[57], Ruth Armstrong[36], J. Kenneth Baillie[36,37], Heather Biggs[58], Ceilia Boz[36], Adam Brown[36], Richard Clark[59], Sara Clohisey Hendry[36], Audrey Coutts[59], Judy Coyle[36], Louise Cullum[36], Sukamal Das[57], Nicky Day[36], Lorna Donnelly[59], Esther Duncan[36], Angie Fawkes[59], Paul Finernan[36], Max Head Fourman[36], Anita Furlong[58], James Furniss[36], Bernadette Gallagher[36], Tammy Gilchrist[59], Ailsa Golightly[36], Fiona Griffiths[36], Katarzyna Hafezi[59], Debbie Hamilton[36], Ross Hendry[36], Andy Law[36], Dawn Law[36], Rachel Law[36], Sarah Law[36], Rebecca Lidstone-Scott[36], Louise Macgillivray[59], Alan Maclean[59], Hanning Mal[36], Sarah McCafferty[59], Ellie Mcmaster[36], Jen Meikle[36], Shona C. Moore[53], Kirstie Morrice[59], Lee Murphy[59], Sheena Murphy[57], Mybaya Hellen[36], Wilna Oosthuyzen[36], Chenqing Zheng[60], Jiantao Chen[60], Nick Parkinson[36], Trevor Paterson[36], Katherine Schon[58], Andrew Stenhouse[36], Mihaela Das[57], Maaike Swets[36,61], Helen Szoor-McElhinney[36], Filip Taneski[36], Lance Turtle[53], Tony Wackett[36], Mairi Ward[36], Jane Weaver[36], Nicola Wrobel[59], Marie Zechner[36] & Mybaya Hellen[36]

**Guy's and St Thomas' Hospital team**

Gill Arbane[62], Aneta Bociek[62], Sara Campos[62], Neus Grau[62], Tim Owen Jones[62], Rosario Lim[62], Martina Marotti[62], Marlies Ostermann[62], Manu Shankar-Hari[62] & Christopher Whitton[62]

**Barts Health NHS Trust team**

Zoe Alldis[63], Raine Astin-Chamberlain[63], Fatima Bibi[63], Jack Biddle[63], Sarah Blow[63], Matthew Bolton[63], Catherine Borra[63], Ruth Bowles[63], Maudrian Burton[63], Yasmin Choudhury[63], David Collier[63], Amber Cox[63], Amy Easthope[63], Patrizia Ebano[63], Stavros Fotiadis[63], Jana Gurasashvili[63], Rosslyn Halls[63], Pippa Hartridge[63], Delordson Kallon[63], Jamila Kassam[63], Ivone Lancoma-Malcolm[63], Maninderpal Matharu[63], Peter May[63], Oliver Mitchelmore[63], Tabitha Newman[63], Mital Patel[63], Jane Pheby[63], Irene Pinzuti[63], Zoe Prime[63], Oleksandra Prysyazhna[63], Julian Shiel[63], Melanie Taylor[63], Carey Tierney[63], Suzanne Wood[63], Anne Zak[63] & Olivier Zongo[63]

**James Cook University Hospital team**

Stephen Bonner[64], Keith Hugill[64], Jessica Jones[64], Steven Liggett[64] & Evie Headlam[64]

**Royal Stoke University Hospital team**

Nageswar Bandla[65], Minnie Gellamucho[65], Michelle Davies[65] & Christopher Thompson[65]

**North Middlesex University Hospital NHS Trust team**

Marwa Abdelrazik[66], Dhanalakshmi Bakthavatsalam[66], Munzir Elhassan[66], Arunkumar Ganesan[66], Anne Haldeos[66], Jeronimo Moreno-Cuesta[66], Dharam Purohit[66], Rachel Vincent[66], Kugan Xavier[66], Rohit Kumar[67], Alasdair Frater[66], Malik Saleem[66], David Carter[66], Samuel Jenkins[66], Zoe Lamond[66] & Alanna Wall[66]

**The Royal Liverpool University Hospital team**

Jaime Fernandez-Roman[68], David O. Hamilton[68], Emily Johnson[68], Brian Johnston[68], Maria Lopez Martinez[68], Suleman Mulla[68], David Shaw[68], Alicia A. C. Waite[68], Victoria Waugh[68], Ingeborg D. Welters[68] & Karen Williams[68]

**King's College Hospital team**

Anna Cavazza[69], Maeve Cockrell[69], Eleanor Corcoran[69], Maria Depante[69], Clare Finney[69], Ellen Jerome[69], Mark McPhail[69], Monalisa Nayak[69], Harriet Noble[69], Kevin O'Reilly[69], Evita Pappa[69], Rohit Saha[69], Sian Saha[69], John Smith[69] & Abigail Knighton[69]

**Charing Cross Hospital team**

David Antcliffe[70], Dorota Banach[70], Stephen Brett[70], Phoebe Coghlan[70], Ziortza Fernandez[70], Anthony Gordon[70], Roceld Rojo[70], Sonia Sousa Arias[70] & Maie Templeton[70]

**Nottingham University Hospital team**

Megan Meredith[71], Lucy Morris[71], Lucy Ryan[71], Amy Clark[71], Julia Sampson[71], Cecilia Peters[71], Martin Dent[71], Margaret Langley[71], Saima Ashraf[71], Shuying Wei[71] & Angela Andrew[71]

**John Radcliffe Hospital team**

Archana Bashyal[72], Neil Davidson[72], Paula Hutton[72], Stuart McKechnie[72] & Jean Wilson[72]

**Kingston Hospital team**

David Baptista[73], Rebecca Crowe[73], Rita Fernandes[73], Rosaleen Herdman-Grant[73], Anna Joseph[73], Denise O'Connor[74], Meryem Allen[73], Adam Loveridge[73], India McKenley[73], Eriko Morino[73], Andres Naranjo[73], Richard Simms[73], Kathryn Sollesta[73], Andrew Swain[73], Harish Venkatesh[73], Jacyntha Khera[73] & Jonathan Fox[73]

**Royal Infirmary of Edinburgh team**

Gillian Andrew[75], J. Kenneth Baillie[75], Lucy Barclay[75], Marie Callaghan[75], Rachael Campbell[75], Sarah Clark[75], Dave Hope[75], Lucy Marshall[75], Corrienne McCulloch[75], Kate Briton[75], Jo Singleton[75] & Sophie Birch[75]

**Queen Alexandra Hospital team**

Lutece Brimfield[76], Zoe Daly[76], David Pogson[76], Steve Rose[76] & Angela Nown[76]

**Morriston Hospital team**

Ceri Battle[77], Elaine Brinkworth[77], Rachel Harford[77], Carl Murphy[77], Luke Newey[77], Tabitha Rees[77], Marie Williams[77] & Sophie Arnold[77]

**Addenbrooke's Hospital team**

Petra Polgarova[78], Katerina Stroud[78], Charlotte Summers[78], Eoghan Meaney[78], Megan Jones[78], Anthony Ng[78], Shruti Agrawal[78], Nazima Pathan[78], Deborah White[78], Esther Daubney[78] & Kay Elston[78]

**BHRUT (Barking Havering)—Queen's Hospital and King George Hospital team**

Lina Grauslyte[79], Musarat Hussain[79], Mandeep Phull[79], Tatiana Pogreban[79], Lace Rosaroso[79], Erika Salciute[79], George Franke[79], Joanna Wong[79] & Aparna George[79]

**Royal Sussex County Hospital team**

Laura Ortiz-Ruiz de Gordoa[80], Emily Peasgood[80] & Claire Phillips[80]

**Queen Elizabeth Hospital team**

Michelle Bates[81], Jo Dasgin[81], Jaspret Gill[81], Annette Nilsson[81], James Scriven[81], Amy Collins[82], Waqas Khaliq[82] & Estefania Treus Gude[82]

**St George's Hospital team**

Carlos Castro Delgado[83], Deborah Dawson[83], Lijun Ding[83], Georgia Durrant[83], Obiageri Ezeobu[83], Sarah Farnell-Ward[83], Abiola Harrison[83], Rebecca Kanu[83], Susannah Leaver[83], Elena Maccacari[83], Soumendu Manna[83], Romina Pepermans Saluzzio[83], Joana Queiroz[83], Tinashe Samakomva[83], Christine Sicat[83], Joana Texeira[83], Edna Fernandes Da Gloria[83], Ana Lisboa[83], John Rawlins[83], Jisha Mathew[83], Ashley Kinch[83], William James Hurt[83], Nirav Shah[83], Victoria Clark[83], Maria Thanasi[83], Nikki Yun[83] & Kamal Patel[83]

**Stepping Hill Hospital team**

Sara Bennett[84], Emma Goodwin[84], Matthew Jackson[84], Alissa Kent[84], Clare Tibke[84], Wiesia Woodyatt[84] & Ahmed Zaki[84]

**Countess of Chester Hospital team**

Azmerelda Abraheem[85], Peter Bamford[85], Kathryn Cawley[85], Charlie Dunmore[85], Maria Faulkner[85], Rumanah Girach[85], Helen Jeffrey[85], Rhianna Jones[85], Emily London[85], Imrun Nagra[85], Farah Nasir[85], Hannah Sainsbury[85] & Clare Smedley[85]

**Royal Blackburn Teaching Hospital team**

Tahera Patel[86], Matthew Smith[86], Srikanth Chukkambotla[86], Aayesha Kazi[86], Janice Hartley[86], Joseph Dykes[86], Muhammad Hijazi[86], Sarah Keith[86], Meherunnisa Khan[86], Janet Ryan-Smith[86], Philippa Springle[86], Jacqueline Thomas[86], Nick Truman[86], Samuel Saad[86], Dabheoc Coleman[86], Christopher Fine[86], Roseanna Matt[86], Bethan Gay[86], Jack Dalziel[86], Syamlan Ali[86], Drew Goodchild[86], Rhiannan Harling[86], Ravi Bhatterjee[86], Wendy Goddard[86], Chloe Davison[86], Stephen Duberly[86], Jeanette Hargreaves[86] & Rachel Bolton[86]

**The Tunbridge Wells Hospital and Maidstone Hospital team**

Miriam Davey[87], David Golden[87] & Rebecca Seaman[87]

**Royal Gwent Hospital team**

Shiney Cherian[88], Sean Cutler[88], Anne Emma Heron[88], Anna Roynon-Reed[88], Tamas Szakmany[88], Gemma Williams[88], Owen Richards[88] & Yusuf Cheema[88]

**Pinderfields General Hospital team**

Hollie Brooke[89], Sarah Buckley[89], Jose Cebrian Suarez[89], Ruth Charlesworth[89], Karen Hansson[89], John Norris[89], Alice Poole[89], Alastair Rose[89], Rajdeep Sandhu[89], Brendan Sloan[89], Elizabeth Smithson[89], Muthu Thirumaran[89], Veronica Wagstaff[89] & Alexandra Metcalfe[89]

**Royal Berkshire NHS Foundation Trust team**

Mark Brunton[90], Jess Caterson[90], Holly Coles[90], Matthew Frise[90], Sabi Gurung Rai[90], Nicola Jacques[90], Liza Keating[90], Emma Tilney[90], Shauna Bartley[90] & Parminder Bhuie[90]

**Broomfield Hospital team**

Sian Gibson[91], Amanda Lyle[91], Fiona McNeela[91], Jayachandran Radhakrishnan[91] & Alistair Hughes[91]

**Northumbria Healthcare NHS Foundation Trust team**

Bryan Yates[92], Jessica Reynolds[92], Helen Campbell[92], Maria Thompsom[92], Steve Dodds[92] & Stacey Duffy[92]

**Whiston Hospital team**

Sandra Greer[93], Karen Shuker[93] & Ascanio Tridente[93]

**Croydon University Hospital team**

Reena Khade[94], Ashok Sundar[94] & George Tsinaslanidis[94]

**York Hospital team**
Isobel Birkinshaw[95], Joseph Carter[95], Kate Howard[95], Joanne Ingham[95], Rosie Joy[95], Harriet Pearson[95], Samantha Roche[95] & Zoe Scott[95]

**Heartlands Hospital team**
Hollie Bancroft[96], Mary Bellamy[96], Margaret Carmody[96], Jacqueline Daglish[96], Faye Moore[96], Joanne Rhodes[96], Mirriam Sangombe[96], Salma Kadiri[96] & James Scriven[96]

**Ashford and St Peter's Hospital team**
Maria Croft[97], Ian White[97], Victoria Frost[97] & Maia Aquino[97]

**Barnet Hospital team**
Rajeev Jha[98], Vinodh Krishnamurthy[98], Lai Lim[98], Rajeev Jha[98], Vinodh Krishnamurthy[98] & Li Lim[98]

**East Surrey Hospital team**
Edward Combes[99], Teishel Joefield[99], Sonja Monnery[99], Valerie Beech[99] & Sallyanne Trotman[99]

**Ninewells Hospital team**
Christine Almaden-Boyle[100], Pauline Austin[100], Louise Cabrelli[100], Stephen Cole[100], Matt Casey[100], Susan Chapman[100], Stephen Cole[100] & Clare Whyte[100]

**Worthing Hospital team**
Yolanda Baird[101,102], Aaron Butler[101,102], Indra Chadbourn[101,102], Linda Folkes[101,102], Heather Fox[101,102], Amy Gardner[101,102], Raquel Gomez[101,102], Gillian Hobden[101,102], Luke Hodgson[101,102], Kirsten King[101,102], Michael Margarson[101,102], Tim Martindale[101,102], Emma Meadows[101,102], Dana Raynard[101,102], Yvette Thirlwall[101,102], David Helm[101,102] & Jordi Margalef[101,102]

**Southampton General Hospital team**
Kristine Criste[103], Rebecca Cusack[103], Kim Golder[103], Hannah Golding[103], Oliver Jones[103], Samantha Leggett[103], Michelle Male[103], Martyna Marani[103], Kirsty Prager[103], Toran Williams[103], Belinda Roberts[103] & Karen Salmon[103]

**The Alexandra Hospital team**
Peter Anderson[104], Katie Archer[104], Karen Austin[104], Caroline Davis[104], Alison Durie[104], Olivia Kelsall[104], Jessica Thrush[104], Charlie Vigurs[104], Laura Wild[104], Hannah-Louise Wood[104], Helen Tranter[104], Alison Harrison[104], Nicholas Cowley[104], Michael McAlindon[104], Andrew Burtenshaw[104], Stephen Digby[104], Emma Low[104], Aled Morgan[104], Naiara Cother[104], Tobias Rankin[104], Sarah Clayton[104] & Alex McCurdy[104]

**Sandwell General Hospital and City Hospital team**
Cecilia Ahmed[105], Balvinder Baines[105], Sarah Clamp[105], Julie Colley[105], Risna Haq[105], Anne Hayes[105], Jonathan Hulme[105], Samia Hussain[105], Sibet Joseph[105], Rita Kumar[105], Zahira Maqsood[105] & Manjit Purewal[105]

**Blackpool Victoria Hospital team**
Leonie Benham[106], Zena Bradshaw[106], Joanna Brown[106], Melanie Caswell[106], Jason Cupitt[106], Sarah Melling[106], Stephen Preston[106], Nicola Slawson[106], Emma Stoddard[106] & Scott Warden[106]

**Royal Glamorgan Hospital team**
Bethan Deacon[107], Ceri Lynch[107], Carla Pothecary[107], Lisa Roche[107], Gwenllian Sera Howe[107], Jayaprakash Singh[107], Keri Turner[107], Hannah Ellis[107] & Natalie Stroud[107]

**The Royal Oldham Hospital team**
Jodie Hunt[108], Joy Dearden[108], Emma Dobson[108], Andy Drummond[108], Michelle Mulcahy[108], Sheila Munt[108], Grainne O'Connor[108], Jennifer Philbin[108], Chloe Rishton[108], Redmond Tully[108] & Sarah Winnard[108]

**Glasgow Royal Infirmary team**
Susanne Cathcart[109], Katharine Duffy[109], Alex Puxty[109], Kathryn Puxty[109], Lynne Turner[109], Jane Ireland[109] & Gary Semple[109]

**St James's University Hospital and Leeds General Infirmary team**
Kate Long[110], Simon Whiteley[110], Elizabeth Wilby[110] & Bethan Ogg[110]

**University Hospital North Durham team**
Amanda Cowton[111,112], Andrea Kay[111,112], Melanie Kent[111,112], Kathryn Potts[111,112], Ami Wilkinson[111,112], Suzanne Campbell[111,112] & Ellen Brown[111,112]

**Fairfield General Hospital team**
Julie Melville[113], Jay Naisbitt[113], Rosane Joseph[113], Maria Lazo[113], Olivia Walton[113] & Alan Neal[113]

**Wythenshawe Hospital team**
Peter Alexander[114], Schvearn Allen[114], Joanne Bradley-Potts[114], Craig Brantwood[114], Jasmine Egan[114], Timothy Felton[114], Grace Padden[114], Luke Ward[114], Stuart Moss[114] & Susannah Glasgow[114]

**Royal Alexandra Hospital team**
Lynn Abel[115], Michael Brett[115], Brian Digby[115], Lisa Gemmell[115], James Hornsby[115], Patrick MacGoey[115], Pauline O'Neil[115], Richard Price[115], Natalie Rodden[115], Kevin Rooney[115], Radha Sundaram[115] & Nicola Thomson[115]

**Good Hope Hospital team**
Bridget Hopkins[116], James Scriven[116], Laura Thrasyvoulou[116] & Heather Willis[116]

**Tameside General Hospital team**
Martyn Clark[117], Martina Coulding[117], Edward Jude[117], Jacqueline McCormick[117], Oliver Mercer[117], Darsh Potla[117], Hafiz Rehman[117], Heather Savill[117] & Victoria Turner[117]

**Royal Derby Hospital team**
Charlotte Downes[118], Kathleen Holding[118], Katie Riches[118], Mary Hilton[118], Mel Hayman[118], Deepak Subramanian[118] & Priya Daniel[118]

**Medway Maritime Hospital team**
Oluronke Adanini[119], Nikhil Bhatia[119], Maines Msiska[119] & Rebecca Collins[119]

**Royal Victoria Infirmary team**
Ian Clement[120], Bijal Patel[120], A. Gulati[120], Carole Hays[120], K. Webster[120], Anne Hudson[120], Andrea Webster[120], Elaine Stephenson[120], Louise McCormack[120], Victoria Slater[120], Rachel Nixon[120], Helen Hanson[120], Maggie Fearby[120], Sinead Kelly[120], Victoria Bridgett[120] & Philip Robinson[120]

**Poole Hospital team**
Julie Camsooksai[121], Charlotte Humphrey[121], Sarah Jenkins[121], Henrik Reschreiter[121], Beverley Wadams[121] & Yasmin Death[121]

**Bedford Hospital team**
Victoria Bastion[122], Daphene Clarke[122], Beena David[122], Harriet Kent[122], Rachel Lorusso[122], Gamu Lubimbi[122], Sophie Murdoch[122], Melchizedek Penacerrada[122], Alastair Thomas[122], Jennifer Valentine[122], Ana Vochin[122], Retno Wulandari[122] & Brice Djeugam[122]

**Queens Hospital Burton team**
Gillian Bell[123], Katy English[123], Amro Katary[123] & Louise Wilcox[123]

**North Manchester General Hospital team**
Michelle Bruce[124], Karen Connolly[124], Tracy Duncan[124], Helen T. Michael[124], Gabriella Lindergard[124], Samuel Hey[124], Claire Fox[124], Jordan Alfonso[124], Laura Jayne Durrans[124], Jacinta Guerin[124], Bethan Blackledge[124], Jade Harris[124], Martin Hruska[124], Ayaa Eltayeb[124], Thomas Lamb[124], Tracey Hodgkiss[124], Lisa Cooper[124] & Joanne Rothwell[124]

**Aberdeen Royal Infirmary team**
Angela Allan[125], Felicity Anderson[125], Callum Kaye[125], Jade Liew[125], Jasmine Medhora[125], Teresa Scott[125], Erin Trumper[125] & Adriana Botello[125]

**Derriford Hospital team**
Liana Lankester[126], Nikitas Nikitas[126], Colin Wells[126], Bethan Stowe[126] & Kayleigh Spencer[126]

**Manchester Royal Infirmary team**
Craig Brandwood[127], Lara Smith[127], Richard Clark[127], Katie Birchall[127], Laurel Kolakaluri[127], Deborah Baines[127] & Anila Sukumaran[127]

**Salford Royal Hospital team**
Elena Apetri[128], Cathrine Basikolo[128], Bethan Blackledge[128], Laura Catlow[128], Bethan Charles[128], Paul Dark[128], Reece Doonan[128], Jade Harris[128], Alice Harvey[128], Daniel Horner[128], Karen Knowles[128], Stephanie Lee[128], Diane Lomas[128], Chloe Lyons[128], Tracy Marsden[128], Danielle McLaughlan[128], Liam McMorrow[128], Jessica Pendlebury[128], Jane Perez[128], Maria Poulaka[128], Nicola Proudfoot[128], Melanie Slaughter[128], Kathryn Slevin[128], Melanie Taylor[128], Vicky Thomas[128], Danielle Walker[128], Angiy Michael[128] & Matthew Collis[128]

**William Harvey Hospital team**
Tracey Cosier[129], Gemma Millen[129], Neil Richardson[129], Natasha Schumacher[129], Heather Weston[129] & James Rand[129]

**Queen Elizabeth University Hospital team**
Nicola Baxter[130], Steven Henderson[130], Sophie Kennedy-Hay[130], Christopher McParland[130], Laura Rooney[130], Malcolm Sim[130] & Gordan McCreath[130]

**Bradford Royal Infirmary team**
Louise Akeroyd[131], Shereen Bano[131], Matt Bromley[131], Lucy Gurr[131], Tom Lawton[131], James Morgan[131], Kirsten Sellick[131], Deborah Warren[131], Brian Wilkinson[131], Janet McGowan[131], Camilla Ledgard[131], Amelia Stacey[131], Kate Pye[131], Ruth Bellwood[131] & Michael Bentley[131]

**Bristol Royal Infirmary team**
Jeremy Bewley[132], Zoe Garland[132], Lisa Grimmer[132], Bethany Gumbrill[132], Rebekah Johnson[132], Katie Sweet[132], Denise Webster[132] & Georgia Efford[132]

**Norfolk and Norwich University Hospital (NNUH) team**
Karen Convery[133], Deirdre Fottrell-Gould[133], Lisa Hudig[133], Jocelyn Keshet-Price[133], Georgina Randell[133] & Katie Stammers[133]

**Queen Elizabeth Hospital Gateshead team**
Maria Bokhari[134], Vanessa Linnett[134], Rachael Lucas[134], Wendy McCormick[134], Jenny Ritzema[134], Amanda Sanderson[134] & Helen Wild[134]

**Sunderland Royal Hospital team**
Anthony Rostron[135], Alistair Roy[135], Lindsey Woods[135], Sarah Cornell[135], Fiona Wakinshaw[135], Kimberley Rogerson[135] & Jordan Jarmain[135]

**Aintree University Hospital team**
Robert Parker[136], Amie Reddy[136], Ian Turner-Bone[136], Laura Wilding[136] & Peter Harding[136]

**Hull Royal Infirmary team**
Caroline Abernathy[137], Louise Foster[137], Andrew Gratrix[137], Vicky Martinson[137], Priyai Parkinson[137], Elizabeth Stones[137] & Llucia Carbral-Ortega[138]

**University College Hospital team**
Georgia Bercades[139], David Brealey[139], Ingrid Hass[139], Niall MacCallum[139], Gladys Martir[139], Eamon Raith[139], Anna Reyes[139] & Deborah Smyth[139]

**Royal Devon and Exeter Hospital team**
Letizia Zitter[140], Sarah Benyon[140], Suzie Marriott[140], Linda Park[140], Samantha Keenan[140], Elizabeth Gordon[140], Helen Quinn[140] & Kizzy Baines[140]

**The Royal Papworth Hospital team**
Lenka Cagova[141], Adama Fofano[141], Lucie Garner[141], Helen Holcombe[141], Sue Mepham[141], Alice Michael Mitchell[141], Lucy Mwaura[141], Krithivasan Praman[141], Alain Vuylsteke[141] & Julie Zamikula[141]

**Ipswich Hospital team**
Bally Purewal[142], Vanessa Rivers[142] & Stephanie Bell[142]

**Southmead Hospital team**
Hayley Blakemore[143], Borislava Borislavova[143], Beverley Faulkner[143], Emma Gendall[143], Elizabeth Goff[143], Kati Hayes[143], Matt Thomas[143], Ruth Worner[143], Kerry Smith[143] & Deanna Stephens[143]

**Milton Keynes University Hospital team**
Louise Mew[144], Esther Mwaura[144], Richard Stewart[144], Felicity Williams[144], Lynn Wren[144] & Sara-Beth Sutherland[144]

**Royal Hampshire County Hospital team**
Emily Bevan[145], Jane Martin[145], Dawn Trodd[145], Geoff Watson[145] & Caroline Wrey Brown[145]

**Great Ormond St Hospital and UCL Great Ormond St Institute of Child Health NIHR Biomedical Research Centre team**
Olugbenga Akinkugbe[146], Alasdair Bamford[146], Emily Beech[146], Holly Belfield[146], Michael Bell[146], Charlene Davies[146], Gareth A. L. Jones[146], Tara McHugh[146], Hamza Meghari[146], Lauran O'Neill[146], Mark J. Peters[146], Samiran Ray[146] & Ana Luisa Tomas[146]

**Stoke Mandeville Hospital team**
Iona Burn[147], Geraldine Hambrook[147], Katarina Manso[147], Ruth Penn[147], Pradeep Shanmugasundaram[147], Julie Tebbutt[147] & Danielle Thornton[147]

**University Hospital of Wales team**
Jade Cole[148], Michelle Davies[148], Rhys Davies[148], Donna Duffin[148], Helen Hill[148], Ben Player[148], Emma Thomas[148] & Angharad Williams[148]

**Basingstoke and North Hampshire Hospital team**
Denise Griffin[149], Nycola Muchenje[149], Mcdonald Mupudzi[149], Richard Partridge[149], Jo-Anna Conyngham[149], Rachel Thomas[149], Mary Wright[149] & Maria Alvarez Corral[149]

**Arrowe Park Hospital team**
Reni Jacob[150], Cathy Jones[150] & Craig Denmade[150]

**Chesterfield Royal Hospital Foundation Trust team**
Sarah Beavis[151], Katie Dale[151], Rachel Gascoyne[151], Joanne Hawes[151], Kelly Pritchard[151], Lesley Stevenson[151] & Amanda Whileman[151]

**Musgrove Park Hospital team**
Patricia Doble[152], Joanne Hutter[152], Corinne Pawley[152], Charmaine Shovelton[152] & Marius Vaida[152]

**Peterborough City Hospital team**
Deborah Butcher[153,154], Susie O'Sullivan[153,154] & Nicola Butterworth-Cowin[153,154]

**Royal Hallamshire Hospital and Northern General Hospital team**
Norfaizan Ahmad[155], Joann Barker[155], Kris Bauchmuller[155], Sarah Bird[155], Kay Cawthron[155], Kate Harrington[155], Yvonne Jackson[155], Faith Kibutu[155], Becky Lenagh[155], Shamiso Masuko[155], Gary H. Mills[155], Ajay Raithatha[155], Matthew Wiles[155], Jayne Willson[155], Helen Newell[155], Alison Lye[155], Lorenza Nwafor[155], Claire Jarman[155], Sarah Rowland-Jones[155], David Foote[155], Joby Cole[155], Roger Thompson[155], James Watson[155], Lisa Hesseldon[155], Irene Macharia[155], Luke Chetam[155], Jacqui Smith[155], Amber Ford[155], Samantha Anderson[155], Kathryn Birchall[155], Kay Housley[155], Sara Walker[155], Leanne Milner[155], Helena Hanratty[155], Helen Trower[155], Patrick Phillips[155], Simon Oxspring[155] & Ben Donne[155]

**Dumfries and Galloway Royal Infirmary team**
Catherine Jardine[156], Dewi Williams[156] & Alasdair Hay[156]

**Royal Bolton Hospital team**
Rebecca Flanagan[157], Gareth Hughes[157], Scott Latham[157], Emma McKenna[157], Jennifer Anderson[157], Robert Hull[157] & Kat Rhead[157]

**Lister Hospital team**
Carina Cruz[158] & Natalie Pattison[158]

**Craigavon Area Hospital team**
Rob Charnock[159], Denise McFarland[159] & Denise Cosgrove[159]

**Southport and Formby District General Hospital team**
Ashar Ahmed[160], Anna Morris[160], Srinivas Jakkula[160] & Arvind Nune[160]

**Calderdale Royal Hospital team**
Asifa Ali[161,162], Megan Brady[161,162], Sam Dale[161,162], Annalisa Dance[161,162], Lisa Gledhill[161,162], Jill Greig[161,162], Kathryn Hanson[161,162], Kelly Holdroyd[161,162], Marie Home[161,162], Diane Kelly[161,162], Ross Kitson[161,162], Lear Matapure[161,162], Deborah Melia[161,162], Samantha Mellor[161,162], Tonicha Nortcliffe[161,162], Jez Pinnell[161,162], Matthew Robinson[161,162], Lisa Shaw[161,162], Ryan Shaw[161,162], Lesley Thomis[161,162], Alison Wilson[161,162], Tracy Wood[161,162], Lee-Ann Bayo[161,162], Ekta Merwaha[161,162], Tahira Ishaq[161,162] & Sarah Hanley[161,162]

**Prince Charles Hospital team**
Bethan Deacon[163], Meg Hibbert[163], Carla Pothecary[163], Dariusz Tetla[163], Christopher Woodford[163], Latha Durga[163] & Gareth Kennard-Holden[163]

**Royal Bournemouth Hospital team**
Debbie Branney[164], Jordan Frankham[164], Sally Pitts[164] & Nigel White[164]

**Royal Preston Hospital team**
Shondipon Laha[165], Mark Verlander[165] & Alexandra Williams[165]

**Whittington Hospital team**
Abdelhakim Altabaibeh[166], Ana Alvaro[166], Kayleigh Gilbert[166], Louise Ma[166], Loreta Mostoles[166], Chetan Parmar[166], Kathryn Simpson[166], Champa Jetha[166], Lauren Booker[166] & Anezka Pratley[166]

**Princess Royal Hospital team**
Colene Adams[167], Anita Agasou[167], Tracie Arden[167], Amy Bowes[167], Pauline Boyle[167], Mandy Beekes[167], Heather Button[167], Nigel Capps[167], Mandy Carnahan[167], Anne Carter[167], Danielle Childs[167], Denise Donaldson[167], Kelly Hard[167], Fran Hurford[167], Yasmin Hussain[167], Ayesha Javaid[167], James Jones[167], Sanal Jose[167], Michael Leigh[167], Terry Martin[167], Helen Millward[167], Nichola Motherwell[167], Rachel Rikunenko[167], Jo Stickley[167], Julie Summers[167], Louise Ting[167], Helen Tivenan[167], Louise Tonks[167], Rebecca Wilcox[167], Denise Skinner[168], Jane Gaylard[168], Dee Mullan[168] & Julie Newman[168]

**Macclesfield District General Hospital team**
Maureen Holland[169], Natalie Keenan[169], Marc Lyons[169], Helen Wassall[169], Chris Marsh[169], Mervin Mahenthran[169], Emma Carter[169] & Thomas Kong[169]

**Royal Surrey County Hospital team**
Helen Blackman[170], Ben Creagh-Brown[170], Sinead Donlon[170], Natalia Michalak-Glinska[170], Sheila Mtuwa[170], Veronika Pristopan[170], Armorel Salberg[170], Eleanor Smith[170], Sarah Stone[170], Charles Piercy[170], Jerik Verula[170], Dorota Burda[170], Rugia Montaser[170], Lesley Harden[170], Irving Mayangao[170], Cheryl Marriott[170], Paul Bradley[170] & Celia Harris[170]

**Hereford County Hospital team**
Susan Anderson[171], Eleanor Andrews[171], Janine Birch[171], Emma Collins[171], Kate Hammerton[171] & Ryan O'Leary[171]

**University Hospital of North Tees team**
Michele Clark[172] & Sarah Purvis[172]

**Lincoln County Hospital team**
Russell Barber[173], Claire Hewitt[173], Annette Hilldrith[173], Karen Jackson-Lawrence[173], Sarah Shepardson[173], Maryanne Wills[173], Susan Butler[173], Silvia Tavares[173], Amy Cunningham[173], Julia Hindale[173] & Sarwat Arif[173]

**Royal Cornwall Hospital team**
Sarah Bean[174], Karen Burt[174] & Michael Spivey[174]

**Royal United Hospital team**
Carrie Demetriou[175], Charlotte Eckbad[175], Sarah Hierons[175], Lucy Howie[175], Sarah Mitchard[175], Lidia Ramos[175], Alfredo Serrano-Ruiz[175], Katie White[175] & Fiona Kelly[175]

**Royal Brompton Hospital team**
Daniele Cristiano[176], Natalie Dormand[176], Zohreh Farzad[176], Mahitha Gummadi[176], Kamal Liyanage[176], Brijesh Patel[176], Sara Salmi[176], Geraldine Sloane[176], Vicky Thwaites[176], Mathew Varghese[176] & Anelise C. Zborowski[176]

**University Hospital Crosshouse team**
John Allan[177], Tim Geary[177], Gordon Houston[177], Alistair Meikle[177] & Peter O'Brien[177]

**Basildon Hospital team**
Miranda Forsey[178], Agilan Kaliappan[178], Anne Nicholson[178], Joanne Riches[178], Mark Vertue[178], Miranda Forsey[178], Agilan Kaliappan[178], Anne Nicholson[178], Joanne Riches[178] & Mark Vertue[178]

**Glan Clwyd Hospital team**
Elizabeth Allan[179], Kate Darlington[179], Ffyon Davies[179], Jack Easton[179], Sumit Kumar[179], Richard Lean[179], Daniel Menzies[179], Richard Pugh[179], Xinyi Qiu[179], Llinos Davies[179], Hannah Williams[179], Jeremy Scanlon[179], Gwyneth Davies[179], Callum Mackay[179], Joanne Lewis[179] & Stephanie Rees[179]

**West Middlesex Hospital team**
Metod Oblak[180], Monica Popescu[180] & Mini Thankachen[180]

**Royal Lancaster Infirmary team**
Andrew Higham[181], Kerry Simpson[181] & Jayne Craig[181]

**Western General Hospital team**
Rosie Baruah[182], Sheila Morris[182], Susie Ferguson[182] & Amy Shepherd[182]

**Chelsea and Westminster NHS Foundation Trust team**
Luke Stephen Prockter Moore[183], Marcela Paola Vizcaychipi[183], Laura Gomes de Almeida Martins[183] & Jaime Carungcong[183]

**The Queen Elizabeth Hospital team**
Inthakab Ali Mohamed Ali[184], Karen Beaumont[184], Mark Blunt[184], Zoe Coton[184], Hollie Curgenven[184], Mohamed Elsaadany[184], Kay Fernandes[184], Sameena Mohamed Ally[184], Harini Rangarajan[184], Varun Sarathy[184], Sivarupan Selvanayagam[184], Dave Vedage[184] & Matthew White[184]

**King's Mill Hospital team**
Mandy Gill[185], Paul Paul[185], Valli Ratnam[185], Sarah Shelton[185] & Inez Wynter[185]

**Watford General Hospital team**
Siobhain Carmody[186] & Valerie Joan Page[186]

**University Hospital Wishaw team**
Claire Marie Beith[187], Karen Black[187], Suzanne Clements[187], Alan Morrison[187], Dominic Strachan[187], Margaret Taylor[187], Michelle Clarkson[187], Stuart D'Sylva[187] & Kathryn Norman[187]

**Forth Valley Royal Hospital team**
Fiona Auld[188], Joanne Donnachie[188], Ian Edmond[188], Lynn Prentice[188], Nikole Runciman[188], Dario Salutous[188], Lesley Symon[188], Anne Todd[188], Patricia Turner[188], Abigail Short[188], Laura Sweeney[188], Euan Murdoch[188] & Dhaneesha Senaratne[188]

**George Eliot Hospital NHS Trust team**
Michaela Hill[189], Thogulava Kannan[189] & Laura Wild[189]

**Barnsley Hospital team**
Rikki Crawley[190], Abigail Crew[190], Mishell Cunningham[190], Allison Daniels[190], Laura Harrison[190], Susan Hope[190], Ken Inweregbu[190], Sian Jones[190], Nicola Lancaster[190], Jamie Matthews[190], Alice Nicholson[190] & Gemma Wray[190]

**The Great Western Hospital team**
Helen Langton[191], Rachel Prout[191], Malcolm Watters[191] & Catherine Novis[191]

**Harefield Hospital team**
Anthony Barron[192], Ciara Collins[192], Sundeep Kaul[192], Heather Passmore[192], Claire Prendergast[192], Anna Reed[192], Paula Rogers[192], Rajvinder Shokkar[192], Meriel Woodruff[192], Hayley Middleton[192], Oliver Polgar[192], Claire Nolan[192], Vicky Thwaites[192] & Kanta Mahay[192]

**Rotherham General Hospital team**
Dawn Collier[193], Anil Hormis[193], Victoria Maynard[193], Cheryl Graham[193], Rachel Walker[193] & Victoria Maynard[193]

**Ysbyty Gwynedd team**
Ellen Knights[194], Alicia Price[194], Alice Thomas[194] & Chris Thorpe[194]

**Diana Princess of Wales Hospital team**
Teresa Behan[195], Caroline Burnett[195], Jonathan Hatton[195], Elaine Heeney[195], Atideb Mitra[195], Maria Newton[195], Rachel Pollard[195] & Rachael Stead[195]

**Russell's Hall Hospital team**
Vishal Amin[196], Elena Anastasescu[196], Vikram Anumakonda[196], Komala Karthik[196], Rizwana Kausar[196], Karen Reid[196], Jacqueline Smith[196], Janet Imeson-Wood[196], Denise Skinner[168], Jane Gaylard[168], Dee Mullan[168] & Julie Newman[168]

**St Mary's Hospital team**
Alison Brown[197], Vikki Crickmore[197], Gabor Debreceni[197], Joy Wilkins[197] & Liz Nicol[197]

**University Hospital Lewisham team**
Waqas Khaliq[198], Rosie Reece-Anthony[198] & Mark Birt[198]

**Colchester General Hospital team**
Alison Ghosh[199] & Emma Williams[199]

**Queen Elizabeth the Queen Mother Hospital team**
Louise Allen[200], Eva Beranova[200], Nikki Crisp[200], Joanne Deery[200], Tracy Hazelton[200], Alicia Knight[200], Carly Price[200], Sorrell Tilbey[200], Salah Turki[200] & Sharon Turney[200]

**Royal Albert Edward Infirmary team**
Joshua Cooper[201], Cheryl Finch[201], Sarah Liderth[201], Alison Quinn[201] & Natalia Waddington[201]

**Victoria Hospital team**
Tina Coventry[202], Susan Fowler[202], Michael MacMahon[202] & Amanda McGregor[202]

**Eastbourne District General Hospital team**
Anne Cowley[203,204] & Judith Highgate[203,204]

**Cumberland Infirmary team**
Alison Brown[205], Jane Gregory[205], Susan O'Connell[205], Tim Smith[205] & Luigi Barberis[205]

**New Cross Hospital team**
Shameer Gopal[206], Nichola Harris[206], Victoria Lake[206], Stella Metherell[206] & Elizabeth Radford[206]

**The Princess Alexandra Hospital team**
Amelia Daniel[207], Joanne Finn[207], Rajnish Saha[207], Nikki White[207] & Amy Easthope[207]

**Salisbury District Hospital team**
Phil Donnison[208], Fiona Trim[208] & Beena Eapen[208]

**Dorset County Hospital team**
Jenny Birch[209], Laura Bough[209], Josie Goodsell[209], Rebecca Tutton[209], Patricia Williams[209], Sarah Williams[209] & Barbara Winter-Goodwin[209]

**University College Dublin team**
Ailstair Nichol[210], Kathy Brickell[210], Michelle Smyth[210] & Lorna Murphy[210]

**Glangwili General Hospital team**
Samantha Coetzee[211], Alistair Gales[211], Igor Otahal[211], Meena Raj[211] & Craig Sell[211]

**Gloucestershire Royal Hospital team**
Paula Hilltout[212], Jayne Evitts[212], Amanda Tyler[212] & Joanne Waldron[212]

**Yeovil Hospital team**
Kate Beesley[213], Sarah Board[213], Agnieszka Kubisz-Pudelko[213], Alison Lewis[213], Jess Perry[213], Lucy Pippard[213], Di Wood[213] & Clare Buckley[213]

**Leicester Royal Infirmary team**
Peter Barry[214], Neil Flint[214], Patel Rekha[214] & Dawn Hales[214]

**Royal Manchester Children's Hospital team**
Lara Bunni[215], Claire Jennings[215], Monica Latif[215], Rebecca Marshall[215] & Gayathri Subramanian[215]

**Royal Victoria Hospital team**
Peter J. McGuigan[216], Christopher Wasson[216], Stephanie Finn[216], Jackie Green[216], Erin Collins[216] & Bernadette King[216]

**Wrexham Maelor Hospital team**
Andy Campbell[217], Sara Smuts[217], Joseph Duffield[217], Oliver Smith[217], Lewis Mallon[217] & Claire Watkins[217]

**Walsall Manor Hospital team**
Liam Botfield[218], Joanna Butler[218], Catherine Dexter[218], Jo Fletcher[218], Atul Garg[218], Aditya Kuravi[218], Poonam Ranga[218] & Emma Virgilio[218]

**Darent Valley Hospital team**
Zakaula Belagodu[219], Bridget Fuller[219], Anca Gherman[219], Olumide Olufuwa[219], Remi Paramsothy[219], Carmel Stuart[219], Naomi Oakley[219], Charlotte Kamundi[219], David Tyl[219], Katy Collins[219], Pedro Silva[219], June Taylor[219], Laura King[219], Charlotte Coates[219], Maria Crowley[219], Phillipa Wakefield[219], Jane Beadle[219], Laura Johnson[219], Janet Sargeant[219] & Madeleine Anderson[219]

**Warrington General Hospital team**
Ailbhe Brady[220], Rebekah Chan[220], Jeff Little[220], Shane McIvor[220], Helena Prady[220], Helen Whittle[220] & Bijoy Mathew[220]

**Warwick Hospital team**
Ben Attwood[221] & Penny Parsons[221]

**University Hospitals Coventry and Warwickshire NHS Trust team**
Geraldine Ward[222] & Pamela Bremmer[222]

**University Hospital Monklands team**
West Joe[223], Baird Tracy[223] & Ruddy Jim[223]

**Princess of Wales Hospital team**
Ellie Davies[224], Lisa Roche[224] & Sonia Sathe[224]

**Northwick Park Hospital team**
Catherine Dennis[225], Alastair McGregor[225], Victoria Parris[225], Sinduya Srikaran[225] & Anisha Sukha[225]

**Raigmore Hospital team**
Rachael Campbell[226], Noreen Clarke[226], Jonathan Whiteside[226], Mairi Mascarenhas[226], Avril Donaldson[226], Joanna Matheson[226], Fiona Barrett[226], Marianne O'Hara[226], Laura Okeefe[226] & Clare Bradley[226]

**Royal Free Hospital team**
Christine Eastgate-Jackson[227], Helder Filipe[227], Daniel Martin[227], Amitaa Maharajh[227], Sara Mingo Garcia[227], Glykeria Pakou[227] & Mark De Neef[227]

**Scunthorpe General Hospital team**
Kathy Dent[228], Elizabeth Horsley[228], Muhammad Nauman Akhtar[228], Sandra Pearson[228], Dorota Potoczna[228] & Sue Spencer[228]

**West Cumberland Hospital team**
Melanie Clapham[229], Rosemary Harper[229], Una Poultney[229], Polly Rice[229], Tim Smith[229], Rachel Mutch[229] & Luigi Barberis[229]

**Airedale General Hospital team**
Lisa Armstrong[230], Hayley Bates[230], Emma Dooks[230], Fiona Farquhar[230], Brigid Hairsine[230], Chantal McParland[230] & Sophie Packham[230]

**Birmingham Children's Hospital team**
Rehana Bi[231], Barney Scholefield[231] & Lydia Ashton[231]

**Liverpool Heart and Chest Hospital team**
Linsha George[232], Sophie Twiss[232] & David Wright[232]

**Pilgrim Hospital team**
Manish Chablani[233], Amy Kirkby[233] & Kimberley Netherton[233]

**Prince Philip Hospital team**
Kim Davies[234], Linda O'Brien[234], Zohra Omar[234], Igor Otahal[234], Emma Perkins[234], Tracy Lewis[234] & Isobel Sutherland[234]

**Furness General Hospital team**
Karen Burns[235] & Andrew Higham[235]

**Scarborough General Hospital team**
Ben Chandler[236], Kerry Elliott[236], Janine Mallinson[236] & Alison Turnbull[236]

**Southend University Hospital team**
Prisca Gondo[237], Bernard Hadebe[237], Abdul Kayani[237] & Bridgett Masunda[237]

**Alder Hey Children's Hospital team**
Taya Anderson[238], Dan Hawcutt[238], Laura O'Malley[238], Laura Rad[238], Naomi Rogers[238], Paula Saunderson[238], Kathryn Sian Allison[238], Deborah Afolabi[238], Jennifer Whitbread[238], Dawn Jones[238] & Rachael Dore[238]

**Torbay Hospital team**
Matthew Halkes[239], Pauline Mercer[239] & Lorraine Thornton[239]

**Borders General Hospital team**
Joy Dawson[240], Sweyn Garrioch[240], Melanie Tolson[240] & Jonathan Aldridge[240]

**Kent and Canterbury Hospital team**
Ritoo Kapoor[241], David Loader[241] & Karen Castle[241]

**West Suffolk Hospital team**
Sally Humphreys[242] & Ruth Tampsett[242]

**James Paget University Hospital NHS Trust team**
Katherine Mackintosh[243], Amanda Ayers[243], Wendy Harrison[243] & Julie North[243]

**The Christie NHS Foundation Trust team**
Suzanne Allibone[244], Roman Genetu[244], Vidya Kasipandian[244], Amit Patel[244], Ainhi Mac[244], Anthony Murphy[244], Parisa Mahjoob[244], Roonak Nazari[244], Lucy Worsley[244] & Andrew Fagan[244]

**The Royal Marsden Hospital team**
Thomas Bemand[245], Ethel Black[245], Arnold Dela Rosa[245], Ryan Howle[245], Shaman Jhanji[245], Ravishankar Rao Baikady[245], Kate Colette Tatham[245] & Benjamin Thomas[245]

**University Hospital Hairmyres team**
Dina Bell[246], Rosalind Boyle[246], Katie Douglas[246], Lynn Glass[246], Emma Lee[246], Liz Lennon[246] & Austin Rattray[246]

**Withybush General Hospital team**
Abigail Taylor[247], Rachel Anne Hughes[247], Helen Thomas[247], Alun Rees[247], Michaela Duskova[247], Janet Phipps[247], Suzanne Brooks[247] & Michelle Edwards[247]

**Ealing Hospital team**
Victoria Parris[248], Sheena Quaid[248] & Ekaterina Watson[248]

**North Devon District Hospital team**
Adam Brayne[249], Emma Fisher[249], Jane Hunt[249], Peter Jackson[249], Duncan Kaye[249], Nicholas Love[249], Juliet Parkin[249], Victoria Tuckey[249], Lynne van Koutrik[249], Sasha Carter[249], Benedict Andrew[249], Louise Findlay[249] & Katie Adams[249]

**St John's Hospital Livingston team**
Jen Service[250], Alison Williams[250], Claire Cheyne[250], Anne Saunderson[250], Sam Moultrie[250] & Miranda Odam[250]

**Northampton General Hospital NHS Trust team**
Kathryn Hall[251], Isheunesu Mapfunde[251], Charlotte Willis[251] & Alex Lyon[251]

**Harrogate and District NHS Foundation Trust team**
Chunda Sri-Chandana[252], Joslan Scherewode[252], Lorraine Stephenson[252] & Sarah Marsh[252]

**National Hospital for Neurology and Neurosurgery team**
David Brealey[253], John Hardy[253], Henry Houlden[253], Eleanor Moncur[253], Eamon Raith[253], Ambreen Tariq[253] & Arianna Tucci[253]

**Bronglais General Hospital team**
Maria Hobrok[254], Ronda Loosley[254], Heather McGuinness[254], Helen Tench[254] & Rebecca Wolf-Roberts[254]

**Golden Jubilee National Hospital team**
Val Irvine[255] & Benjamin Shelley[255]

**Homerton University Hospital Foundation NHS Trust team**
Amy Easthope[256], Claire Gorman[256], Abhinav Gupta[256], Elizabeth Timlick[256] & Rebecca Brady[256]

**Royal Hospital for Children team**
Colin Begg[38] & Barry Milligan[38]

**Sheffield Children's Hospital team**
Arianna Bellini[257], Jade Bryant[257], Anton Mayer[257], Amy Pickard[257], Nicholas Roe[257], Jason Sowter[257] & Alex Howlett[257]

**The Royal Alexandra Children's Hospital team**
Katy Fidler[258], Emma Tagliavini[258] & Kevin Donnelly[258]

[36]Roslin Institute, University of Edinburgh, Edinburgh, UK. [37]Intensive Care Unit, Royal Infirmary of Edinburgh, Edinburgh, UK. [38]Royal Hospital for Children, Glasgow, UK. [39]William Harvey Research Institute, Barts and the London School of Medicine and Dentistry, Queen Mary University of London, London, UK. [40]Centre for Tropical Medicine and Global Health, Nuffield Department of Medicine, University of Oxford, Oxford, UK. [41]Wellcome Centre for Human Genetics, University of Oxford, Oxford, UK. [42]Prince of Wales Hospital, Hong Kong, China. [43]Department of Critical Care Medicine, Queen's University and Kingston Health Sciences Centre, Kingston, Ontario, Canada. [44]Wellcome–Wolfson Institute for Experimental Medicine, Queen's University Belfast, Belfast, UK. [45]Department of Intensive Care Medicine, Royal Victoria Hospital, Belfast, UK. [46]UCL Centre for Human Health and Performance, London, UK. [47]Clinical Research Centre at St Vincent's University Hospital, University College Dublin, Dublin, Ireland. [48]National Heart and Lung Institute, Imperial College London, London, UK. [49]Imperial College Healthcare NHS Trust: London, London, UK. [50]Heart Institute, University of São Paulo, São Paulo, Brazil. [51]MRC Human Genetics Unit, Institute of Genetics and Molecular Medicine, University of Edinburgh, Western General Hospital, Edinburgh, UK. [52]Intensive Care National Audit and Research Centre, London, UK. [53]NIHR Health Protection Research Unit for Emerging and Zoonotic Infections, Institute of Infection, Veterinary and Ecological Sciences, University of Liverpool, Liverpool, UK. [54]Respiratory Medicine and Institute in the Park, Alder Hey Children's Hospital and University of Liverpool, Liverpool, UK. [55]Department of Intensive Care Medicine, Guy's and St Thomas' NHS Foundation Trust, London, UK. [56]Department of Medicine, University of Cambridge, Cambridge, UK. [57]NIHR Clinical Research Network (CRN), Hammersmith Hospital, London, UK. [58]Cambridge University Hospitals NHS Foundation Trust, Cambridge, UK. [59]Edinburgh Clinical Research

Facility, Western General Hospital, University of Edinburgh, Edinburgh, UK. [60]Biostatistics Group, State Key Laboratory of Biocontrol, School of Life Sciences, Sun Yat-sen University, Guangzhou, China. [61]Department of Infectious Diseases, Leiden University Medical Center, Leiden, The Netherlands. [62]Guys and St Thomas' Hospital, London, UK. [63]Barts Health NHS Trust, London, UK. [64]James Cook University Hospital, Middlesbrough, UK. [65]Royal Stoke University Hospital, Stoke-on-Trent, UK. [66]North Middlesex University Hospital NHS Trust, London, UK. [67]North Middlesex University Hospital NHS Trust, London, UK. [68]The Royal Liverpool University Hospital, Liverpool, UK. [69]King's College Hospital, London, UK. [70]Charing Cross Hospital, St Mary's Hospital and Hammersmith Hospital, London, UK. [71]Nottingham University Hospital, Nottingham, UK. [72]John Radcliffe Hospital, Oxford, UK. [73]Kingston Hospital, Kingston-upon-Thames, UK. [74]Kingston Hospital, Kingston-upon-Thames, UK. [75]Royal Infirmary of Edinburgh, Edinburgh, UK. [76]Queen Alexandra Hospital, Portsmouth, UK. [77]Morriston Hospital, Swansea, UK. [78]Addenbrooke's Hospital, Cambridge, UK. [79]BHRUT (Barking Havering)—Queen's Hospital and King George Hospital, Romford, UK. [80]Royal Sussex County Hospital, Brighton, UK. [81]Queen Elizabeth Hospital, Birmingham, UK. [82]Queen Elizabeth Hospital, Woolwich, London, UK. [83]St George's Hospital, London, UK. [84]Stepping Hill Hospital, Stockport, UK. [85]Countess of Chester Hospital, Chester, UK. [86]Royal Blackburn Teaching Hospital, Blackburn, UK. [87]The Tunbridge Wells Hospital and Maidstone Hospital, Tunbridge Wells, UK. [88]Royal Gwent Hospital, Newport, UK. [89]Pinderfields General Hospital, Wakefield, UK. [90]Royal Berkshire NHS Foundation Trust, Reading, UK. [91]Broomfield Hospital, Chelmsford, UK. [92]Northumbria Healthcare NHS Foundation Trust, North Shields, UK. [93]Whiston Hospital, Prescot, UK. [94]Croydon University Hospital, Croydon, UK. [95]York Hospital, York, UK. [96]Heartlands Hospital, Birmingham, UK. [97]Ashford and St Peter's Hospital, Chertsey, UK. [98]Barnet Hospital, London, UK. [99]East Surrey Hospital, Redhill, UK. [100]Ninewells Hospital, Dundee, UK. [101]Worthing Hospital, Worthing, UK. [102]St Richard's Hospital, Chichester, UK. [103]Southampton General Hospital, Southampton, UK. [104]The Alexandra Hospital, Redditch and Worcester Royal Hospital, Worcester, UK. [105]Sandwell General Hospital and City Hospital, Birmingham, UK. [106]Blackpool Victoria Hospital, Blackpool, UK. [107]Royal Glamorgan Hospital, Pontyclun, UK. [108]The Royal Oldham Hospital, Manchester, UK. [109]Glasgow Royal Infirmary, Glasgow, UK. [110]St James's University Hospital and Leeds General Infirmary, Leeds, UK. [111]University Hospital North Durham, Durham, UK. [112]Darlington Memorial Hospital, Darlington, UK. [113]Fairfield General Hospital, Bury, UK. [114]Wythenshawe Hospital, Manchester, UK. [115]Royal Alexandra Hospital, Paisley, UK. [116]Good Hope Hospital, Birmingham, UK. [117]Tameside General Hospital, Ashton-under-Lyne, UK. [118]Royal Derby Hospital, Derby, UK. [119]Medway Maritime Hospital, Gillingham, UK. [120]Royal Victoria Infirmary, Newcastle-upon-Tyne, UK. [121]Poole Hospital, Poole, UK. [122]Bedford Hospital, Bedford, UK. [123]Queens Hospital Burton, Burton-on-Trent, UK. [124]North Manchester General Hospital, Manchester, UK. [125]Aberdeen Royal Infirmary, Aberdeen, UK. [126]Derriford Hospital, Plymouth, UK. [127]Manchester Royal Infirmary, Manchester, UK. [128]Salford Royal Hospital, Manchester, UK. [129]William Harvey Hospital, Ashford, UK. [130]Queen Elizabeth University Hospital, Glasgow, UK. [131]Bradford Royal Infirmary, Bradford, UK. [132]Bristol Royal Infirmary, Bristol, UK. [133]Norfolk and Norwich University Hospital (NNUH), Norwich, UK. [134]Queen Elizabeth Hospital Gateshead, Gateshead, UK. [135]Sunderland Royal Hospital, Sunderland, UK. [136]Aintree University Hospital, Liverpool, UK. [137]Hull Royal Infirmary, Hull, UK. [138]Hull Royal Infirmary, Hull, UK. [139]University College Hospital, London, UK. [140]Royal Devon and Exeter Hospital, Exeter, UK. [141]The Royal Papworth Hospital, Cambridge, UK. [142]Ipswich Hospital, Ipswich, UK. [143]Southmead Hospital, Bristol, UK. [144]Milton Keynes University Hospital, Milton Keynes, UK. [145]Royal Hampshire County Hospital, Winchester, UK. [146]Great Ormond St Hospital and UCL Great Ormond St Institute of Child Health NIHR Biomedical Research Centre, London, UK. [147]Stoke Mandeville Hospital, Aylesbury, UK. [148]University Hospital of Wales, Cardiff, UK. [149]Basingstoke and North Hampshire Hospital, Basingstoke, UK. [150]Arrowe Park Hospital, Wirral, UK. [151]Chesterfield Royal Hospital Foundation Trust, Chesterfield, UK. [152]Musgrove Park Hospital, Taunton, UK. [153]Peterborough City Hospital, Peterborough, UK. [154]Hinchingbrooke Hospital, Huntingdon, UK. [155]Royal Hallamshire Hospital and Northern General Hospital, Sheffield, UK. [156]Dumfries and Galloway Royal Infirmary, Dumfries, UK. [157]Royal Bolton Hospital, Bolton, UK. [158]Lister Hospital, Stevenage, UK. [159]Craigavon Area Hospital, Craigavon, UK. [160]Southport and Formby District General Hospital, Ormskirk, UK. [161]Calderdale Royal Hospital, Halifax, UK. [162]Huddersfield Royal Infirmary, Huddersfield, UK. [163]Prince Charles Hospital, Merthyr Tydfil, UK. [164]Royal Bournemouth Hospital, Bournemouth, UK. [165]Royal Preston Hospital, Preston, UK. [166]Whittington Hospital, London, UK. [167]Princess Royal Hospital, Telford and Royal Shrewsbury Hospital, Shrewsbury, UK. [168]Princess Royal Hospital, Haywards Heath, UK. [169]Macclesfield District General Hospital, Macclesfield, UK. [170]Royal Surrey County Hospital, Guildford, UK. [171]Hereford County Hospital, Hereford, UK. [172]University Hospital of North Tees, Stockton-on-Tees, UK. [173]Lincoln County Hospital, Lincoln, UK. [174]Royal Cornwall Hospital, Truro, UK. [175]Royal United Hospital, Bath, UK. [176]Royal Brompton Hospital, London, UK. [177]University Hospital Crosshouse, Kilmarnock, UK. [178]Basildon Hospital, Basildon, UK. [179]Glan Clwyd Hospital, Bodelwyddan, UK. [180]West Middlesex Hospital, Isleworth, UK. [181]Royal Lancaster Infirmary, Lancaster, UK. [182]Western General Hospital, Edinburgh, UK. [183]Chelsea and Westminster NHS Foundation Trust, London, UK. [184]The Queen Elizabeth Hospital, King's Lynn, UK. [185]King's Mill Hospital, Nottingham, UK. [186]Watford General Hospital, Watford, UK. [187]University Hospital Wishaw, Wishaw, UK. [188]Forth Valley Royal Hospital, Falkirk, UK. [189]George Eliot Hospital NHS Trust, Nuneaton, UK. [190]Barnsley Hospital, Barnsley, UK. [191]The Great Western Hospital, Swindon, UK. [192]Harefield Hospital, London, UK. [193]Rotherham General Hospital, Rotherham, UK. [194]Ysbyty Gwynedd, Bangor, UK. [195]Diana Princess of Wales Hospital, Grimsby, UK. [196]Russell's Hall Hospital, Dudley, UK. [197]St Mary's Hospital, Newport, UK. [198]University Hospital Lewisham, London, UK. [199]Colchester General Hospital, Colchester, UK. [200]Queen Elizabeth the Queen Mother Hospital, Margate, UK. [201]Royal Albert Edward Infirmary, Wigan, UK. [202]Victoria Hospital, Kirkcaldy, UK. [203]Eastbourne District General Hospital, Eastbourne, UK. [204]Conquest Hospital, St Leonards-on-Sea, UK. [205]Cumberland Infirmary, Carlisle, UK. [206]New Cross Hospital, Wolverhampton, UK. [207]The Princess Alexandra Hospital, Harlow, UK. [208]Salisbury District Hospital, Salisbury, UK. [209]Dorset County Hospital, Dorchester, UK. [210]University College Dublin, St Vincent's University Hospital, Dublin, Ireland. [211]Glangwili General Hospital, Carmarthen, UK. [212]Gloucestershire Royal Hospital, Gloucester, UK. [213]Yeovil Hospital, Yeovil, UK. [214]Leicester Royal Infirmary, Leicester, UK. [215]Royal Manchester Children's Hospital, Manchester, UK. [216]Royal Victoria Hospital, Belfast, UK. [217]Wrexham Maelor Hospital, Wrexham, UK. [218]Walsall Manor Hospital, Walsall, UK. [219]Darent Valley Hospital, Dartford, UK. [220]Warrington General Hospital, Warrington, UK. [221]Warwick Hospital, Warwick, UK. [222]University Hospitals Coventry and Warwickshire NHS Trust, Coventry, UK. [223]University Hospital Monklands, Airdrie, UK. [224]Princess of Wales Hospital, Llantrisant, UK. [225]Northwick Park Hospital, London, UK. [226]Raigmore Hospital, Inverness, UK. [227]Royal Free Hospital, London, UK. [228]Scunthorpe General Hospital, Scunthorpe, UK. [229]West Cumberland Hospital, Whitehaven, UK. [230]Airedale General Hospital, Keighley, UK. [231]Birmingham Children's Hospital, Birmingham, UK. [232]Liverpool Heart and Chest Hospital, Liverpool, UK. [233]Pilgrim Hospital, Lincoln, UK. [234]Prince Philip Hospital, Llanelli, UK. [235]Furness General Hospital, Barrow-in-Furness, UK. [236]Scarborough General Hospital, Scarborough, UK. [237]Southend University Hospital, Westcliff-on-Sea, UK. [238]Alder Hey Children's Hospital, Liverpool, UK. [239]Torbay Hospital, Torquay, UK. [240]Borders General Hospital, Melrose, UK. [241]Kent and Canterbury Hospital, Canterbury, UK. [242]West Suffolk Hospital, Bury St Edmunds, UK. [243]James Paget University Hospital NHS Trust, Great Yarmouth, UK. [244]The Christie NHS Foundation Trust, Manchester, UK. [245]The Royal Marsden Hospital, London, UK. [246]University Hospital Hairmyres, East Kilbride, UK. [247]Withybush General Hospital, Haverfordwest, Wales, UK. [248]Ealing Hospital, Southall, UK. [249]North Devon District Hospital, Barnstaple, UK. [250]St John's Hospital Livingston, Livingston, UK. [251]Northampton General Hospital NHS Trust, Northampton, UK. [252]Harrogate and District NHS Foundation Trust, Harrogate, UK. [253]National Hospital for Neurology and Neurosurgery, London, UK. [254]Bronglais General Hospital, Aberystwyth, UK. [255]Golden Jubilee National Hospital, Clydebank, UK. [256]Homerton University Hospital Foundation NHS Trust, London, UK. [257]Sheffield Children's Hospital, Sheffield, UK. [258]The Royal Alexandra Children's Hospital, Brighton, UK.

**23andMe investigators**

**Janie F. Shelton**[259]**, Anjali J. Shastri**[259]**, Chelsea Ye**[259]**, Catherine H. Weldon**[259]**, Teresa Filshtein-Sonmez**[259]**, Daniella Coker**[259]**, Antony Symons**[259]**, Jorge Esparza-Gordillo**[260]**, Stella Aslibekyan**[259] **& Adam Auton**[259]

[259]23andMe, Sunnyvale, CA, USA. [260]Human Genetics R&D and Target Sciences R&D, GSK Medicines Research Centre, Stevenage, UK.

**COVID-19 Human Genetics Initiative**

**Gita A. Pathak**[261]**, Juha Karjalainen**[262]**, Christine Stevens**[263]**, Shea J. Andrews**[264]**, Masahiro Kanai**[263]**, Mattia Cordioli**[262]**, Renato Polimanti**[261]**, Matti Pirinen**[262]**, Nadia Harerimana**[264]**, Kumar Veerapen**[263]**, Brooke Wolford**[265]**, Huy Nguyen**[263]**, Matthew Solomonson**[263]**, Rachel G. Liao**[263]**, Karolina Chwialkowska**[266]**, Amy Trankiem**[263]**, Mary K. Balaconis**[263]**, Caroline Hayward**[267]**, Anne Richmond**[267]**, Archie Campbell**[267]**, Marcela Morris**[268]**, Chloe Fawns-Ritchie**[267]**, Joseph T. Glessner**[269,270]**, Douglas M. Shaw**[271]**, Xiao Chang**[269]**, Hannah Polikowski**[271]**, Lauren E. Petty**[271]**, Hung-Hsin Chen**[271]**, Zhu Wanying**[271]**, Hakon Hakonarson**[269,270]**, David J. Porteous**[267]**, Jennifer Below**[271]**, Kari North**[272]**, Joseph B. McCormick**[268]**, Paul R. H. J. Timmers**[267]**, James F. Wilson**[267]**, Albert Tenesa**[267,273]**, Kenton D'Mellow**[273]**, Shona M. Kerr**[267]**, Mari E. K. Niemi**[262]**, Lindokuhle Nkambul**[263,274]**, Kathrin Aprile von Hohenstaufen**[275]**, Ali Sobh**[276]**, Madonna M. Eltoukhy**[277]**, Amr M. Yassen**[278]**, Mohamed A. F. Hegazy**[279]**, Kamal Okasha**[280]**, Mohammed A. Eid**[281]**, Hanteera S. Moahmed**[282]**, Doaa Shahin**[283]**, Yasser M. El-Sherbiny**[283,284]**, Tamer A. Elhadidy**[285]**, Mohamed S. Abd Elghafar**[286]**, Jehan J. El-Jawhari**[283,284]**, Attia A. S. Mohamed**[277]**, Marwa H. Elnagdy**[287]**, Amr Samir**[279]**, Mahmoud Abdel-Aziz**[288]**, Walid T. Khafaga**[289]**, Walaa M. El-Lawaty**[282]**, Mohamed S. Torky**[282]**, Mohamed R. El-shanshory**[290]**, Chiara Batini**[291]**, Paul H. Lee**[291]**, Nick Shrine**[291]**, Alexander T. Williams**[291]**, Martin D. Tobin**[291,292]**, Anna L. Guyatt**[291]**, Catherine John**[291]**, Richard J. Packer**[291]**, Altaf Ali**[291]**, Robert C. Free**[293]**, Xueyang Wang**[291]**, Louise V. Wain**[291]**, Edward J. Hollox**[294]**, Laura D. Venn**[291]**, Catherine E. Bee**[291]**, Emma L. Adams**[291]**, Ahmadreza Niavarani**[295]**, Bahareh Sharififard**[295]**, Rasoul Aliannejad**[296]**, Ali Amirsavadkouhi**[297]**, Zeinab Naderpour**[296]**, Hengameh Ansari Tadi**[298]**, Afshar Etemadi Aleagha**[299]**, Saeideh Ahmadi**[300]**, Seyed Behrooz Mohseni Moghaddam**[301]**, Alireza Adamsara**[302]**, Morteza Saeedi**[303]**, Hamed Abdollahi**[304]**, Abdolmajid Hosseini**[305]**, Pajaree Chariyavilaskul**[306,307]**, Monpat Chamnanphon**[306,308]**, Thitima B. Suttichet**[306]**, Vorasuk Shotelersuk**[309,310]**, Monnat Pongpanich**[311,312]**, Chureerat Phokaew**[309,310,313]**, Wanna Chetruengchai**[309,310]**, Watsamon Jantarabenjakul**[314,315]**, Opass Putcharoen**[314,316]**, Pattama Torvorapanit**[314,316]**, Thanyawee Puthanakit**[315,317]**, Pintip Suchartlikitwong**[317,318]**, Nattiya Hirankarn**[319,320]**, Voraphoi Nilaratanakul**[316,321]**, Pimpayao Sodsai**[319,320]**, Ben M. Brumpton**[322,323,324]**, Kristian Hveem**[322,323]**, Cristen Willer**[265,325,326]**, Wei Zhou**[274,327]**, Tormod Rogne**[328,329,330]**, Erik Solligård**[328,330]**, Bjørn Olav Åsvold**[322,323,324]**, Malak Abedalthagafi**[331]**, Manal Alaamery**[332,333]**, Saleh Alqahtani**[334,335]**, Duna Barakeh**[336]⊠**, Fawz Al Harthi**[331]**, Ebtehal Alsolm**[331]**, Leen Abu Safieh**[331]**, Albandary M. Alowayn**[331]**, Fatimah Alqubaishi**[331]**, Amal Al Mutairi**[331]**, Serghei Mangul**[337]**, Abdulraheem Alshareef**[338]**, Mona Sawaji**[339]**, Mansour Almutairi**[332,333]**, Nora Aljawini**[340]**, Nour Albesher**[340]**, Yaseen M. Arabi**[341]**, Ebrahim S. Mahmoud**[341]**, Amin K. Khattab**[342]**, Roaa T. Halawani**[342]**, Ziab Z. Alahmadey**[342]**, Jehad K. Albakri**[342]**, Walaa A. Felemban**[342]**, Bandar A. Suliman**[338]**, Rana Hasanato**[336]**, Laila Al-Awdah**[343]**, Jahad Alghamdi**[344]**, Deema AlZahrani**[345]**, Sameera AlJohani**[346]**, Hani Al-Afghani**[347]**, May Alrashed**[348]**, Nouf AlDhawi**[345]**, Hadeel AlBardis**[331]**, Sarah Alkwai**[340]**, Moneera Alswailm**[340]**, Faisal Almalki**[345]**, Maha Albeladi**[345]**, Iman Almohammed**[340]**, Eman Barhoush**[349]**, Anoud Albader**[345]**, Salam Massadeh**[332,333]**, Abdulaziz AlMalik**[350]**, Sara Alotaibi**[331]**, Bader Alghamdi**[351]**, Junghyun Jung**[352]**, Mohammad S. Fawzy**[331]**, Yunsung Lee**[353]**, Per Magnus**[353]**, Lill-Iren S. Trogstad**[354]**, Øyvind Helgeland**[355]**, Jennifer R. Harris**[355]**, Massimo Mangino**[356,357]**, Tim D. Spector**[356]**, Emma Duncan**[356]**, Sandra P. Smieszek**[358]**, Bartlomiej P. Przychodzen**[358]**, Christos Polymeropoulos**[358]**, Vasilios Polymeropoulos**[358]**, Mihael H. Polymeropoulos**[358]**, Israel Fernandez-Cadenas**[359]**, Jordi Perez-Tur**[360,361,362]**, Laia Llucià-Carol**[359,363]**, Natalia Cullell**[359,364]**, Elena Muiño**[359]**, Jara Cárcel-Márquez**[359]**, Marta L. DeDiego**[365]**, Lara Lloret Iglesias**[366]**, Anna M. Planas**[363,367]**, Alex Soriano**[368]**, Veronica Rico**[368]**, Daiana Agüero**[368]**, Josep L. Bedini**[368]**, Francisco Lozano**[369]**, Carlos Domingo**[368]**, Veronica Robles**[368]**, Francisca Ruiz-Jaén**[370]**, Leonardo Márquez**[371]**, Juan Gomez**[372]**, Eliecer Coto**[372]**, Guillermo M. Albaiceta**[372]**, Marta García-Clemente**[372]**, David Dalmau**[373]**, Maria J. Arranz**[373]**, Beatriz Dietl**[373]**, Alex Serra-Llovich**[373]**, Pere Soler**[374]**, Roger Colobrán**[374]**,

Andrea Martín-Nalda[374], Alba Parra Martínez[374], David Bernardo[375], Silvia Rojo[376], Aida Fiz-López[375], Elisa Arribas[375], Paloma de la Cal-Sabater[375], Tomás Segura[377], Esther González-Villa[377], Gemma Serrano-Heras[377], Joan Martí-Fàbregas[378], Elena Jiménez-Xarrié[378], Alicia de Felipe Mimbrera[379], Jaime Masjuan[379], Sebastian García-Madrona[379], Anna Domínguez-Mayoral[380,381], Joan Montaner Villalonga[380,381], Paloma Menéndez-Valladares[380,381], Daniel I. Chasman[382,383], Julie E. Buring[382,383], Paul M. Ridker[382,383], Giuliani Franco[382], Howard D. Sesso[382,383], JoAnn E. Manson[382,383], Joseph R. Glessner[269,384], Hakon Hakonarson[269,384,385], Carolina Medina-Gomez[386], Andre G. Uitterlinden[386], M. Arfan Ikram[386], Kati Kristiansson[387], Sami Koskelainen[387], Markus Perola[387,388], Kati Donner[262], Katja Kivinen[262], Aarno Palotie[262], Samuli Ripatti[262,263,389], Sanni Ruotsalainen[262], Mari Kaunisto[262], Tomoko Nakanishi[390,391,392,393], Guillaume Butler-Laporte[390,391], Vincenzo Forgetta[390], David R. Morrison[390], Biswarup Ghosh[390], Laetitia Laurent[390], Alexandre Belisle[390], Danielle Henry[390], Tala Abdullah[390], Olumide Adeleye[390], Noor Mamlouk[390], Nofar Kimchi[390], Zaman Afrasiabi[390], Nardin Rezk[390], Branka Vulesevic[390], Meriem Bouab[390], Charlotte Guzman[390], Louis Petitjean[390], Chris Tselios[390], Xiaoqing Xue[390], Erwin Schurr[390], Jonathan Afilalo[390], Marc Afilalo[390], Maureen Oliveira[390], Bluma Brenner[390], Pierre Lepage[390], Jiannis Ragoussis[390], Daniel Auld[390], Nathalie Brassard[390], Madeleine Durand[390], Michaël Chassé[390], Daniel E. Kaufmann[390], G. Mark Lathrop[390], Vincent Mooser[390], J. Brent Richards[390], Rui Li[390], Darin Adra[390], Souad Rahmouni[394], Michel Georges[394], Michel Moutschen[395], Benoît Misset[394,395], Gilles Darcis[394,395], Julien Guiot[394,395], Julien Guntz[395], Samira Azarzar[394,395], Stéphanie Gofflot[396], Yves Beguin[396], Sabine Claassen[397], Olivier Malaise[395], Pascale Huynen[395], Christelle Meuris[395], Marie Thys[395], Jessica Jacques[395], Philippe Léonard[395], Frederic Frippiat[395], Jean-Baptiste Giot[395], Anne-Sophie Sauvage[395], Christian von Frenckell[395], Yasmine Belhaj[394], Bernard Lambermont[395], Mari E. K. Niemi[262], Sara Pigazzini[262], Lindokuhle Nkambule[263,263,274], Michelle Daya[398], Jonathan Shortt[398], Nicholas Rafaels[398], Stephen J. Wicks[398], Kristy Crooks[398], Kathleen C. Barnes[398], Christopher R. Gignoux[398], Sameer Chavan[398], Triin Laisk[399], Kristi Läll[399], Maarja Lepamets[399], Reedik Mägi[399], Tõnu Esko[399], Ene Reimann[399], Lili Milani[399], Helene Alavere[399], Kristjan Metsalu[399], Mairo Puusepp[399], Andres Metspalu[399], Paul Naaber[400], Edward Laane[401,402], Jaana Pesukova[401], Pärt Peterson[403], Kai Kisand[403], Jekaterina Tabri[404], Raili Allos[404], Kati Hensen[404], Joel Starkopf[402], Inge Ringmets[405], Anu Tamm[402], Anne Kallaste[402], Pierre-Yves Bochud[406], Carlo Rivolta[407,408], Stéphanie Bibert[406], Mathieu Quinodoz[407,408], Dhryata Kamdar[407,408], Noémie Boillat[406], Semira Gonseth Nussle[409], Werner Albrich[410], Noémie Suh[411], Dionysios Neofytos[412], Véronique Erard[413], Cathy Voide[414], Rafael de Cid[415], Iván Galván-Femenía[415], Natalia Blay[415], Anna Carreras[415], Beatriz Cortés[415], Xavier Farré[415], Lauro Sumoy[415], Victor Moreno[416], Josep Maria Mercader[417], Marta Guindo-Martinez[418], David Torrents[418], Manolis Kogevinas[419,420,421,422], Judith Garcia-Aymerich[419,421,422], Gemma Castaño-Vinya ls[419,420,421,422], Carlota Dobaño[419,422], Alessandra Renieri[423,424,425], Francesca Mari[423,424,425], Chiara Fallerini[423,425], Sergio Daga[423,425], Elisa Benetti[423,425], Margherita Baldassarri[423,425], Francesca Fava[423,424,425], Elisa Frullanti[423,425], Floriana Valentino[423,425], Gabriella Doddato[423,425], Annarita Giliberti[423,425], Rossella Tita[424], Sara Amitrano[424], Mirella Bruttini[423,424,425], Susanna Croci[423,425], Ilaria Meloni[423,425], Maria Antonietta Mencarelli[424], Caterina Lo Rizzo[424], Anna Maria Pinto[424], Giada Beligni[423,425], Andrea Tommasi[426], Laura Di Sarno[423,425], Maria Palmieri[423,425], Miriam Lucia Carriero[423,425], Diana Alaverdian[423,425], Stefano Busani[427], Raffaele Bruno[428,429], Marco Vecchia[428], Mary Ann Belli[430], Nicola Picchiotti[431,432], Maurizio Sanarico[433], Marco Gori[432], Simone Furini[433], Stefania Mantovani[428], Serena Ludovisi[434], Mario Umberto Mondelli[428,429], Francesco Castelli[435], Eugenia Quiros-Roldan[435], Melania Degli Antoni[435], Isabella Zanella[436,437], Massimo Vaghi[438], Stefano Rusconi[439,440], Matteo Siano[440], Francesca Montagnani[425,441], Arianna Emiliozzi[442], Massimiliano Fabbiani[441], Barbara Rossetti[441], Elena Bargagli[443], Laura Bergantini[443], Miriana D'Alessandro[443], Paolo Cameli[443], David Bennett[443], Federico Anedda[444], Simona Marcantonio[444], Sabino Scolletta[444], Federico Franchi[444], Maria Antonietta Mazzei[445], Susanna Guerrini[445], Edoardo Conticini[446], Luca Cantarini[446], Bruno Frediani[446], Danilo Tacconi[447], Chiara Spertilli[447], Marco Feri[448], Alice Donati[448], Raffaele Scala[449], Luca Guidelli[449], Genni Spargi[450], Marta Corridi[450], Cesira Nencioni[451], Leonardo Croci[451], Maria Bandini[452], Gian Piero Caldarelli[453], Paolo Piacentini[452], Elena Desanctis[452], Silvia Cappelli[452], Anna Canaccini[454], Agnese Verzuri[454], Valentina Anemoli[454], Agostino Ognibene[455], Alessandro Pancrazzi[455], Maria Lorubbio[455], Antonella D'Arminio Monforte[456], Federica Gaia Miraglia[456], Massimo Girardis[457], Sophie Venturelli[457], Andrea Cossarizza[457], Andrea Antinori[442], Alessandra Vergori[442], Arianna Gabrieli[440], Agostino Riva[439,440], Daniela Francisci[426,458], Elisabetta Schiaroli[426,458], Francesco Paciosi[458], Pier Giorgio Scotton[459], Francesca Andretta[459], Sandro Panese[460], Renzo Scaggiante[461], Francesca Gatti[461], Saverio Giuseppe Parisi[462], Stefano Baratti[462], Matteo Della Monica[463], Carmelo Piscopo[463], Mario Capasso[464,465,466], Roberta Russo[464,465], Immacolata Andolfo[464,465], Achille Iolascon[464,465], Giuseppe Fiorentino[467], Massimo Carella[468], Marco Castori[468], Giuseppe Merla[464,469], Gabriella Maria Squeo[469], Filippo Aucella[470], Pamela Raggi[471], Carmen Marciano[471], Rita Perna[471], Matteo Bassetti[472,473], Antonio Di Biagio[473], Maurizio Sanguinetti[474,475], Luca Masucci[474,475], Serafina Valente[476], Marco Mandalà[477], Alessia Giorli[477], Lorenzo Salerni[477], Patrizia Zucchi[478], Pierpaolo Parravicini[478], Elisabetta Menatti[479], Tullio Trotta[480], Ferdinando Giannattasio[480], Gabriella Coiro[480], Fabio Lena[481], Domenico A. Coviello[482], Cristina Mussini[483], Enrico Martinelli[484], Sandro Mancarella[430], Luisa Tavecchia[430], Lia Crotti[485,486,487,487], Chiara Gabbi[433], Marco Rizzi[488], Franco Maggiolo[488], Diego Ripamonti[488], Tiziana Bachetti[489], Maria Teresa La Rovere[490], Simona Sarzi-Braga[491], Maurizio Bussotti[492], Stefano Ceri[493], Pietro Pinoli[493], Francesco Raimondi[494], Filippo Biscarini[495], Alessandra Stella[495], Kristina Zguro[425], Katia Capitani[425,425], Claudia Suardi[425], Simona Dei[498], Gianfranco Parati[485,486], Sabrina Ravaglia[499], Rosangela Artuso[500], Giordano Bottà[501], Paolo Di Domenico[501], Ilaria Rancan[441], Antonio Perrella[502], Francesco Bianchi[425,502], Davide Romani[452], Paola Bergomi[503], Emanuele Catena[503], Riccardo Colombo[503], Marco Tanfoni[432], Antonella Vincenti[504], Claudio Ferri[505], Davide Grassi[505], Gloria Pessina[506], Mario Tumbarello[425,507], Massimo Di Pietro[508], Ravaglia Sabrina[499], Sauro Luchi[509], Chiara Barbieri[510], Donatella Acquilini[511], Elena Andreucci[500], Francesco Vladimiro Segala[512], Giusy Tiseo[510], Marco Falcone[510], Mirjam Lista[423,425], Monica Poscente[506], Oreste De Vivo[476],

Paola Petrocelli[509], Alessandra Guarnaccia[474], Silvia Baroni[513], Albert V. Smith[265], Andrew P. Boughton[265], Kevin W. Li[265], Jonathon LeFaive[265], Aubrey Annis[265], Anne E. Justice[514], Tooraj Mirshahi[515], Geetha Chittoor[514], Navya Shilpa Josyula[514], Jack A. Kosmicki[516], Manuel A. R. Ferreira[516], Joseph B. Leader[517], Dave J. Carey[515], Matthew C. Gass[517], Julie E. Horowitz[516], Michael N. Cantor[516], Ashish Yadav[516], Aris Baras[516], Goncalo R. Abecasis[516], David A. van Heel[518], Karen A. Hunt[518], Dan Mason[519], Qin Qin Huang[520], Sarah Finer[518], Bhavi Trivedi[518], Christopher J. Griffiths[518], Hilary C. Martin[520], John Wright[519], Richard C. Trembath[518], Nicole Soranzo[522,523,524], Jing Hua Zhao[525], Adam S. Butterworth[523,525,526,527], John Danesh[522,523,525,526,527], Emanuele Di Angelantonio[523,525,526,527], Lude Franke[528], Marike Boezen[528], Patrick Deelen[529], Annique Claringbould[528], Esteban Lopera[528], Robert Warmerdam[528], Judith M. Vonk[530], Irene van Blokland[531], Pauline Lanting[531], Anil P. S. Ori[528,532], Sebastian Zöllner[265], Jiongming Wang[265], Andrew Beck[265], Gina Peloso[533,534], Yuk-Lam Ho[535], Yan V. Sun[536], Jennifer E. Huffman[534], Christopher J. O'Donnell[534], Kelly Cho[535], Phil Tsao[537], J. Michael Gaziano[535], Michel Nivard[538], Eco de Geus[538], Meike Bartels[538], Jouke Jan Hottenga[538], Scott T. Weiss[382], Elizabeth W. Karlson[382], Jordan W. Smoller[417], Robert C. Green[539], Yen-Chen Anne Feng[417], Josep Mercader[539], Shawn N. Murphy[417], James B. Meigs[417], Ann E. Woolley[382], Emma F. Perez[417], Daniel Rader[540], Anurag Verma[540], Marylyn D. Ritchie[540], Binglan Li[537], Shefali S. Verma[540], Anastasia Lucas[540], Yuki Bradford[540], Hugo Zeberg[541,542], Robert Frithiof[543], Michael Hultström[543,544], Miklos Lipcsey[543,543], Lindo Nkambul[263,274,545], Nicolas Tardif[546], Olav Rooyackers[546], Jonathan Grip[546], Tomislav Maricic[542], Konrad J. Karczewski[263,417], Elizabeth G. Atkinson[263,417], Kristin Tsuo[263,417], Nikolas Baya[263,417], Patrick Turley[263,417], Rahul Gupta[263,417], Shawneequa Callier[547], Raymond K. Walters[263,417], Duncan S. Palmer[263,417], Gopal Sarma[263,417], Nathan Cheng[263,417], Wenhan Lu[263,417], Sam Bryant[263,417], Claire Churchhouse[263,417], Caroline Cusick[263], Jacqueline I. Goldstein[263,417], Daniel King[263,417], Cotton Seed[263,417], Hilary Finucane[263,417], Alicia R. Martin[263,417], F. Kyle Satterstrom[263,417], Daniel J. Wilson[548], Jacob Armstrong[548], Justine K. Rudkin[548], Gavin Band[549], Sarah G. Earle[548], Shang-Kuan Lin[548], Nicolas Arning[548], Derrick W. Crook[550], David H. Wyllie[550], Anne Marie O'Connell[552], Chris C. A. Spencer[553], Nils Koelling[553], Mark J. Caulfield[554], Richard H. Scott[554], Tom Fowler[554], Loukas Moutsianas[554], Athanasios Kousathanas[554], Dorota Pasko[554], Susan Walker[554], Augusto Rendon[554], Alex Stuckey[554], Christopher A. Odhams[554], Daniel Rhodes[554], Georgia Chan[554], Prabhu Arumugam[554], Catherine A. Ball[555], Eurie L. Hong[555], Kristin Rand[555], Ahna Girshick[555], Harendra Guturu[555], Asher Haug Baltzell[555], Genevieve Roberts[555], Danny Park[555], Marie Coignet[555], Shannon McCurdy[555], Spencer Knight[555], Raghavendran Partha[555], Brooke Rhead[555], Miao Zhang[555], Nathan Berkowitz[555], Michael Gaddis[555], Keith Noto[555], Luong Ruiz[555], Milos Pavlovic[555], Laura G. Sloofman[264], Alexander W. Charney[264], Noam D. Beckmann[264], Eric E. Schadt[264], Daniel M. Jordan[264], Ryan C. Thompson[264], Kyle Gettler[264], Noura S. Abul-Husn[264], Steven Ascolillo[264], Joseph D. Buxbaum[264], Kumardeep Chaudhary[264], Judy H. Cho[264], Yuval Itan[264], Eimear E. Kenny[264], Gillian M. Belbin[264], Stuart C. Sealfon[264], Robert P. Sebra[264], Irene Salib[264], Brett L. Collins[264], Tess Levy[264], Bari Britvan[264], Katherine Keller[264], Lara Tang[264], Michael Peruggia[264], Liam L. Hiester[264], Kristi Niblo[264], Alexandra Aksentijevich[264], Alexander Labkowsky[264], Avrohom Karp[264], Menachem Zlatopolsky[264], Michael Preuss[264], Ruth J. F. Loos[264], Girish N. Nadkarni[264], Ron Do[264], Clive Hoggart[264], Sam Choi[264], Slayton J. Underwood[264], Paul O'Reilly[264], Laura M. Huckins[264], Marissa Zyndorf[264], Mark J. Daly[263,263] & Benjamin M. Neale[263] & Andrea Ganna[262,263]

[261]Yale University, New Haven, CT, USA. [262]Institute for Molecular Medicine Finland (FIMM), University of Helsinki, Helsinki, Finland. [263]Broad Institute of MIT and Harvard, Cambridge, MA, USA. [264]Icahn School of Medicine at Mount Sinai, New York, NY, USA. [265]University of Michigan, Ann Arbor, MI, USA. [266]Centre for Bioinformatics and Data Analysis, Medical University of Bialystok, Bialystok, Poland. [267]Institute of Genetics and Cancer, University of Edinburgh, Western General Hospital, Edinburgh, UK. [268]University of Texas Health, Houston, TX, USA. [269]Center for Applied Genomics, Children's Hospital of Philadelphia, Philadelphia, PA, USA. [270]Department of Pediatrics, Perelman School of Medicine, University of Pennsylvania, Philadelphia, PA, USA. [271]Vanderbilt University Medical Center, Nashville, TN, USA. [272]University of North Carolina at Chapel Hill, Chapel Hill, NC, USA. [273]Roslin Institute, The Royal (Dick) School of Veterinary Studies, University of Edinburgh, Edinburgh, UK. [274]Analytic and Translational Genetics Unit, Massachusetts General Hospital, Boston, MA, USA. [275]Genolier Innovation Network and Hub, Swiss Medical Network, Genolier Healthcare Campus, Genolier, Switzerland. [276]Department of Pediatrics, Faculty of Medicine, Mansoura University, Mansoura, Egypt. [277]Department of Clinical Pathology, Faculty of Medicine, Tanta University, Tanta, Egypt. [278]Department of Anaesthesia and Critical Care, Faculty of Medicine, Mansoura University, Mansoura, Egypt. [279]Department of Surgery, Faculty of Medicine, Mansoura University, Mansoura, Egypt. [280]Department of Internal Medicine, Faculty of Medicine, Tanta University, Tanta, Egypt. [281]Faculty of Science, Tanta University, Tanta, Egypt. [282]Chest Department, Faculty of Medicine, Tanta University, Tanta, Egypt. [283]Department of Clinical Pathology, Faculty of Medicine, Mansoura University, Mansoura, Egypt. [284]Department of Biosciences, School of Science and Technology, Nottingham Trent University, Nottingham, UK. [285]Chest Department, Faculty of Medicine, Mansoura University, Mansoura, Egypt. [286]Anesthesia, Surgical Intensive Care and Pain Management Department, Faculty of Medicine, Tanta University, Tanta, Egypt. [287]Department of Medical Biochemistry, Faculty of Medicine, Mansoura University, Mansoura, Egypt. [288]Department of Tropical Medicine, Faculty of Medicine, Mansoura University, Mansoura, Egypt. [289]Pediatric and Neonatology, Kafr El-Zayat General Hospital, Kafr El-Zayat, Egypt. [290]Pediatrics Department, Faculty of Medicine, Tanta University, Tanta, Egypt. [291]Department of Health Sciences, University of Leicester, Leicester, UK. [292]Leicester NIHR Biomedical Research Centre, Leicester, UK. [293]Department of Respiratory Sciences, University of Leicester, Leicester, UK. [294]University of Leicester, Leicester, UK. [295]Digestive Oncology Research Center, Digestive Disease Research Institute, Shariati Hospital, Tehran University of Medical Sciences, Tehran, Iran. [296]Department of Pulmonology, School of Medicine, Shariati Hospital, Tehran University of Medical Sciences, Tehran, Iran. [297]Department of Critical Care Medicine, Noorafshar Hospital, Tehran, Iran. [298]Department of Emergency Intensive Care Unit, School of Medicine, Shariati Hospital, Tehran University of Medical Sciences, Tehran, Iran. [299]Department of Anesthesiology,

School of Medicine, Amir Alam Hospital, Tehran University of Medical Sciences, Tehran, Iran. [300]Department of Pulmonology, School of Medicine, Tehran University of Medical Sciences, Tehran, Iran. [301]Department of Pathology, Parseh Pathobiology and Genetics Laboratory, Tehran, Iran. [302]Department of Microbiology, Health and Family Research Center, NIOC Hospital, Tehran, Iran. [303]Department of Emergency Medicine, School of Medicine, Shariati Hospital, Tehran University of Medical Sciences, Tehran, Iran. [304]Department of Anesthesiology, School of Medicine, Tehran University of Medical Sciences, Tehran, Iran. [305]Department of Pathology, Faculty of Medicine, Tehran Azad University, Tehran, Iran. [306]Clinical Pharmacokinetics and Pharmacogenomics Research Unit, Faculty of Medicine, Chulalongkorn University, Bangkok, Thailand. [307]Department of Pharmacology, Faculty of Medicine, Chulalongkorn University, Bangkok, Thailand. [308]Department of Pathology, Faculty of Medicine, Nakornnayok, Srinakharinwirot University, Bangkok, Thailand. [309]Center of Excellence for Medical Genomics, Medical Genomics Cluster, Faculty of Medicine, Chulalongkorn University, Bangkok, Thailand. [310]Excellence Center for Genomics and Precision Medicine, King Chulalongkorn Memorial Hospital, The Thai Red Cross Society, Bangkok, Thailand. [311]Department of Mathematics and Computer Science, Faculty of Science, Chulalongkorn University, Bangkok, Thailand. [312]Omics Sciences and Bioinfomatics Center, Faculty of Science, Chulalongkorn University, Bangkok, Thailand. [313]Research Affairs, Faculty of Medicine, Chulalongkorn University, Bangkok, Thailand. [314]Thai Red Cross Emerging Infectious Diseases Clinical Centre, King Chulalongkorn Memorial Hospital, Bangkok, Thailand. [315]Department of Pediatrics, Faculty of Medicine, Chulalongkorn University, Bangkok, Thailand. [316]Division of Infectious Diseases, Department of Medicine, Faculty of Medicine, Chulalongkorn University, Bangkok, Thailand. [317]Center of Excellence in Pediatric Infectious Diseases and Vaccines, Chulalongkorn University, Bangkok, Thailand. [318]Department of Microbiology, Faculty of Medicine, Chulalongkorn University, Bangkok, Thailand. [319]Immunology Division, Department of Microbiology, Faculty of Medicine, Chulalongkorn University, Bangkok, Thailand. [320]Center of Excellence in Immunology and Immune-mediated Diseases, Department of Microbiology, Faculty of Medicine, Chulalongkorn University, Bangkok, Thailand. [321]Healthcare-associated Infection Research Group STAR (Special Task Force for Activating Research), Chulalongkorn University, Bangkok, Thailand. [322]K.G. Jebsen Center for Genetic Epidemiology, Department of Public Health and Nursing, Norwegian University of Science and Technology (NTNU), Trondheim, Norway. [323]HUNT Research Center, Department of Public Health and Nursing, Norwegian University of Science and Technology (NTNU), Levanger, Norway. [324]Clinic of Medicine, St Olav's Hospital, Trondheim University Hospital, Trondheim, Norway. [325]Division of Cardiovascular Medicine, Department of Internal Medicine, University of Michigan, Ann Arbor, MI, USA. [326]Department of Computational Medicine and Bioinformatics, University of Michigan, Ann Arbor, MI, USA. [327]Program in Medical and Population Genetics, Broad Institute of Harvard and MIT, Cambridge, MA, USA. [328]Gemini Center for Sepsis Research, Department of Circulation and Medical Imaging, Norwegian University of Science and Technology (NTNU), Trondheim, Norway. [329]Department of Chronic Disease Epidemiology and Center for Perinatal, Pediatric and Environmental Epidemiology, Yale School of Public Health, New Haven, CT, USA. [330]Clinic of Anaesthesia and Intensive Care, St Olav's Hospital, Trondheim University Hospital, Trondheim, Norway. [331]Genomics Research Department, Saudi Human Genome Project, King Fahad Medical City and King Abdulaziz City for Science and Technology (KACST), Riyadh, Saudi Arabia. [332]Developmental Medicine Department, King Abdullah International Medical Research Center, King Saud Bin Abdulaziz University for Health Sciences, Ministry of National Guard Health Affairs, Riyadh, Saudi Arabia. [333]Saudi Human Genome Project (SHGP), King Abdulaziz City for Science and Technology (KACST), Satellite Lab at King Abdulaziz Medical City, Ministry of National Guard Health Affairs, Riyadh, Saudi Arabia. [334]The Liver Transplant Unit, King Faisal Specialist Hospital and Research Centre, Riyadh, Saudi Arabia. [335]The Division of Gastroenterology and Hepatology, Johns Hopkins University, Baltimore, MD, USA. [336]Department of Pathology, College of Medicine, King Saud University, Riyadh, Saudi Arabia. [337]Titus Family Department of Clinical Pharmacy, USC School of Pharmacy, University of Southern California, Los Angeles, CA, USA. [338]College of Applied Medical Sciences, Taibah University, Madina, Saudi Arabia. [339]Developmental Medicine Department, King Abdullah International Medical Research Center, King Saud Bin Abdulaziz University for Health Sciences, Ministry of National Guard Health Affairs, Riyadh, Saudi Arabia. [340]KACST-BWH Centre of Excellence for Biomedicine, Joint Centers of Excellence Program, King Abdulaziz City for Science and Technology (KACST), Riyadh, Saudi Arabia. [341]Ministry of the National Guard Health Affairs, King Abdullah International Medical Research Center and King Saud Bin Abdulaziz University for Health Sciences, Riyadh, Saudi Arabia. [342]Ohud Hospital, Ministry of Health, Madinah, Saudi Arabia. [343]Pediatric Infectious Diseases, Children's Specialized Hospital, King Fahad Medical City, Riyadh, Saudi Arabia. [344]The Saudi Biobank, King Abdullah International Medical Research Center, King Saud bin Abdulaziz University for Health Sciences, Ministry of National Guard Health Affairs, Riyadh, Saudi Arabia. [345]Developmental Medicine Department, King Abdullah International Medical Research Center and King Saud Bin Abdulaziz University for Health Sciences, King Abdulaziz Medical City, Ministry of National Guard Health Affairs, Riyadh, Saudi Arabia. [346]Department of Pathology and Laboratory Medicine, King Abdulaziz Medical City, Ministry of National Guard Health Affairs, King Saud Bin Abdulaziz University for Health Sciences and King Abdullah International Medical Research Center, Riyadh, Saudi Arabia. [347]Laboratory Department, Security Forces Hospital, General Directorate of Medical Services, Ministry of Interior, Riyadh, Saudi Arabia. [348]Department of Clinical Laboratory Sciences, College of Applied Medical Sciences, King Saud University, Riyadh, Saudi Arabia. [349]King Abdulaziz City for Science and Technology (KACST), Riyadh, Saudi Arabia. [350]Life Science and Environmental Institute, King Abdulaziz City for Science and Technology (KACST), Riyadh, Saudi Arabia. [351]Department of Developmental Medicine, King Abdullah International Medical Research Center, King Saud Bin Abdulaziz University for Health Sciences, King Abdulaziz Medical City, Ministry of National Guard Health Affairs, Riyadh, Saudi Arabia. [352]Titus Family Department of Clinical Pharmacy, USC School of Pharmacy University of Southern California, Los Angeles, CA, USA. [353]Centre for Fertility and Health, Norwegian Institute of Public Health, Oslo, Norway. [354]Department of Method Development and Analytics, Norwegian Institute of Public Health, Oslo, Norway. [355]Department of Genetics and Bioinformatics, Norwegian Institute of Public Health, Oslo, Norway. [356]Department of Twin Research and Genetic Epidemiology, King's College London, London, UK. [357]NIHR Biomedical Research Centre at Guy's and St Thomas' Foundation Trust, London, UK. [358]Vanda Pharmaceuticals, London, UK. [359]Stroke Pharmacogenomics and Genetics, Biomedical Research Institute Sant Pau, Sant Pau Hospital, Barcelona, Spain. [360]Institute of Biomedicine of Valencia (IBV), National Spanish Research Council (CSIC), València, Spain. [361]Network Center for Biomedical Research on Neurodegenerative Diseases (CIBERNED), València, Spain. [362]Neurology and Genetic Mixed Unit, La Fe Health Research Institute, València, Spain. [363]Institute for Biomedical Research of Barcelona (IIBB), National Spanish Research Council (CSIC), Barcelona, Spain. [364]Department of Neurology, Hospital Universitari MútuaTerrassa, Fundació Docència i Recerca MútuaTerrassa, Terrassa, Spain. [365]Department of Molecular and Cell Biology, Centro Nacional de Biotecnología (CNB-CSIC), Campus Universidad Autónoma de Madrid, Madrid, Spain. [366]Instituto de Física de Cantabria (IFCA-CSIC), Santander, Spain. [367]Institut d'Investigacions Biomèdiques August Pi i Sunyer (IDIBAPS), Barcelona, Spain. [368]Hospital Clínic, Barcelona, Spain. [369]Hospital Clínic, IDIBAPS, School of Medicine, University of Barcelona, Barcelona, Spain. [370]IDIBAPS, Barcelona, Spain. [371]IIBB-CSIC, Barcelona, Spain. [372]Servicio de Salud del Principado de Asturias, Oviedo, Spain. [373]Hospital Mutua de Terrassa, Terrassa, Spain. [374]Hospital Valle Hebrón, Barcelona, Spain. [375]Instituto de Biomedicina y Genética Molecular (IBGM), CSIC-Universidad de Valladolid, Valladolid, Spain. [376]Hospital Clínico Universitario de Valladolid (SACYL), Valladolid, Spain. [377]University Hospital of Albacete, Albacete, Spain. [378]Department of Neurology, Biomedical Research Institute Sant Pau (IIB Sant Pau), Hospital de la Santa Creu i Sant Pau, Barcelona, Spain. [379]Hospital Universitario Ramon y Cajal, IRYCIS, Madrid, Spain. [380]Institute of Biomedicine of Seville (IBiS), Hospital Universitario Virgen del Rocío, CSIC and University of Seville, Seville, Spain. [381]Department of Neurology, Hospital Universitario Virgen Macarena, Seville, Spain. [382]Brigham and Women's Hospital, Boston, MA, USA. [383]Harvard Medical School, Boston, MA, USA. [384]Division of Human Genetics, Department of Pediatrics, The Perelman School of Medicine, University of Pennsylvania, Philadelphia, PA, USA. [385]Faculty of Medicine, University of Iceland, Reykjavik, Iceland. [386]Erasmus MC, Rotterdam, The Netherlands. [387]Finnish Institute for Health and Welfare (THL), Helsinki, Finland. [388]University of Helsinki, Faculty of Medicine, Clinical and Molecular Metabolism Research Program, Helsinki, Finland. [389]Public Health, Faculty of Medicine, University of Helsinki, Helsinki, Finland. [390]Department of Human Genetics, McGill University, Montréal, Québec, Canada. [391]Lady Davis Institute, Jewish General Hospital, McGill University, Montréal, Québec, Canada. [392]Kyoto–McGill International Collaborative School in Genomic Medicine, Graduate School of Medicine, Kyoto University, Kyoto, Japan. [393]Research Fellow, Japan Society for the Promotion of Science, Kyoto, Japan. [394]University of Liege, Liege, Belgium. [395]CHU of Liege, Liege, Belgium. [396]5BHUL (Liege Biobank), CHU of Liege, Liege, Belgium. [397]CHC Mont-Legia, Liege, Belgium. [398]University of Colorado Anschutz Medical Campus, Aurora, CO, USA. [399]Estonian Genome Centre, Institute of Genomics, University of Tartu, Tartu, Estonia. [400]University of Tartu, Tartu, Estonia. [401]Kuressaare Hospital, Kuressaare, Estonia. [402]Tartu University Hospital, Tartu, Estonia. [403]Institute of Biomedicine and Translational Medicine, University of Tartu, Tartu, Estonia. [404]West Tallinn Central Hospital, Tallinn, Estonia. [405]Estonian Health Insurance Fund, Tallinn, Estonia. [406]Infectious Diseases Service, Department of Medicine, University Hospital and University of Lausanne, Lausanne, Switzerland. [407]Institute of Molecular and Clinical Ophthalmology Basel (IOB), Basel, Switzerland. [408]Department of Ophthalmology, University of Basel, Basel, Switzerland. [409]Centre for Primary Care and Public Health, University of Lausanne, Lausanne, Switzerland. [410]Division of Infectious Diseases and Hospital Epidemiology, Cantonal Hospital St Gallen, St Gallen, Switzerland. [411]Division of Intensive Care, Geneva University Hospitals and the University of Geneva Faculty of Medicine, Geneva, Switzerland. [412]Infectious Disease Service, Department of Internal Medicine, Geneva University Hospital, Geneva, Switzerland. [413]Clinique de Médecine et Spécialités, Infectiologie, HFR-Fribourg, Fribourg, Switzerland. [414]Infectious Diseases Division, University Hospital Centre of the canton of Vaud, Hospital of Valais, Sion, Switzerland. [415]GCAT-Genomes for Life, Germans Trias i Pujol Health Sciences Research Institute (IGTP), Badalona, Spain. [416]Catalan Institute of Oncology, Bellvitge Biomedical Research Institute, Consortium for Biomedical Research in Epidemiology and Public Health and University of Barcelona, Barcelona, Spain. [417]Massachusetts General Hospital, Boston, MA, USA. [418]Life and Medical Sciences, Barcelona Supercomputing Center–Centro Nacional de Supercomputación (BSC-CNS), Barcelona, Spain. [419]ISGlobal, Barcelona, Spain. [420]IMIM (Hospital del Mar Medical Research Institute), Barcelona, Spain. [421]Universitat Pompeu Fabra (UPF), Barcelona, Spain. [422]CIBER Epidemiología y Salud Pública (CIBERESP), Madrid, Spain. [423]Medical Genetics, University of Siena, Siena, Italy. [424]Genetica Medica, Azienda Ospedaliero-Universitaria Senese, Siena, Italy. [425]Med Biotech Hub and Competence Center, Department of Medical Biotechnologies, University of Siena, Siena, Italy. [426]Infectious Diseases Clinic, Department of Medicine 2, Azienda Ospedaliera di Perugia and University of Perugia, Santa Maria Hospital, Perugia, Italy. [427]Department of Anesthesia and Intensive Care, University of Modena and Reggio Emilia, Modena, Italy. [428]Division of Infectious Diseases and Immunology, Fondazione IRCCS Policlinico San Matteo, Pavia, Italy. [429]Department of Internal Medicine and Therapeutics, University of Pavia, Pavia, Italy. [430]U.O.C. Medicina, ASST Nord Milano, Ospedale Bassini, Milan, Italy. [431]Department of Mathematics, University of Pavia, Pavia, Italy. [432]University of Siena, DIISM-SAILAB, Siena, Italy. [433]Independent researcher, Milan, Italy. [434]Fondazione IRCCS Ca' Granda Ospedale Maggiore Policlinico, Milan, Italy. [435]Department of Infectious and Tropical Diseases, University of Brescia and ASST Spedali Civili Hospital, Brescia, Italy. [436]Department of Molecular and Translational Medicine, University of Brescia, Brescia, Italy. [437]Clinical Chemistry Laboratory, Cytogenetics and Molecular Genetics Section, Diagnostic Department, ASST Spedali Civili di Brescia, Brescia, Italy. [438]Chirurgia Vascolare, Ospedale Maggiore di Crema, Crema, Italy. [439]III Infectious Diseases Unit, ASST-FBF-Sacco, Milan, Italy. [440]Department of Biomedical and Clinical Sciences Luigi Sacco, University of Milan, Milan, Italy. [441]Department of Specialized and Internal Medicine, Tropical and Infectious Diseases Unit, Azienda Ospedaliera Universitaria Senese, Siena, Italy. [442]HIV/AIDS Department, National Institute for Infectious Diseases Lazzaro Spallanzani, IRCCS, Rome, Italy. [443]Unit of Respiratory Diseases and Lung Transplantation, Department of Internal and Specialist Medicine, University of Siena, Siena, Italy. [444]Unit of Intensive Care Medicine. Departments of Emergency and Urgency, Medicine, Surgery and Neurosciences, Siena University Hospital, Siena, Italy. [445]Unit of Diagnostic Imaging, Departments of Medical, Surgical and Neurosciences and Radiological Sciences, University of Siena, Siena, Italy. [446]Rheumatology Unit, Department of Medicine, Surgery and Neurosciences, University of Siena, Policlinico Le Scotte, Siena, Italy. [447]Infectious Diseases Unit, Department of Specialized and Internal Medicine, San Donato Hospital Arezzo, Arezzo, Italy. [448]Anesthesia Unit, Department of Emergency, San Donato Hospital, Arezzo, Italy. [449]Pneumology Unit and UTIP, Department of

Specialized and Internal Medicine, San Donato Hospital, Arezzo, Italy. [450]Anesthesia Unit, Department of Emergency, Misericordia Hospital, Grosseto, Italy. [451]Infectious Diseases Unit, Department of Specialized and Internal Medicine, Misericordia Hospital, Grosseto, Italy. [452]Department of Preventive Medicine, Azienda USL Toscana Sud Est, Tuscany, Italy. [453]Clinical Chemical Analysis Laboratory, Misericordia Hospital, Grosseto, Italy. [454]Territorial Scientific Technician Department, Azienda USL Toscana Sud Est, Arezzo, Italy. [455]Clinical Chemical Analysis Laboratory, San Donato Hospital, Arezzo, Italy. [456]Department of Health Sciences, Clinic of Infectious Diseases, ASST Santi Paolo e Carlo, University of Milan, Milan, Italy. [457]Department of Medical and Surgical Sciences for Children and Adults, University of Modena and Reggio Emilia, Modena, Italy. [458]Infectious Diseases Clinic, Santa Maria Hospital, University of Perugia, Perugia, Italy. [459]Department of Infectious Diseases, Treviso Hospital, Treviso, Italy. [460]Clinical Infectious Diseases, Mestre Hospital, Venezia, Italy. [461]Infectious Diseases Clinic, Belluno, Italy. [462]Department of Molecular Medicine, University of Padova, Padua, Italy. [463]Medical Genetics and Laboratory of Medical Genetics Unit, A.O.R.N. Antonio Cardarelli Hospital, Naples, Italy. [464]Department of Molecular Medicine and Medical Biotechnology, University of Naples Federico II, Naples, Italy. [465]CEINGE Biotecnologie Avanzate, Naples, Italy. [466]IRCCS SDN, Naples, Italy. [467]Unit of Respiratory Physiopathology, AORN dei Colli, Monaldi Hospital, Naples, Italy. [468]Division of Medical Genetics, Fondazione IRCCS Casa Sollievo della Sofferenza Hospital, San Giovanni Rotondo, Italy. [469]Laboratory of Regulatory and Functional Genomics, Fondazione IRCCS Casa Sollievo della Sofferenza Hospital, San Giovanni Rotondo, Italy. [470]Department of Medical Sciences, Fondazione IRCCS Casa Sollievo della Sofferenza Hospital, San Giovanni Rotondo, Italy. [471]Clinical Trial Office, Fondazione IRCCS Casa Sollievo della Sofferenza Hospital, San Giovanni Rotondo, Italy. [472]Department of Health Sciences, University of Genova, Genova, Italy. [473]Infectious Diseases Clinic, Policlinico San Martino Hospital, IRCCS for Cancer Research, Genova, Italy. [474]Microbiology, Fondazione Policlinico Universitario Agostino Gemelli IRCCS, Catholic University of Medicine, Rome, Italy. [475]Department of Laboratory Sciences and Infectious Diseases, Fondazione Policlinico Universitario A. Gemelli IRCCS, Rome, Italy. [476]Department of Cardiovascular Diseases, University of Siena, Siena, Italy. [477]Otolaryngology Unit, University of Siena, Siena, Italy. [478]Department of Internal Medicine, ASST Valtellina e Alto Lario, Sondrio, Italy. [479]Oncologia Medica e Ufficio Flussi Sondrio, Sondrio, Italy. [480]First Aid Department, Luigi Curto Hospital, Polla, Salerno, Italy. [481]Local Health Unit, Pharmaceutical Department of Grosseto, Toscana Sud Est Local Health Unit, Grosseto, Italy. [482]U.O.C. Laboratorio di Genetica Umana, IRCCS Istituto Giannina Gaslini, Genoa, Italy. [483]Infectious Diseases Clinics, University of Modena and Reggio Emilia, Modena, Italy. [484]Department of Respiratory Diseases, Azienda Ospedaliera di Cremona, Cremona, Italy. [485]Department of Cardiovascular, Neural and Metabolic Sciences, Istituto Auxologico Italiano, IRCCS, San Luca Hospital, Milan, Italy. [486]Department of Medicine and Surgery, University of Milano-Bicocca, Milan, Italy. [487]Laboratory of Cardiovascular Genetics, Istituto Auxologico Italiano, IRCCS, Milan, Italy. [488]Unit of Infectious Diseases, ASST Papa Giovanni XXIII Hospital, Bergamo, Italy. [489]Direzione Scientifica, Istituti Clinici Scientifici Maugeri IRCCS, Pavia, Italy. [490]Department of Cardiology, Istituti Clinici Scientifici Maugeri IRCCS, Institute of Montescano, Pavia, Italy. [491]Department of Cardiac Rehabilitation, Institute of Tradate (VA) and Istituti Clinici Scientifici Maugeri IRCCS, Pavia, Italy. [492]Department of Cardiology, Istituti Clinici Scientifici Maugeri IRCCS, Institute of Milan, Milan, Italy. [493]Department of Electronics, Information and Bioengineering (DEIB), Politecnico di Milano, Milan, Italy. [494]Scuola Normale Superiore, Pisa, Italy. [495]CNR-Consiglio Nazionale delle Ricerche, Istituto di Biologia e Biotecnologia Agraria (IBBA), Milano, Italy. [496]Core Research Laboratory, ISPRO, Florence, Italy. [497]Fondazione per la Ricerca Ospedale di Bergamo, Bergamo, Italy. [498]Health Management, Azienda USL Toscana Sud Est, Tuscany, Italy. [499]IRCCS Mondino Foundation, Pavia, Italy. [500]Medical Genetics Unit, Meyer Children's University Hospital, Florence, Italy. [501]Allelica, New York, NY, USA. [502]Pneumology Unit, Department of Medicine, Misericordia Hospital, Grosseto, Italy. [503]Intensive Care Unit and Department of Anesthesia, ASST Fatebenefratelli Sacco, Luigi Sacco Hospital, Polo Universitario, University of Milan, Milan, Italy. [504]Infectious Disease Unit, Hospital of Massa, Massa, Italy. [505]Department of Clinical Medicine, Public Health, Life and Environment Sciences, University of L'Aquila, L'Aquila, Italy. [506]UOSD Laboratorio di Genetica Medica—ASL Viterbo, San Lorenzo, Italy. [507]Department of Medical Sciences, Infectious and Tropical Diseases Unit, Azienda Ospedaliera Universitaria Senese, Siena, Italy. [508]Unit of Infectious Diseases, Santa Maria Annunziata Hospital, Florence, Italy. [509]Infectious Disease Unit, Hospital of Lucca, Lucca, Italy. [510]Infectious Diseases Unit, Department of Clinical and Experimental Medicine, University of Pisa, Pisa, Italy. [511]Infectious Disease Unit, Santo Stefano Hospital, AUSL Toscana Centro, Prato, Italy. [512]Clinic of Infectious Diseases, Catholic University of the Sacred Heart, Rome, Italy. [513]Department of Diagnostic and Laboratory Medicine, Institute of Biochemistry and Clinical Biochemistry, Fondazione Policlinico Universitario A. Gemelli IRCCS, Catholic University of the Sacred Heart, Rome, Italy. [514]Department of Population Health Sciences, Geisinger Health System, Danville, PA, USA. [515]Department of Molecular and Functional Genomics, Geisinger Health System, Danville, PA, USA. [516]Regeneron Genetics Center, Tarrytown, NY, USA. [517]Phenomic Analytics and Clinical Data Core, Geisinger Health System, Danville, PA, USA. [518]Queen Mary University of London, London, UK. [519]Bradford Institute for Health Research, Bradford Teaching Hospitals National Health Service (NHS) Foundation Trust, Bradford, UK. [520]Medical and Population Genomics, Wellcome Sanger Institute, Hinxton, UK. [521]School of Basic and Medical Biosciences, Faculty of Life Sciences and Medicine, King's College London, London, UK. [522]Department of Human Genetics, Wellcome Sanger Institute, Hinxton, UK. [523]National Institute for Health Research Blood and Transplant Research Unit in Donor Health and Genomics, University of Cambridge, Cambridge, UK. [524]Department of Haematology, University of Cambridge, Cambridge, UK. [525]British Heart Foundation Cardiovascular Epidemiology Unit, Department of Public Health and Primary Care, University of Cambridge, Cambridge, UK. [526]British Heart Foundation Centre of Research Excellence, University of Cambridge, Cambridge, UK. [527]Health Data Research UK Cambridge, Wellcome Genome Campus, University of Cambridge, Cambridge, UK. [528]Department of Genetics, University Medical Centre Groningen, University of Groningen, Groningen, The Netherlands. [529]Department of Genetics, University Medical Centre Utrecht, Utrecht, The Netherlands. [530]Department of Genetics, University Medical Center Groningen, University of Groningen, Groningen, The Netherlands. [531]Department of Genetics, University Medical Centre Groningen, University of Groningen, Groningen, The Netherlands. [532]Department of Psychiatry, University Medical Center Groningen, Groningen, The Netherlands. [533]Department of Biostatistics, Boston University School of Public Health, Boston, MA, USA. [534]Center for Population Genomics, MAVERIC, VA Boston Healthcare System, Boston, MA, USA. [535]MAVERIC, VA Boston Healthcare System, Boston, MA, USA. [536]Department of Epidemiology, Emory University Rollins School of Public Health, Atlanta, GA, USA. [537]Stanford University, Stanford, CA, USA. [538]Vrije Universiteit Amsterdam, Amsterdam, The Netherlands. [539]Broad Institute of MIT and Harvard, Boston, MA, USA. [540]Department of Genetics, University of Pennsylvania Perelman School of Medicine, Philadelphia, PA, USA. [541]Department of Neuroscience, Karolinska Institutet, Stockholm, Sweden. [542]Max Planck Institute for Evolutionary Anthropology, Leipzig, Germany. [543]Anaesthesiology and Intensive Care Medicine, Department of Surgical Sciences, Uppsala University, Uppsala, Sweden. [544]Integrative Physiology, Department of Medical Cell Biology, Uppsala University, Uppsala, Sweden. [545]Stanley Center for Psychiatric Research and Program in Medical and Population Genetics, Muscatine, IA, USA. [546]Division Anesthesiology and Intensive Care, CLINTEC, Karolinska Institutet, Stockholm, Sweden. [547]Department of Clinical Research and Leadership, George Washington University, Washington, DC, USA. [548]Big Data Institute, Nuffield Department of Population Health, University of Oxford, Oxford, UK. [549]Wellcome Centre for Human Genetics, University of Oxford, Oxford, UK. [550]Nuffield Department of Medicine, Experimental Medicine Division, University of Oxford, John Radcliffe Hospital, Oxford, UK. [551]Public Health England, Field Service, Addenbrooke's Hospital, Cambridge, UK. [552]Public Health England, Data and Analytical Services, National Infection Service, London, UK. [553]Genomics PLC, Oxford, UK. [554]Genomics England, London, UK. [555]Ancestry, Lehi, UT, USA. ✉e-mail: dbaraka90@hotmail.com

# Methods

## Ethics

GenOMICC study: GenOMICC was approved by the following research ethics committees: Scotland 'A' Research Ethics Committee (15/SS/0110) and Coventry and Warwickshire Research Ethics Committee (England, Wales and Northern Ireland) (19/WM/0247). Current and previous versions of the study protocol are available at https://genomicc.org/protocol/. 100,000 Genomes project: the 100,000 Genomes project was approved by the East of England—Cambridge Central Research Ethics Committee (REF 20/EE/0035). Only individuals from the 100,000 Genomes project for whom WGS data were available and who consented for their data to be used for research purposes were included in the analyses. UK Biobank study: ethical approval for the UK Biobank was previously obtained from the North West Centre for Research Ethics Committee (11/NW/0382). The work described herein was approved by UK Biobank under application number 26041. Geisinger Health Systems (GHS) study: approval for DiscovEHR analyses was provided by the GHS Institutional Review Board under project number 2006-0258. AncestryDNA study: all data for this research project were from individuals who provided prior informed consent to participate in AncestryDNA's Human Diversity Project, as reviewed and approved by our external institutional review board, Advarra (formerly Quorum). All data were de-identified before use. Penn Medicine Biobank study: appropriate consent was obtained from each participant regarding the storage of biological specimens, genetic sequencing and genotyping, and access to all available EHR data. This study was approved by the institutional review board of the University of Pennsylvania and complied with the principles set out in the Declaration of Helsinki. Informed consent was obtained for all study participants. 23andMe study: participants in this study were recruited from the customer base of 23andMe, a personal genetics company. All individuals included in the analyses provided informed consent and answered surveys online according to the 23andMe protocol for research in humans, which was reviewed and approved by Ethical and Independent Review Services, a private institutional review board (http://www.eandireview.com).

## Recruitment of cases (patients with COVID-19)

Patients were recruited to the GenOMICC study in 224 UK intensive care units (https://genomicc.org). All individuals had confirmed COVID-19 according to local clinical testing and were deemed, in the view of the treating clinician, to require continuous cardiorespiratory monitoring. In UK practice this kind of monitoring is undertaken in high-dependency or intensive care units.

## Recruitment of control individuals

**Mild or asymptomatic control individuals.** Participants were recruited to the mild COVID-19 cohort on the basis of having experienced mild (non-hospitalized) or asymptomatic COVID-19. Participants volunteered to take part in the study via a microsite and were required to self-report the details of a positive COVID-19 test. Volunteers were prioritized for genome sequencing on the basis of demographic matching with the critical COVID-19 cohort considering self-reported ancestry, sex, age and location within the UK. We refer to this cohort as the COVID-19 mild cohort.

**Control individuals from the 100,000 Genomes project.** Participants were enrolled in the 100,000 Genomes Project from families with a broad range of rare diseases, cancers and infection by 13 regional NHS Genomic Medicine Centres across England and in Northern Ireland, Scotland and Wales. For this analysis, participants for whom a positive SARS-CoV-2 test had been recorded as of March 2021 were not included owing to uncertainty in the severity of COVID-19 symptoms.

Only participants for whom genome sequencing was performed from blood-derived DNA were included and participants with haematological malignancies were excluded to avoid potential tumour contamination.

## DNA extraction

For severe cases of COVID-19 and mild cohort controls, DNA was extracted from whole blood either manually using a Nucleon Kit (Cytiva) and resuspended in 1 ml TE buffer pH 7.5 (10 mM Tris-Cl pH 7.5, 1 mM EDTA pH 8.0), or automated on the Chemagic 360 platform using the Chemagic DNA blood kit (PerkinElmer) and re-suspended in 400 μl elution buffer. The yield of the DNA was measured using Qubit and normalized to 50 ng μl$^{-1}$ before sequencing. For the 100,000 Genomes Project samples, DNA was extracted from whole blood at designated extraction centres following sample handling guidance provided by Genomics England and NHS England.

## WGS

Sequencing libraries were generated using the Illumina TruSeq DNA PCR-Free High Throughput Sample Preparation kit and sequenced with 150-bp paired-end reads in a single lane of an Illumina Hiseq X instrument (for 100,000 Genomes Project samples) or a NovaSeq instrument (for the COVID-19 critical and mild cohorts).

**Sequencing data quality control.** All genome sequencing data were required to meet minimum quality metrics and quality control measures were applied for all genomes as part of the bioinformatics pipeline. The minimum data requirements for all genomes were: more than $85 \times 10^{-9}$ bases with $Q \geq 30$ and at least 95% of the autosomal genome covered at 15× or higher calculated from reads with mapping quality greater than 10 after removing duplicate reads and overlapping bases, after adaptor and quality trimming. Assessment of germline cross-sample contamination was performed using VerifyBamID and samples with more than 3% contamination were excluded. Sex checks were performed to confirm that the sex reported for a participant was concordant with the sex inferred from the genomic data.

## WGS alignment and variant calling

**COVID-19 cohorts.** For the critical and mild COVID-19 cohorts, sequencing data alignment and variant calling were performed with Genomics England pipeline 2.0, which uses the DRAGEN software (v.3.2.22). Alignment was performed to genome reference GRCh38 including decoy contigs and alternative haplotypes (ALT contigs), with ALT-aware mapping and variant calling to improve specificity.

**100,000 Genomes Project cohort.** All genomes from the 100,000 Genomes Project cohort were analysed with the Illumina North Star Version 4 Whole Genome Sequencing Workflow (NSV4, v.2.6.53.23); which comprises the iSAAC Aligner (v.03.16.02.19) and Starling Small Variant Caller (v.2.4.7). Samples were aligned to the Homo Sapiens NCBI GRCh38 assembly with decoys.

A subset of the genomes from the cancer program of the 100,000 Genomes Project were reprocessed (alignment and variant calling) using the same pipeline used for the COVID-19 cohorts (DRAGEN v.3.2.22) for equity of alignment and variant calling.

## Aggregation

Aggregation was conducted separately for the samples analysed with Genomics England pipeline 2.0 (severe cohort, mild cohort, cancer-realigned 100,000 Genomes Project) and those analysed with the Illumina North Star Version 4 pipeline (100,000 Genomes Project).

For the first three, the WGS data were aggregated from single-sample gVCF files to multi-sample VCF files using GVCFGenotyper (GG) v.3.8.1, which accepts gVCF files generated by the DRAGEN pipeline as input. GG

outputs multi-allelic variants (several ALT variants per position on the same row), and for downstream analyses the output was decomposed to bi-allelic variants per row using the software vt v.0.57721. We refer to the aggregate as aggCOVID_vX, in which X is the specific freeze. The analysis in this manuscript uses data from freeze v.4.2 and the respective aggregate is referred to as aggCOVID_v4.2.

Aggregation for the 100,000 Genomes Project cohort was performed using Illumina's gvcfgenotyper v.2019.02.26, merged with bcftools v.1.10.2 and normalized with vt v.0.57721.

### Sample quality control

Samples that failed any of the following four BAM-level quality control filters: freemix contamination > 3%, mean autosomal coverage < 25×, per cent mapped reads < 90% or per cent chimeric reads > 5% were excluded from the analysis.

In addition, a set of VCF-level quality control filters were applied after aggregation on all autosomal bi-allelic single-nucleotide variants (SNVs) (akin to gnomAD v.3.1)[18]. Samples were filtered out on the basis of the residuals of eleven quality control metrics (calculated using bcftools) after regressing out the effects of sequencing platform and the first three ancestry assignment principal components (PCs) (including all linear, quadratic and interaction terms) taken from the sample projections onto the SNP loadings from the individuals of 1000 Genomes Project phase 3 (1KGP3). Samples were removed that were four median absolute deviations (MADs) above or below the median for the following metrics: ratio of heterozygous to homozygous, ratio of insertions to deletions, ratio of transitions to transversions, total deletions, total insertions, total heterozygous SNPs, total homozygous SNPs, total transitions and total transversions. For the number of total singletons (SNPs), samples were removed that were more than 8 MADs above the median. For the ratio of heterozygous to homozygous alternative SNPs, samples were removed that were more than 4 MADs above the median.

After quality control, 79,803 individuals were included in the analysis with the breakdown according to cohort shown in Supplementary Table 2.

### Selection of high-quality independent SNPs

We selected high-quality independent variants for inferring kinship coefficients, performing PCA, assigning ancestry and for the conditioning on the genetic relatedness matrix by the logistic mixed model of SAIGE and SAIGE-GENE. To avoid capturing platform and/or analysis pipeline effects for these analyses, we performed very stringent variant quality control as described below.

**High-quality common SNPs.** We started with autosomal, bi-allelic SNPs which had a frequency of higher than 5% in aggV2 (100,000 Genomes Project participant aggregate) and in the 1KGP3. We then restricted to variants that had missingness < 1%, median genotype quality control > 30, median depth (DP) ≥ 30 and at least 90% of heterozygote genotypes passing an ABratio binomial test with $P$ value > $10^{-2}$ for aggV2 participants. We also excluded variants in complex regions from the list available in https://genome.sph.umich.edu/wiki/Regions_of_high_linkage_disequilibrium_(LD) (lifted over for GRCh38), and variants where the REF/ALT combination was CG or AT (C/G, G/C, A/T, T/A). We also removed all SNPs that were out of Hardy–Weinberg equilibrium (HWE) in any of the AFR, EAS, EUR or SAS super-populations of aggV2, with a $P$ value cut-off of $P_{HWE} < 10^{-5}$. We then LD-pruned using PLINK v.1.9 with $r^2 = 0.1$ and in 500-kb windows. This resulted in a total of 63,523 high-quality sites from aggV2.

We then extracted these high-quality sites from the aggCOVID_v4.2 aggregate and further applied variant quality filters (missingness < 1%, median quality control > 30, median depth ≥ 30 and at least 90% of heterozygote genotypes passing an ABratio binomial test with $P$ value > $10^{-2}$), per batch of sequencing platform (that is, HiseqX, NovaSeq6000).

After applying variant filters in aggV2 and aggCOVID_v4.2, we merged the genomic data from the two aggregates for the intersection of the variants, which resulted in a final total of 58,925 sites.

**High-quality rare SNPs.** We selected high-quality rare (MAF < 0.005) bi-allelic SNPs to be used with SAIGE for aggregate variant testing (AVT) analysis. To create this set, we applied the same variant quality control procedure as with the common variants: We selected variants that had missingness < 1%, median quality control > 30, median depth ≥ 30 and at least 90% of heterozygote genotypes passing an ABratio binomial test with $P$ value > $10^{-2}$ per batch of sequencing and genotyping platform (that is, HiSeq + NSV4, HiSeq + Pipeline 2.0, NovaSeq + Pipeline 2.0). We then subsetted those to the following groups of minor allele count (MAC) and MAF categories: MAC 1, 2, 3, 4, 5, 6–10, 11–20, MAC 20–MAF 0.001, MAF 0.001–0.005.

### Relatedness, ancestry and principal components

**Kinship.** We calculated kinship coefficients among all pairs of samples using the software PLINK v.2.0 and its implementation of the KING robust algorithm. We used a kinship cut-off of <0.0442 to select unrelated individuals with argument "–king-cutoff".

**Genetic ancestry prediction.** To infer the ancestry of each individual, we performed principal component analysis (PCA) on unrelated 1KGP3 individuals with GCTA v.1.93.1_beta software using high-quality common SNPs[43], and inferred the first 20 PCs. We calculated loadings for each SNP, which we used to project aggV2 and aggCOVID_v4.2 individuals onto the 1KGP3 PCs. We then trained a random forest algorithm from the R package randomForest with the first 10 1KGP3 PCs as features and the super-population ancestry of each individual as labels. These were 'AFR' for individuals of African ancestry, 'AMR' for individuals of American ancestry, 'EAS' for individuals of East Asian ancestry, 'EUR' for individuals of European ancestry and 'SAS' for individuals of South Asian ancestry. We used 500 trees for the training. We then used the trained model to assign a probability of belonging to a certain super-population class for each individual in our cohorts. We assigned individuals to a super-population when class probability ≥ 0.8. Individuals for whom no class had probability ≥ 0.8 were labelled as 'unassigned' and were not included in the analyses.

**PCA.** After labelling each individual with predicted genetic ancestry, we calculated ancestry-specific PCs using GCTA v.1.93.1_beta[43]. We computed 20 PCs for each of the ancestries that were used in the association analyses (AFR, EAS, EUR and SAS).

### Variant quality control

Variant quality control was performed to ensure high quality of variants and to minimize batch effects due to using samples from different sequencing platforms (NovaSeq6000 and HiseqX) and different variant callers (Strelka2 and DRAGEN). We first masked low-quality genotypes setting them to missing, merged aggregate files and then performed additional variant quality control separately for the two major types of association analyses, GWAS and AVT, which concerned common and rare variants, respectively.

**Masking.** Before any analysis, we masked low-quality genotypes using the bcftools setGT module. Genotypes with DP < 10, genotype quality (GQ) < 20 and heterozygote genotypes failing an ABratio binomial test with $P$ value < $10^{-3}$ were set to missing.

We then converted the masked VCF files to PLINK and bgen format using PLINK v.2.0.

**Merging of aggregate samples.** Merging of aggV2 and aggCOVID_v4.2 samples was done using PLINK files with masked genotypes and the merge function of PLINK v.1.9[44]. for variants that were found in both aggregates.

## GWAS analyses

**Variant quality control.** We restricted all GWAS analyses to common variants applying the following filters using PLINK v.1.9: MAF > 0 in both cases and controls, MAF > 0.5% and MAC > 20, missingness < 2%, differential missingness between cases and controls, mid-$P$ value < $10^{-5}$, HWE deviations on unrelated controls, mid-$P$ value < $10^{-6}$. Multi-allelic variants were in addition required to have MAF > 0.1% in both aggV2 and aggCOVID_v4.2.

**Control–control quality control filter.** 100,000 Genomes Project aggV2 samples that were aligned and genotype called with the Illumina North Star version 4 pipeline represented the majority of control samples in our GWAS analyses, whereas all of the cases were aligned and called with Genomics England pipeline 2.0 (Supplementary Table 1). Therefore, the alignment and genotyping pipelines partially match the case–control status, which necessitates additional filtering for adjusting for between-pipeline differences in alignment and variant calling. To control for potential batch effects, we used the overlap of 3,954 samples from the Genomics England 100,000 Genomes Project participants that were aligned and called with both pipelines. For each variant, we computed and compared between platforms the inferred allele frequency for the population samples. We then filtered out all variants that had >1% relative difference in allele frequency between platforms. The relative difference was computed on a per-population basis for EUR ($n$ = 3,157), SAS ($n$ = 373), AFR ($n$ = 354) and EAS ($n$ = 81).

**Model.** We used a two-step logistic mixed model regression approach as implemented in SAIGE v.0.44.5 for single-variant association analyses. In step 1, SAIGE fits the null mixed model and covariates. In step 2, single-variant association tests are performed with the saddlepoint approximation (SPA) correction to calibrate unbalanced case–control ratios. We used the high-quality common variant sites for fitting the null model and sex, age, age$^2$, age-by-sex and 20 PCs as covariates in step 1. The PCs were computed separately by predicted genetic ancestry (that is, EUR-specific, AFR-specific and so on), to capture subtle structure effects.

**Analyses.** All analyses were done on unrelated individuals with a pairwise kinship coefficient < 0.0442. We conducted GWAS analyses per predicted genetic ancestry, for all populations for which we had more than 100 cases and more than 100 controls (AFR, EAS, EUR and SAS).

**Multiple testing correction.** As our study is testing variants that were directly sequenced by WGS and not imputed, we calculated the $P$ value significance threshold by estimating the effective number of tests. After selecting the final filtered set of tested variants for each population, we LD-pruned in a window of 250 kb and $r^2$ = 0.8 with PLINK 1.9. We then computed the Bonferroni-corrected $P$ value threshold as 0.05 divided by the number of LD-pruned variants tested in the GWAS. The $P$ value thresholds that were used for declaring statistical significance are provided in Supplementary Table 5.

**LD-clumping.** We used PLINK v.1.9 to do clumping of variants that were genome-wide significant for each analysis with $P1$ set to per-population $P$ value from Supplementary Table 5, $P2$ = 0.01, clump distance 1,500 kb and $r^2$ = 0.1.

**Conditional analysis and signal independence.** To find the set of independent variants in the per-population analyses, we performed a step-wise conditional analysis with the GWAS summary statistics for each population using GCTA 1.9.3 –cojo-slct function[43]. The parameters for the function were pval = 2.2 × $10^{-8}$, a distance of 10,000 kb and a colinear threshold of 0.9 (ref. [45]). For establishing independence of multi-ancestry meta-analysis signals from per-population discovered signals, we performed LD-clumping using the meta-analysis summaries

and identified signals with no overlap with the LD-clumped results from the per-population analyses. In addition to the GCTA-cojo analysis, we also performed confirmatory individual-level conditional analysis as implemented in SAIGE. For every lead variant signal (including the multi-ancestry meta-analysis signals), we conditioned on the lead variants of all other signals identified as independent by GCTA-cojo and located on the same chromosome with option –condition of SAIGE (Supplementary Table 6).

**Fine-mapping.** We performed fine-mapping for genome-wide-significant signals using the R package SusieR v.0.11.42[13]. For each genome-wide-significant variant locus, we selected the variants 1.5 Mbp on each side and computed the correlation matrix among them with PLINK v.1.9. We then ran the susieR summary-statistics-based function susie_rss and provided the summary $z$ scores from SAIGE (that is, effect size divided by its standard error) and the correlation matrix computed with the same samples that were used for the corresponding GWAS. We required coverage ≥ `` {=html}0.95 for each identified credible set and minimum and median absolute correlation coefficients (purity) of $r$ = 0.1 and 0.5, respectively.

**Functional annotation of credible sets.** We annotated all variants included in each credible set identified by SusieR using the online Variant Effect Predictor (VEP) v.104 and selected the worst consequence across GENCODE basic transcripts (Supplementary Information). We also ranked each variant within each credible set according to the predicted consequence and the ranking was based on the table provided by Ensembl: https://www.ensembl.org/info/genome/variation/prediction/predicted_data.html.

**Multi-ancestry meta-analysis.** We performed a meta-analysis across all ancestries using an inverse-variance weighting method and control for population stratification for each separate analysis in the METAL software[46]. The meta-analysed variants were filtered for variants with heterogeneity $P$ value $P$ < 2.22 × $10^{-8}$ and variants that are not present in at least half of the individuals. We used the meta R package to plot forest plots of the clumped multi-ancestry meta-analysis variants[47].

**LD-based validation of lead GWAS signals.** To quantify the support for genome-wide-significant signals from nearby variants in LD, we assessed the internal consistency of GWAS results of the lead variants and their surroundings. To this end, we compared observed $z$-scores at lead variants with the expected $z$-scores based on those observed at neighbouring variants. Specifically, we computed the observed $z$-score for a variant $i$ as $s_i = \hat{\beta}/\hat{\sigma}_{\hat{\beta}}$ and, following a previous approach[48], the imputed $z$-score at a target variant $t$ as

$$\hat{s}_t = \mathbf{\Sigma}_{t,P}(\mathbf{\Sigma}_{P,P} + \lambda\mathbf{I})^{-1}\mathbf{s}_P$$

where $\mathbf{s}_P$ are the observed z-scores at a set $P$ of predictor variants, $\mathbf{\Sigma}_{x,y}$ is the empirical correlation matrix of dosage coded genotypes computed on the GWAS sample between the variants in $x$ and $y$, and $\lambda$ is a regularization parameter set to $10^{-5}$. The set $P$ of predictor variants consisted of all variants within 100 kb of the target variant with a genotype correlation with the target variant greater than 0.25. This approach is similar to one proposed recently[49].

**Stratified analysis.** We performed sex-specific analysis (male and female individuals separately) as well as analysis stratified by age (that is, participants of younger than 60 years old and 60 years old or above) for the EUR ancestry group. To compare the effect of variants within groups for the age- and sex-stratified analysis we first adjusted the effect and error of each variant for the standard deviation of the trait in each stratified group and then used the following $t$-statistic, as in previous studies[50,51]

$$t = \frac{b_1 - b_2}{\sqrt{se_1^2 + se_2^2 - 2 \times rse_1 \times rse_2}}$$

where $b_1$ is the adjusted effect for group 1, $b_2$ is the adjusted effect for group 2, $se_1$ and $se_2$ are the adjusted standard errors for groups 1 and 2, respectively, and $r$ is the Spearman rank correlation between groups across all genetic variants.

**Replication.** To generate a replication set, we conducted a meta-analysis of data from 23andMe, together with a meta-analysis of the COVID-19 HGI data freeze 6 (hospitalized COVID versus population) GWAS (B2 analysis), including all genetic ancestries. Although the HGI programme included an analysis designed to mirror the GenOMICC study (analysis 'A2'), most of these cases come from GenOMICC and are already included in the discovery cohort. We therefore used the broader hospitalized phenotype ('B2') for replication.

To account for signal due to sample overlap we performed a mathematical subtraction from HGI v.6 B2, of the GenOMICC GWAS of European genetic ancestry. Publicly available HGI data were downloaded from https://www.covid19hg.org/results/r6/. The subtraction was performed using the MetaSubtract package (v.1.60) for R (v.4.0.2) after removing variants with the same genomic position and using the lambda.cohorts with genomic inflation calculated on the GenOMICC summary statistics.

We calculated a multi-ancestry meta-analysis for the three ancestries with summary statistics in 23andMe—African, Latino and European—using variants that passed the 23andMe ancestry quality control, with imputation score > 0.6 and with MAF > 0.005, before performing a final meta-analysis of 23andMe and HGI B2 without GenOMICC to create the final replication set. Meta-analysis was performed using METAL[46], with the inverse-variance weighting method (STDERR mode) and genomic control ON. We considered that a hit was replicated if the direction of effect in the GenOMICC-subtracted HGI summary statistics was the same as in our GWAS, and the $P$ value was significant after Bonferroni correction for the number of attempted replications (pval < 0.05/25). If the main hit was not present in the HGI–23andMe meta-analysis or if the hit was not replicating, we looked for replication in variants in high LD with the top variant ($r^2 > 0.9$), which helped replicate two regions.

To attempt additional replication of two associations, we performed a multi-ancestry meta-analysis across five continental ancestry groups in the UK Biobank, AncestryDNA, Penn Medicine Biobank and GHS, totalling 9,937 hospitalized cases of COVID-19 and 1,059,390 controls (COVID-19 negative or unknown). Hospitalization status (positive, negative or unknown) was determined on the basis of COVID-19-related ICD10 codes U071, U072, U073 in variable 'diag_icd10' (table 'hesin_diag') in the UK Biobank study; self-reported hospitalization due to COVID-19 in the AncestryDNA study; and medical records in the GHS and Penn Medicine Biobank studies. Association analyses in each study were performed using the genome-wide Firth logistic regression test implemented in REGENIE. In this implementation, Firth's approach is applied when the $P$ value from a standard logistic regression score test is less than 0.05. We included in step 1 of REGENIE (that is, prediction of individual trait values based on the genetic data) directly genotyped variants with MAF > 1%, missingness < 10%, HWE test $P > 1 \times 10^{-15}$ and LD-pruning (1,000 variant windows, 100 variant sliding windows and $r^2 < 0.9$). The association model used in step 2 of REGENIE included as covariates age, age$^2$, sex, age-by-sex, and the first 10 ancestry-informative PCs derived from the analysis of a stricter set of LD-pruned (50 variant windows, 5 variant sliding windows and $r^2 < 0.5$) common variants from the array (imputed for the GHS study) data. Within each study, association analyses were performed separately for five different continental ancestries defined on the basis of the array data: African (AFR), Hispanic or Latin American (HLA),

East Asian (EAS), European (EUR) and South Asian (SAS). Results were subsequently meta-analysed across studies and ancestries using an inverse-variance-weighted fixed-effects meta-analysis.

### HLA imputation and association analysis

HLA types were imputed at two-field (four-digit) resolution for all samples within aggV2 and aggCOVID_v4.2 for the following seven loci: HLA-A, HLA-C, HLA-B, HLA-DRB1, HLA-DQA1, HLA-DQB1 and HLA-DPB1, using the HIBAG package in R[15]. At the time of writing, HLA types were also imputed for ~82% of samples using HLA*LA[52]. Inferred HLA alleles between HIBAG and HLA*LA were more than 96% identical at four-digit resolution. HLA association analysis was run under an additive model using SAIGE, in an identical manner to the SNV GWAS. The multi-sample VCF of aggregated HLA type calls from HIBAG was used as input in cases in which any allele call with posterior probability ($T$) < 0.5 were set to missing.

### AVT

AVT on aggCOVID_v4.2 was performed using SKAT-O as implemented in SAIGE-GENE v.0.44.5[17] on all protein-coding genes. Variant and sample quality control for the preparation and masking of the aggregate files have been described elsewhere. We further excluded SNPs with differential missingness between cases and controls (mid-$P$ value < $10^{-5}$) or a site-wide missingness above 5%. Only bi-allelic SNPs with MAF < 0.5% were included.

We filtered the variants to include in the AVT by applying two functional annotation filters: a putative loss of function (pLoF) filter, in which only variants that are annotated by LOFTEE[18] as high-confidence loss of function were included; and a more lenient (missense) filter, in which variants that have a consequence of missense or worse as annotated by VEP, with a CADD_PHRED score of ≥10, were also included. All variants were annotated using VEP v99. SAIGE-GENE was run with the same covariates used in the single variant analysis: sex, age, age$^2$, age-by-sex and 20 (population-specific) PCs generated from common variants (MAF ≥ 5%).

We ran the tests separately by genetically predicted ancestry, as well as across all four ancestries as a mega-analysis. We considered a gene-wide-significant threshold on the basis of the genes tested per ancestry, correcting for the two masks (pLoF and missense; Supplementary Table 14).

### Post-GWAS analysis

**TWASs.** We performed TWASs in the MetaXcan framework and the GTEx v.8 eQTL and splicing quantitative trait loci (sQTL) MASHR-M models available for download in http://predictdb.org/. We first calculated, using the European summary statistics, individual TWASs for whole blood and lung with the S-PrediXcan function[53,54]. Then we performed a metaTWAS including data from all tissues to increase statistical power using s-MultiXcan[55]. We applied the Bonferroni correction to the results to choose significant genes and introns for each analysis.

**Colocalization analysis.** Significant genes from the TWAS, splicing TWAS, metaTWAS and splicing metaTWAS, as well as genes for which one of the top variants was a significant eQTL or sQTL, were selected for a colocalization analysis using the coloc R package[56]. We chose the lead SNPs from the European ancestry GWAS summary statistics and a region of ±200 kb around each SNP to do the colocalization with the identified genes in the region. GTEx v.8 whole-blood and lung tissue summary statistics and eqtlGen (which has blood eQTL summary statistics for more than 30,000 individuals) were used for the analysis[22,57]. We first performed a sensitivity analysis of the posterior probability of colocalization (PP$_{H4}$) on the prior probability of colocalization ($P_{12}$), going from $P_{12} = 10^{-8}$ to $P_{12} = 10^{-4}$, with the default threshold being $P_{12} = 10^{-5}$. eQTL signal and GWAS signals were deemed to colocalize if these two criteria were met: (1) at $P_{12} = 5 \times 10^{-5}$ the probability of colocalization

$PP_{H4} > 0.5$; and (2) at $P_{12} = 10^{-5}$ the probability of independent signal ($PP_{H3}$) was not the main hypothesis ($PP_{H3} < 0.5$). These criteria were chosen to allow eQTLs with weaker $P$ values, owing to lack of power in GTEx v.8, to be colocalized with the signal when the main hypothesis using small priors was that there was not any signal in the eQTL data.

As the chromosome 3-associated interval is larger than 200 kb, we performed additional colocalization including a region up to 500 kb, but no further colocalizations were found.

**Mendelian randomization.** We performed GSMR[23] in a replicated outcome study design. As exposures, we used the pQTLs from the IN-TERVAL study[24]. We used the 1000 Genomes Project imputed data of the Health and Retirement Study (HRS) ($n$ = 8,557) as the LD reference data required for GSMR analysis. The HRS data are available from dbGap (accession number: phs000428).

GSMR was undertaken using all exposures for which we were able to identify two or more independent SNPs associated with the exposure ($P$ value(exposure) $< 5 \times 10^{-8}$; LD clumping ±1 Mb, $r^2 < 0.05$; HEIDI-outlier filtering test, for the removal of SNPs with evidence of horizontal pleiotropy, was performed at the default threshold value of 0.01). Using GSMR, we identified those proteins implicated in determining COVID-19 severity in the new GenOMICC results (following genomic-control correction for inflation) at a false discovery rate (FDR) of less than 0.05, and attempted replication in the GWAS of 'Hospitalized COVID versus population' (phenotype B2) of the COVID-19 HGI (ref. [58]) having excluded the previous GenOMICC results. We achieved this by mathematically removing the contribution of GenOMICC[1] from the meta-analysis. We considered as replicated those results that passed a Bonferroni-corrected $P$ value threshold, correcting for the total number of replication tests attempted (that is, the number of observations from the discovery set with FDR < 0.05).

**Heritability.** For the SNP-based narrow-sense heritabilities of severe COVID-19 and HGI COVID phenotypes, both high-definition likelihood (HDL) and LD score regression (LDSC)[59] methods were applied. The HGI summary statistics were based on the GWAS analysis of all available samples, in which the majority were European populations (see https://www.covid19hg.org/results/r6/). The munge_sumstats.py procedure in the LDSC software was used to harmonize the summary statistics, and in LDSC, the reference panel was built using the 1000 Genome European samples with SNPs that have MAF > 0.05. As both HDL and LDSC are based on GWAS summary $z$-score statistics, the estimated heritabilities are thus on the observed scale.

**Enrichment analysis.** Enrichment analysis was performed to identify ontologies in which discovery genes were overrepresented. Using the XGR algorithm (http://galahad.well.ox.ac.uk/XGR)[60], 19 genes identified through lead variant proximity, credible variant sets, mutation consequence and TWAS analyses were tested for enrichment in disease ontology[61], gene ontologies (biological process, molecular function and cellular component)[62] and KEGG[63] and Reactome[64] pathways using default settings. This generated a $P$ value and FDR for overrepresentation of genes within each of the ontologies (Supplementary Table 15).

### Reporting summary

Further information on research design is available in the Nature Research Reporting Summary linked to this paper.

### Data availability

All data are available through https://genomicc.org/data. This includes downloadable summary data tables and instructions for applying to access individual-level data. Individual-level genome sequence data for the COVID-19 severe and mild cohorts can be analysed by qualified researchers in the UK Outbreak Data Analysis Platform at the University of Edinburgh by application at https://genomicc.org/data. Genomic data for the 100,000 Genomes Project participants and a subset of COVID-19 cases are also available through the Genomics England research environment, which can be accessed by application at https://www.genomicsengland.co.uk/join-a-gecip-domain. The full GWAS summary statistics for the 23andMe discovery dataset are available through 23andMe to qualified researchers under an agreement with 23andMe that protects the privacy of the 23andMe participants. More information and access to the data are provided at https://research.23andMe.com/dataset-access/.

### Code availability

Code to calculate the imputation of $P$ values based on LD SNPs is available at https://github.com/baillielab/GenOMICC_GWAS.

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

**Acknowledgements** We thank the patients and their loved ones who volunteered to contribute to this study at one of the most difficult times in their lives, and the research staff in every intensive care unit who recruited patients at personal risk under challenging conditions. GenOMICC was funded by the Department of Health and Social Care (DHSC), Illumina, LifeArc, the Medical Research Council (MRC), UKRI, Sepsis Research (the Fiona Elizabeth Agnew Trust), the Intensive Care Society, a Wellcome Trust Senior Research Fellowship (J.K.B., 223164/Z/21/Z) a BBSRC Institute Program Support Grant to the Roslin Institute (BBS/E/D/20002172, BBS/E/D/10002070 and BBS/E/D/30002275) and UKRI grants MC_PC_20004, MC_PC_19025, MC_PC_1905 and MRNO2995X/1. WGS was performed by Illumina

at Illumina Laboratory Services and was overseen by Genomics England. We would like to thank all at Genomics England who have contributed to the sequencing, clinical and genomic data analysis. This research is supported in part by the Data and Connectivity National Core Study, led by Health Data Research UK in partnership with the Office for National Statistics and funded by UK Research and Innovation (grant ref. MC_PC_20029). A.D.B. would like to acknowledge funding from the Wellcome PhD training fellowship for clinicians (204979/Z/16/Z) and the Edinburgh Clinical Academic Track (ECAT) programme. We thank the research participants and employees of 23andMe for making this work possible. Genomics England and the 100,000 Genomes Project were funded by the National Institute for Health Research, the Wellcome Trust, the MRC, Cancer Research UK, the DHSC and NHS England. We are grateful for the support from S. Hill and the team in NHS England and the 13 Genomic Medicine Centres that delivered the 100,000 Genomes Project, which provided most of the control genome sequences for this study. We thank the participants in the 100,000 Genomes Project, who made this study possible, and the Genomics England Participant Panel for their strategic advice, involvement and engagement. We acknowledge NHS Digital, Public Health England and the Intensive Care National Audit and Research Centre, who provided life-course longitudinal clinical data on the participants. This work forms part of the portfolio of research of the National Institute for Health Research Barts Biomedical Research Centre. Mark Caulfield is an NIHR Senior Investigator. This study owes a great deal to the National Institute for Healthcare Research Clinical Research Network (NIHR CRN) and the Chief Scientist's Office (Scotland), who facilitate recruitment into research studies in NHS hospitals, and to the global ISARIC and InFACT consortia. Additional replication was conducted using the UK Biobank Resource (project 26041). The Penn Medicine BioBank is funded by a gift from the Smilow family; the National Center for Advancing Translational Sciences of the National Institutes of Health under CTSA award number UL1TR001878; and the Perelman School of Medicine at the University of Pennsylvania. We thank the AncestryDNA customers who voluntarily contributed information in the COVID-19 survey. HRS (dbGaP accession: phs000428.v1.p1): HRS was supported by the National Institute on Aging (NIA U01AG009740). The genotyping was funded separately by the National Institute on Aging (RC2 AG036495, RC4 AG039029). Genotyping was conducted by the NIH Center for Inherited Disease Research (CIDR) at Johns Hopkins University. Genotyping quality control and final preparation of the data were performed by the Genetics Coordinating Center at the University of Washington. The Genotype-Tissue Expression (GTEx) Project was supported by the Common Fund of the Office of the Director of the National Institutes of Health, and by the NCI, NHGRI, NHLBI, NIDA, NIMH and NINDS. The data used for the analyses described in this manuscript were obtained from the GTEx Portal on 22 August 2021 (GTEx Analysis Release v.8 (dbGaP Accession phs000424.v8.p2). We thank the research participants and employees of 23andMe for making this work possible. A full list of contributors who have provided data that were collated in the HGI project, including previous iterations, is available at https://www.covid19hg.org/acknowledgements. The views expressed are those of the authors and not necessarily those of the DHSC, NHS, Department for International Development (DID), NIHR, MRC, Wellcome Trust or Public Health England.

**Author contributions** A.K., E.P.-C., K. Rawlik, A. Stuckey, C.A.O., S.W., T. Malinauskas, Y.W., X.S., K.S.E., B.W., D.R., L.K., M.Z., N.P., J.A.K., J.E.H., A.B., G.R.A., M.A.R.F., A.J., T. Mirshahi, M.O., D.J.R., M.D.R., A.V., J.Y., A.D.B., S.C.H., L. Moutsianas, A.L. and J.K.B. contributed to data analysis. A.K., E.P.-C., K. Rawlik, A. Stuckey, C.A.O., S.W., C.D.R., J.M., A.R., S.C.H., L. Moutsianas and A.L. contributed to bioinformatics. A.K., E.P.-C., K. Rawlik, C.D.R., J.M., D.M., A.N., M.G.S., S.C.H., L. Moutsianas, M.J.C. and J.K.B. contributed to writing and reviewing the manuscript. E.P.-C., K. Rawlik, K.M., S.K., A.F., L. Murphy, K. Rowan, C.P.P., V.V., J.F.W., S.C.H., A.L., M.J.C. and J.K.B. contributed to design. S.W., F.G., W.O., P.G. and S.D. contributed to project management. F.G., W.O., K.M., S.K., P.G., S.D., D.M., A.N., M.G.S., S.S., J.K., T.A.F., M.S.-H., C.S., C.H., P.H., L.L., D. McAuley, H.M., P.J.O., P.E., T.W., A.T., A.F., L. Murphy, K. Rowan, C.P.P., R.H.S., S.C.H. and A.L. contributed to oversight. F.G., W.O., F.M.-C. and J.K.B. contributed to ethics and governance. K.M., A. Siddiq, A.F. and L. Murphy contributed to sample handling and sequencing. A. Siddiq contributed to data collection. T.Z. contributed to sample handing. T.Z., G.E., C.P., D.B. and C.K. contributed to sequencing. L.T. contributed to the recruitment of controls. G.C., P.A., K. Rowan and A.L. contributed to clinical data management. K. Rowan, C.P.P., S.C.H. and J.K.B. contributed to conception. K. Rowan, C.P.P., V.V. and J.F.W. contributed to reviewing the manuscript. M.J.C. and J.K.B. contributed to scientific leadership.

**Competing interests** J.A.K., J.E.H., A.B., G.R.A. and M.A.R.F. are current employees and/or stockholders of Regeneron Genetics Center or Regeneron Pharmaceuticals. Genomics England is a wholly owned Department of Health and Social Care company created in 2013 to work with the NHS to introduce advanced genomic technologies and analytics into healthcare. All Genomics England affiliated authors are, or were, salaried by Genomics England during this programme. All other authors declare that they have no competing interests relating to this work.

**Additional information**
**Correspondence and requests for materials** should be addressed to Mark J. Caulfield, J. Kenneth Baillie or Duna Barakeh.

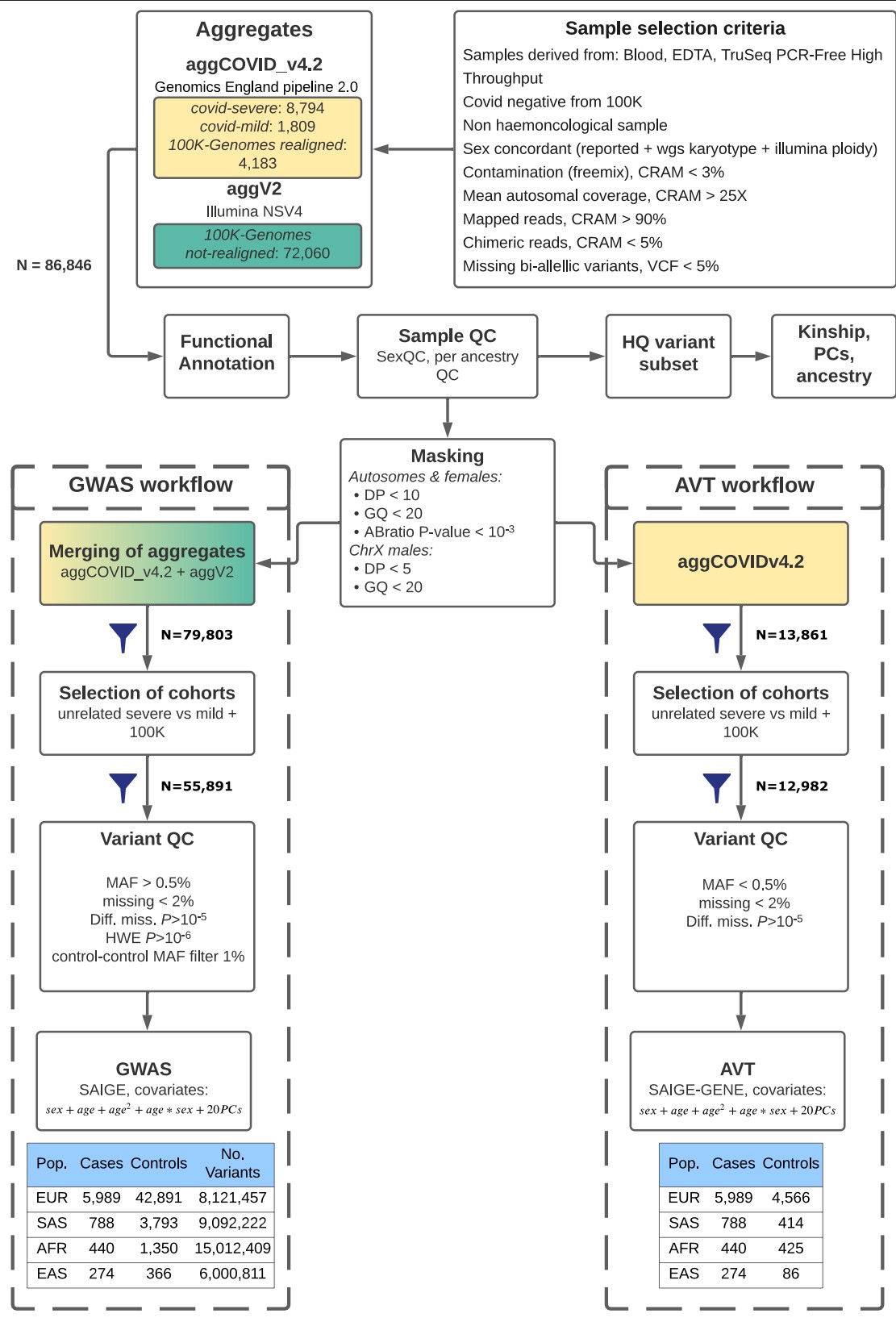

**Extended Data Fig. 1 | Analysis workflow for GWAS and AVT analyses of this study.** The cohorts displayed in yellow and green in the top box were processed with Genomics England Pipeline 2.0 and Illumina NSV4, respectively (see Methods on WGS Alignment and variant calling for details on differences between pipelines). We used individuals that were processed with either pipeline for the GWAS analyses and individuals processed only with Genomics England Pipeline 2.0 for the AVT analyses. The definition of the cases and controls was the same for GWAS and AVT, cases were the COVID-19 severe individuals for both, and controls included individuals from the 100,000 Genomes Project (100,000 Genomes Project) and also COVID-19 positive individuals that were recruited for this study and experienced only mild symptoms (COVID-mild).

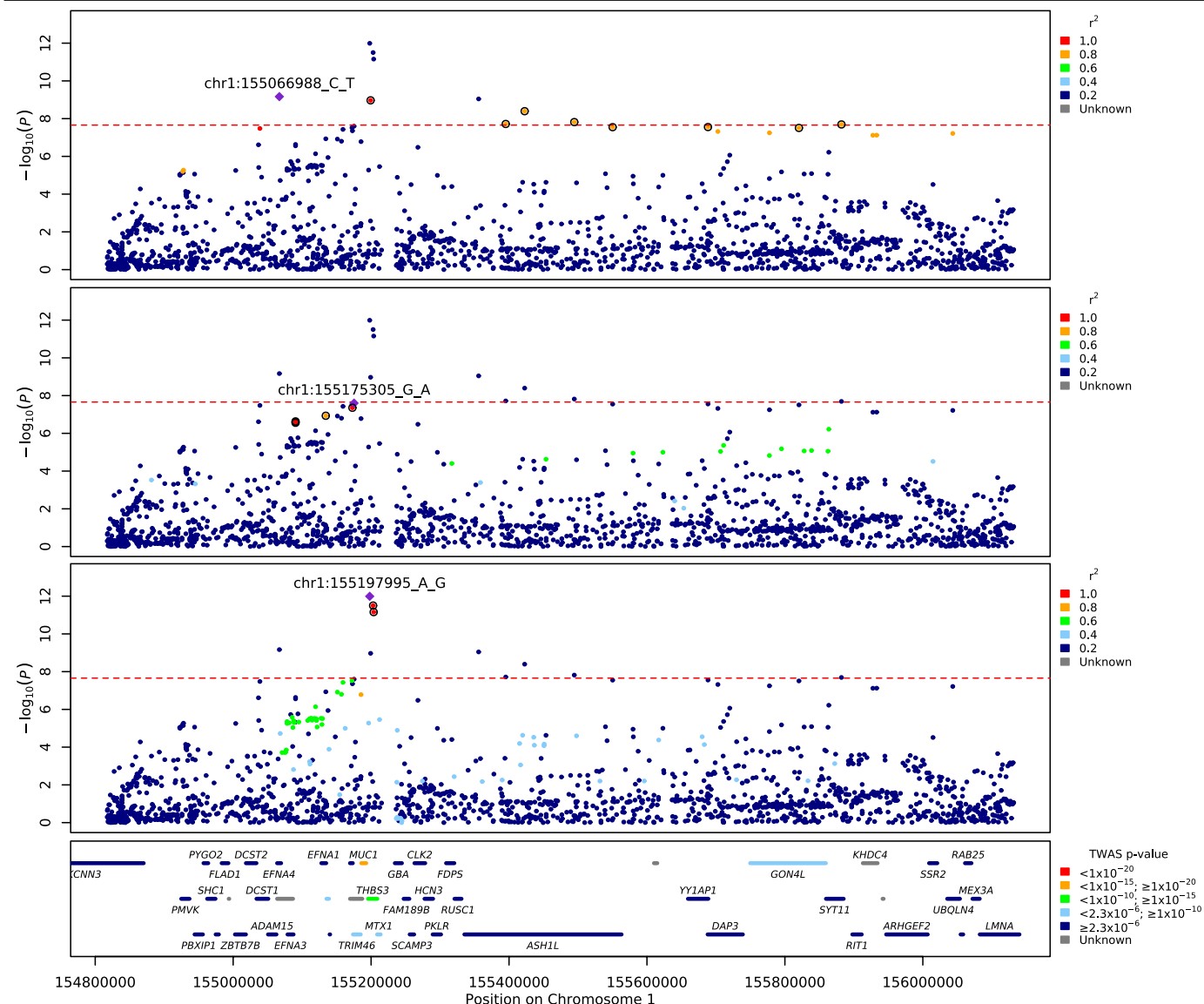

**Extended Data Fig. 2 | Regional detail showing fine-mapping to identify three adjacent independent signals on chromosome 1.** Top two panels: variants in LD with the lead variants shown. The variants that are included in two independent credible sets are displayed with black outline circles. $r^2$ values in the legend denote upper limits, 0.2=[0,0.2], 0.4=[0.2,0.4], 0.6=[0.4,0.6], 0.8=[0.6,0.8],1=[0.8,1]. Bottom panel: locations of protein-coding genes, coloured by TWAS $P$-value. The red dashed line shows the Bonferroni-corrected $P$-value=$2.2 \times 10^{-8}$ for Europeans.

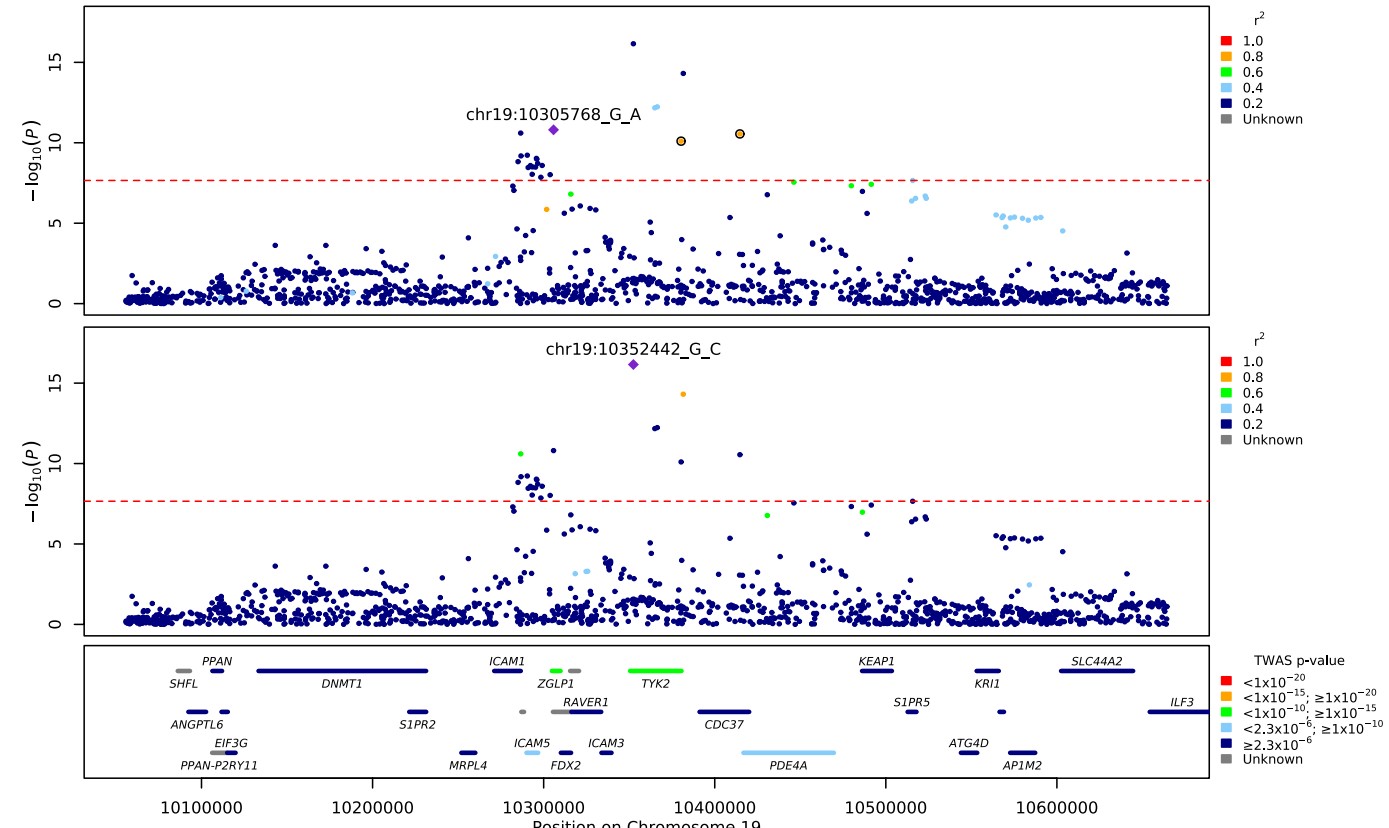

**Extended Data Fig. 3 | Regional detail showing fine-mapping to identify two adjacent independent signals on chromosome 19.** Top two panels: variants in LD with the lead variants shown. The variants that are included in two independent credible sets are displayed with black outline circles. $r^2$ values in the legend denote upper limits, 0.2=[0,0.2], 0.4=[0.2,0.4], 0.6=[0.4,0.6], 0.8=[0.6,0.8],1=[0.8,1]. Bottom panel: locations of protein-coding genes, coloured by TWAS *P*-value. The red dashed line shows the Bonferroni-corrected *P*-value=2.2×10$^{-8}$ for Europeans.

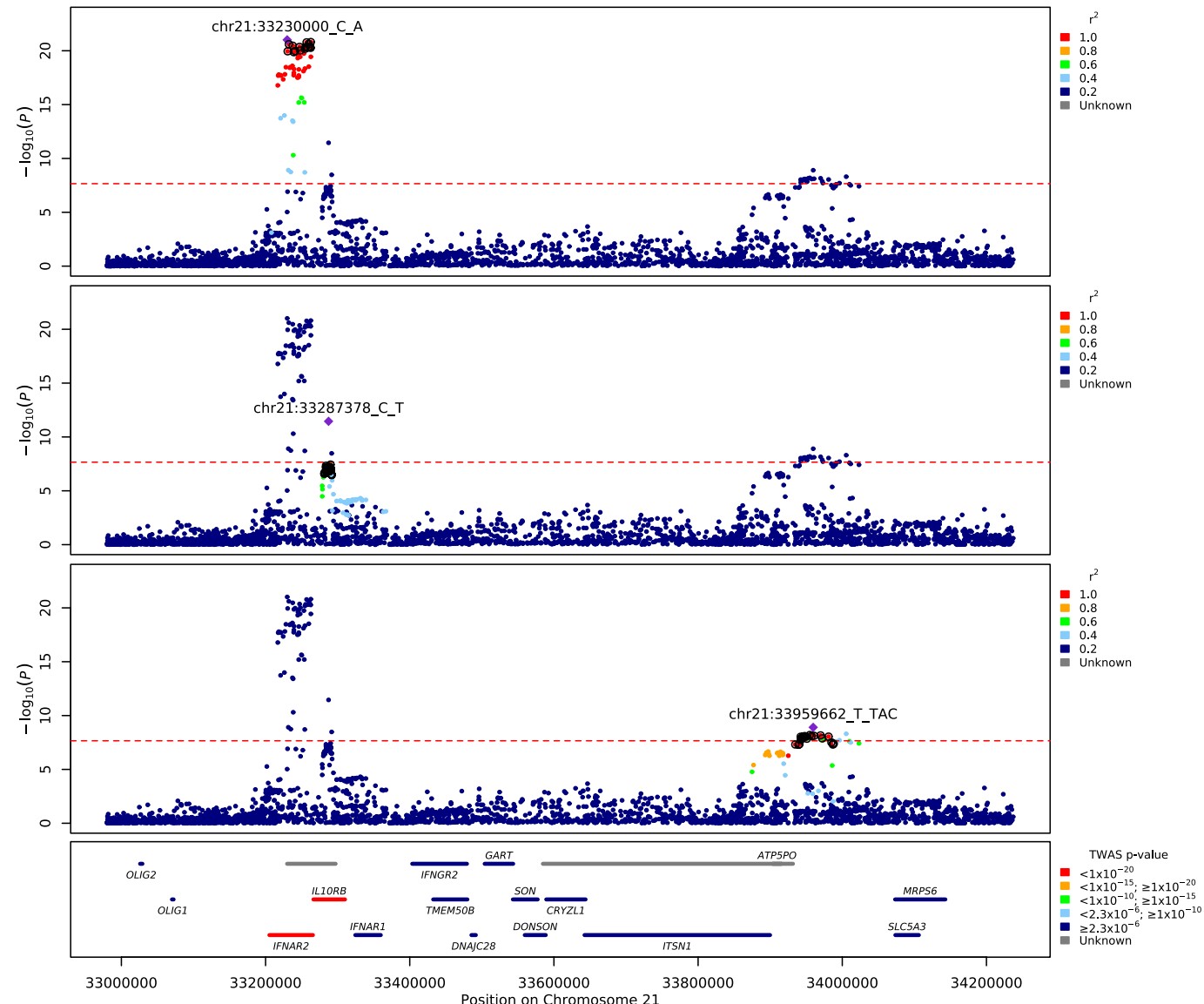

**Extended Data Fig. 4 | Regional detail showing fine-mapping to identify three adjacent independent signals on chromosome 21.** Top three panels: variants in LD with the lead variants shown. The variants that are included in three independent credible sets are displayed with black outline circles.

$r^2$ values in the legend denote upper limits, 0.2=[0,0.2], 0.4=[0.2,0.4], 0.6=[0.4,0.6], 0.8=[0.6,0.8],1=[0.8,1]. Bottom panel: locations of protein-coding genes, coloured by TWAS *P*-value. The red dashed line shows the Bonferroni-corrected *P*-value=2.2 × 10$^{-8}$ for Europeans.

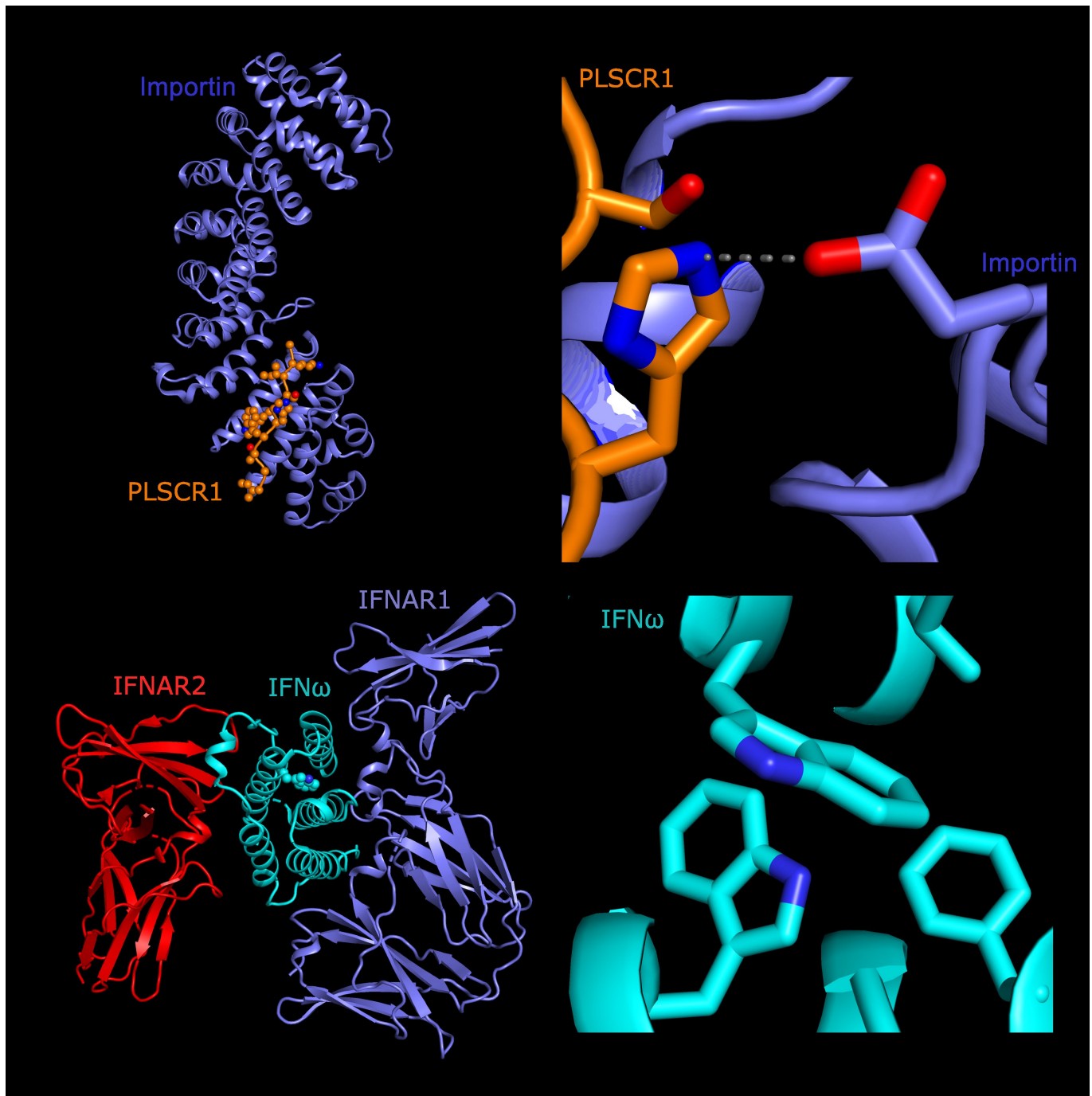

**Extended Data Fig. 5 | Predicted structural consequences of lead variants at PLSCR1 and IFNA10. (a)** Crystal structure of PLSCR1 nuclear localization signal (orange, Gly257–Ile266, numbering correspond to UniProt entry O15162) in complex with Importin α (blue), Protein Data Bank (PDB) ID 1Y2A (ref. [65]). Side chains of PLSCR1 are shown as connected spheres with carbon atoms coloured in orange, nitrogens in blue and oxygens in red. Hydrogen atoms were not determined at this resolution (2.20) and are not shown. **(b)** Close-up view showing side chains of PLSCR1 Ser260, His262 and Importin Glu107 as sticks. Distance (in) between selected atoms (PLSCR1 His262 $N\epsilon2$ and Importin Glu107 carboxyl O) is indicated. A hydrogen bond between PLSCR1 His262 and Importin Glu107 is indicated with a dashed line. The risk variant is predicted to eliminate this bond, disrupting nuclear import, an essential step for effect on antiviral signalling[27] and neutrophil maturation[66]. **(c)** Because there is very strong sequence conservation between IFNA10 and the gene encoding IFNω, we used existing crystal structure data (Protein Data Bank ID 3SE4 (ref. [67])) for IFNω (cyan) to display a ternary complex with interferon α/β receptor IFNAR1 (blue), IFNAR2 (red). The side chain of Trp164 is shown as spheres and indicated with a black line. **(d)** The hydrophobic core of IFNω with Trp164 shielded from the solvent in the center. Trp164-surrounding residues of IFNω are numbered and correspond to UniProt entry P05000. Trp164 and surrounding residues are conserved in IFNA10 (UniProt ID P01566) and share the same numbering as in IFNω (P05000). Side chains of four residues are shown as sticks. Carbon and nitrogen atoms coloured in cyan and blue, respectively. The critical COVID-19-associated mutation, Trp164Cys, would replace an evolutionarily conserved, bulky side chain in the hydrophobic core of IFNA10 with a smaller one, which may destabilize IFNA10.

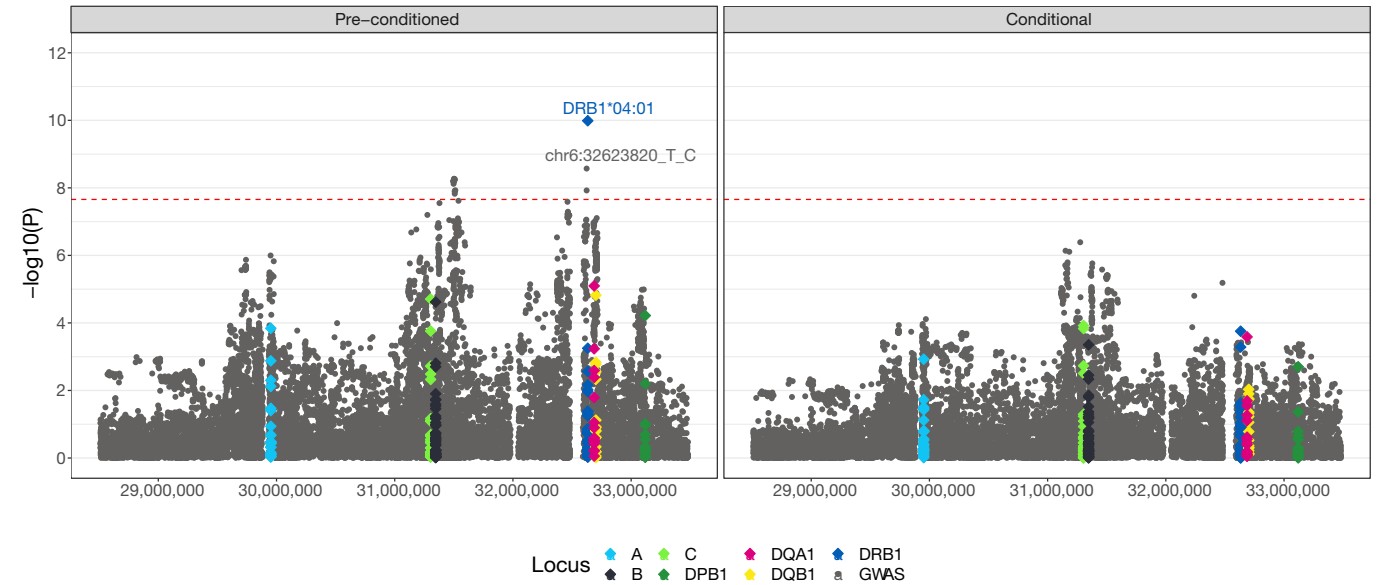

**Extended Data Fig. 6 | Manhattan plot of HLA and GWAS signal across the extended MHC region for the EUR cohort.** Grey circles mark the GWAS (small variant) associations and diamonds represent the HLA each allele association, coloured by locus. The lead variant from the GWAS and lead allele from HLA are labelled. The left-panel shows the raw association −log₁₀(*P* values) per variant - prior to conditional analysis. The right-panel shows the −log₁₀(*P* values) per variant following conditioning on DRB1*04:01. The dashed red line shows the Bonferroni-corrected genome-wide significance threshold for Europeans.

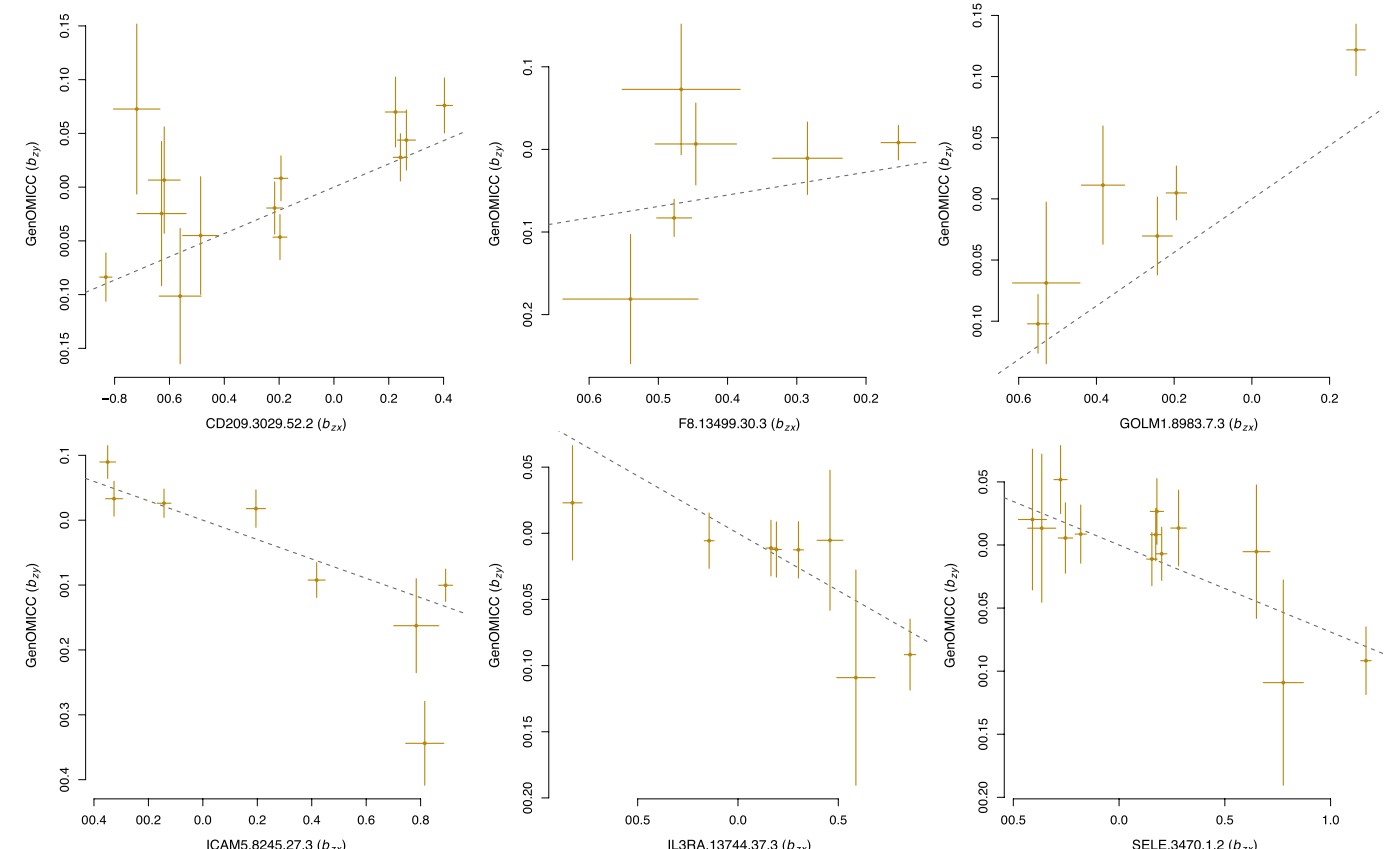

**Extended Data Fig. 7 | Effect–effect plots for Mendelian randomization analyses to assess causal evidence for circulating proteins in critical COVID-19.** Each plot shows effect size (β) of variants associated with protein concentration (*x* axis) and critical COVID-19 (*y* axis). A full list of instruments is found in Supplementary Table 13.

**Extended Data Table 1 | Fine-mapping results for lead variants and worst consequence variant in each credible set**

| Lead variant | Pop | Focal CS | nCS | Worst variant | Worst variant Pval | Lead variant CADD | Worst variant CADD | Worst Consequence | Worst gene |
|---|---|---|---|---|---|---|---|---|---|
| chr1:155066988:C:T | EUR | chr1:155197995:A:G | 9 | chr1:155066988:C:T | $6.8 \times 10^{-10}$ | 10.3 | 10.3 | synonymous | *EFNA4* |
| chr1:155175305:G:A | META | chr1:155197995:A:G | 5 | chr1:155134292:T:C | $1.17 \times 10^{-07}$ | 8.34 | 1.37 | 3' UTR | *EFNA1* |
| chr1:155197995:A:G | EUR | chr1:155197995:A:G | 3 | chr1:155202934:T:C | $3.15 \times 10^{-12}$ | 2.1 | 21.2 | missense | *THBS3* |
| chr3:45796521:G:T | EUR | chr3:45859597:C:T | 1 | chr3:45796521:G:T | $9.9 \times 10^{-17}$ | 9.19 | 9.19 | 5' UTR | *SLC6A20* |
| chr3:45859597:C:T | EUR | chr3:45859597:C:T | 9 | chr3:45825948:A:G | $6.1 \times 10^{-132}$ | 0.143 | 7.96 | 3' UTR | *LZTFL1* |
| chr3:146517122:G:A | EUR | chr3:146517122:G:A | 9 | chr3:146517122:G:A | $4.94 \times 10^{-09}$ | 22.6 | 22.6 | missense | *PLSCR1* |
| chr5:131995059:C:T | EUR | chr5:131995059:C:T | 32 | chr5:132075767:T:C | $1.48 \times 10^{-09}$ | 0.206 | 6.09 | missense | *CSF2* |
| chr6:32623820:T:C | EUR | chr6:32623820:T:C | 33 | chr6:32467073:G:C | $6.65 \times 10^{-08}$ | 10.1 | 8.12 | intron | *HLA-DRB9* |
| chr6:41515007:A:C | META | chr6:41515652:G:C | 8 | chr6:41515652:G:C | $5.17 \times 10^{-08}$ | 4.11 | 4.17 | intron | *LINC01276* |
| chr9:21206606:C:G | EUR | chr9:21206606:C:G | 3 | chr9:21206606:C:G | $1.93 \times 10^{-09}$ | 23.9 | 23.9 | missense | *IFNA10* |
| chr11:34482745:G:A | EUR | chr11:34482745:G:A | 4 | chr11:34479140:G:A | $2.56 \times 10^{-10}$ | 0.073 | 1.32 | 3' UTR | *ELF5* |
| chr12:132489230:GC:G | EUR | chr12:132489230:GC:G | 25 | chr12:132565387:T:C | $1.42 \times 10^{-07}$ | 4.91 | 4.64 | non coding transcript exon | - |
| chr13:112889041:C:T | EUR | chr13:112889041:C:T | 4 | chr13:112886111:C:T | $5.36 \times 10^{-11}$ | 0.676 | 5.5 | 3' UTR | *ATP11A* |
| chr15:93046840:T:A | EUR | chr15:93046840:T:A | 2 | chr15:93046840:T:A | $8.61 \times 10^{-13}$ | 4.45 | 4.45 | intron | *RGMA* |
| chr16:89196249:G:A | EUR | chr16:89196249:G:A | 4 | chr16:89196249:G:A | $4.4 \times 10^{-09}$ | 22.8 | 22.8 | missense | *SLC22A31* |
| chr17:46152620:T:C | EUR | chr17:46152620:T:C | 1430 | chr17:45830530:T:C | $1.14 \times 10^{-07}$ | 5.27 | 3.96 | stop lost | *CRHR1* |
| chr17:49863260:C:A | EUR | chr17:49863260:C:A | 5 | chr17:49880589:C:T | $1.91 \times 10^{-09}$ | 5.38 | 7.22 | TF binding site | - |
| chr19:4717660:A:G | EUR | chr19:4717660:A:G | 1 | chr19:4717660:A:G | $3.91 \times 10^{-36}$ | 16.3 | 16.3 | intron | *DPP9* |
| chr19:10305768:G:A | EUR | chr19:10352442:G:C | 3 | chr19:10380329:C:G | $7.93 \times 10^{-11}$ | 0.422 | 7.91 | intron | *TYK2* |
| chr19:10352442:G:C | EUR | chr19:10352442:G:C | 1 | chr19:10352442:G:C | $6.98 \times 10^{-17}$ | 25.1 | 25.1 | missense | *TYK2* |
| chr19:48697960:C:T | EUR | chr19:48697960:C:T | 10 | chr19:48703346:C:T | $6.75 \times 10^{-10}$ | 2.44 | 7.02 | synonymous | *FUT2* |
| chr21:33230000:C:A | EUR | chr21:33230000:C:A | 16 | chr21:33262573:G:A | $5.35 \times 10^{-21}$ | 10.1 | 3.43 | missense | *IFNAR2* |
| chr21:33287378:C:T | EUR | chr21:33230000:C:A | 33 | chr21:33288868:T:G | $1.59 \times 10^{-07}$ | 3.63 | 5.84 | intron | *IL10RB* |
| chr21:33959662:T:TAC | EUR | chr21:33230000:C:A | 23 | chr21:33972178:G:A | $1.32 \times 10^{-08}$ | 0.246 | 0.282 | non coding transcript exon | *LINC00649* |

Fine-mapping was performed in EUR for all variants except chr6:41515007:A:C, which was fine-mapped in the SAS population for which the signal was strongest among the per-population analyses. The lead variant chr2:60480453:A:G (rs1123573) that was discovered in multi-ancestry meta-analysis is not included in the table as fine-mapping did not generate any credible sets with the required posterior inclusion probability of >0.95 for any of the populations. Focal CS is the index SNP that was used for fine-mapping with SusieR, 1.5 Mb on each side. nCS indicates the number of variants included in each credible set. Consequence annotation for all variants across credible sets was generated using VEP v.104 and the worst consequence across GENCODE basic transcripts was chosen. All variants were ranked according to their consequence type and chr:pos$_{hg38}$:ref$_{hg38}$:alt, *P* value and CADD score are provided for the variant with the worst consequence across all variants in each credible set.

**Extended Data Table 2 | Identification of 16 proteins by the GSMR analysis for COVID-19 severity at FDR < 0.05**

| Gene | $BETA$ | $SE$ | $P$ | $BETA_{hgib2.23m}$ | $SE_{hgib2.23m}$ | $P_{hgib2.23m}$ |
|------|--------|------|-----|---------------------|-------------------|------------------|
| ICAM5 | -0.15 | 0.025 | $2.82 \times 10^{-9}$ | -0.07 | 0.013 | $7.65 \times 10^{-8}$ * |
| GOLM1 | 0.22 | 0.037 | $2.92 \times 10^{-9}$ | 0.20 | 0.021 | $1.04 \times 10^{-21}$ * |
| ICAM1 | 0.10 | 0.017 | $6.33 \times 10^{-9}$ | 0.013 | 0.009 | 0.14 |
| ICAM5 | -0.19 | 0.033 | $1.58 \times 10^{-8}$ | -0.048 | 0.017 | 0.0054 |
| FAM3D | 0.13 | 0.024 | $2.12 \times 10^{-8}$ | 0.12 | 0.013 | $3.12 \times 10^{-18}$ * |
| PDGFRL | 0.10 | 0.021 | $1.85 \times 10^{-6}$ | 0.021 | 0.010 | 0.041 |
| CD209 | 0.11 | 0.024 | $6.58 \times 10^{-6}$ | 0.11 | 0.014 | $1.88 \times 10^{-15}$ * |
| ABO | 0.064 | 0.017 | 0.00012 | 0.084 | 0.0088 | $7.76 \times 10^{-22}$ * |
| C1GALT1C1 | 0.13 | 0.037 | 0.00026 | 0.055 | 0.030 | 0.063 |
| CCL25 | 0.15 | 0.040 | 0.00026 | 0.035 | 0.023 | 0.13 |
| F8 | 0.14 | 0.042 | 0.0011 | 0.16 | 0.020 | $1.46 \times 10^{-14}$ * |
| TLR4:LY96 | -0.12 | 0.038 | 0.0014 | - | - | - |
| IL3RA | -0.087 | 0.028 | 0.0019 | -0.065 | 0.014 | $4.33 \times 10^{-6}$ * |
| SELE | -0.069 | 0.022 | 0.0019 | -0.095 | 0.013 | $3.76 \times 10^{-14}$ * |
| CAMK1 | -0.064 | 0.021 | 0.00205 | 0.0047 | 0.0110 | 0.664 |
| IL27RA | -0.084 | 0.028 | 0.00229 | 0.0020 | 0.0150 | 0.892 |

We report the effect size BETA, the standard error SE and the P value *P* for the GenOMICC analysis and the replication with HGI B2 and 23andme meta-analysis. An asterisk (*) next to the replication *P* value ($P_{hgib2.23m}$) indicates that the protein result is replicated with concordant direction of effect. We considered as replicated those results that passed a Bonferroni correction of the *P* values of the replicated outcome Mendelian randomization.

# Reporting Summary

## Statistics

For all statistical analyses, confirm that the following items are present in the figure legend, table legend, main text, or Methods section.

| n/a | Confirmed | |
|---|---|---|
| ☐ | ☒ | The exact sample size (*n*) for each experimental group/condition, given as a discrete number and unit of measurement |
| ☐ | ☒ | A statement on whether measurements were taken from distinct samples or whether the same sample was measured repeatedly |
| ☐ | ☒ | The statistical test(s) used AND whether they are one- or two-sided<br>*Only common tests should be described solely by name; describe more complex techniques in the Methods section.* |
| ☐ | ☒ | A description of all covariates tested |
| ☐ | ☒ | A description of any assumptions or corrections, such as tests of normality and adjustment for multiple comparisons |
| ☐ | ☒ | A full description of the statistical parameters including central tendency (e.g. means) or other basic estimates (e.g. regression coefficient) AND variation (e.g. standard deviation) or associated estimates of uncertainty (e.g. confidence intervals) |
| ☐ | ☒ | For null hypothesis testing, the test statistic (e.g. $F$, $t$, $r$) with confidence intervals, effect sizes, degrees of freedom and $P$ value noted<br>*Give P values as exact values whenever suitable.* |
| ☐ | ☒ | For Bayesian analysis, information on the choice of priors and Markov chain Monte Carlo settings |
| ☒ | ☐ | For hierarchical and complex designs, identification of the appropriate level for tests and full reporting of outcomes |
| ☐ | ☒ | Estimates of effect sizes (e.g. Cohen's *d*, Pearson's *r*), indicating how they were calculated |

*Our web collection on statistics for biologists contains articles on many of the points above.*

## Software and code

Policy information about availability of computer code

| | |
|---|---|
| Data collection | Nucleon Kit (Cytiva), Chemagic 360 platform using Chemagic DNA blood kit (Perkin Elmer) . T Qubit and normalised Illumina TruSeq DNA PCR-Free High Throughput Sample Preparation kit, Illumina Hiseq X instrument (for 100,000 Genomes Project samples), NovaSeq instrument (for the COVID-19 critical and mild cohorts). |
| Data analysis | VerifyBAMID, DRAGEN(v3.2.22), Illumina North Star Version 4 Whole Genome Sequencing Workflow (NSV4, version 2.6.53.23), iSAAC Aligner (version 03.16.02.19), Starling Small Variant Caller (version 2.4.7), GVGCFGenotyper (GG) v3.8.1, vt v0.57721, gvcfgenotyper v2019.02.26, bcftools v1.10.2, plink 1.9, HiSeq+NSV4, HiSeq+Pipeline 2.0, NovaSeq+Pipeline 2.0, KING 2.1, plink2, GCTA v1.93.1_beta, Strelka2, SAIGE v0.44.5, GTCA 1.9.3, SusieR v0.11.42, VEPv104, metal 2018-08-28, MetaSubtract package (v1.60), R(v4.0.2), SAIGE-GENE v0.44.5, LOFTEE, VEPv99, MetaXCan (v0.6.5), coloc R package 5.1.0, R 3.6.3, GSMR, Heidi, HIBAG R package 1.8.3, XGR package (20-Apr-2020), LDSC (v1.0.1), HDL(v.1.4.0), REGENIEv2.2 |

For manuscripts utilizing custom algorithms or software that are central to the research but not yet described in published literature, software must be made available to editors and reviewers. We strongly encourage code deposition in a community repository (e.g. GitHub). See the Nature Portfolio guidelines for submitting code & software for further information.

March 2021

## Data

Policy information about [availability of data](availability of data)

All manuscripts must include a [data availability statement](data availability statement). This statement should provide the following information, where applicable:

- Accession codes, unique identifiers, or web links for publicly available datasets
- A description of any restrictions on data availability
- For clinical datasets or third party data, please ensure that the statement adheres to our [policy](policy)

> Full summary data in support of the findings of this study will available for download from https://genomicc.org/data concurrently with publication. Individual-level data can be analysed by qualified researchers in the UK Outbreak Data Analysis Platform at the University of Edinburgh by application at  https://genomicc.org/data. Genomicc data for 1,000,000 genomes participants and cases are available through the Genomics England research environment.
> The full GWAS summary statistics for the 23andMe discovery data set will be made available through 23andMe to qualified researchers under an agreement with 23andMe that protects the privacy of the 23andMe participants. Please visit https://research.23andMe.com/dataset-access/ for more information and to apply to access the data.

# Field-specific reporting

Please select the one below that is the best fit for your research. If you are not sure, read the appropriate sections before making your selection.

☒ Life sciences   ☐ Behavioural & social sciences   ☐ Ecological, evolutionary & environmental sciences

For a reference copy of the document with all sections, see [nature.com/documents/nr-reporting-summary-flat.pdf](nature.com/documents/nr-reporting-summary-flat.pdf)

# Life sciences study design

All studies must disclose on these points even when the disclosure is negative.

| | |
|---|---|
| Sample size | Cases: n=7,491, controls n=48,400 (mild cases n=1630, 100k controls n=46,770).<br>European ancestry ncases=5,989 ncontrols=41,891<br>South Asian ancestry ncases=788, ncontrols=3,793<br>African ancestry ncases=440, ncontrols=1,350<br> East Asian ancestry ncases=274, ncontrols=366 |
| Data exclusions | no exclusions |
| Replication | Data from the Host Genetics Initiative (HGI) data freeze 6 B2 analysis (hospitalised cases) were combined in a meta-analysis with data shared by 23andMe.Inc . We removed signals in HGI derived from GenOMICC cases for independence. 22 of the 25 independent GWAS signals were replicated.<br>Replication of rs28368148 and rs4424872 was attempted using a trans-ancestry meta-analysis UKB, AncestryDNA, Penn Medicine Biobank (PMBB), andGeisinger Health Systems (GHS) totaling 9937 hospitalized COVID-19 casesand 1,059,390 controls (COVID-19 negative or unknown).<br>1 extra loci (rs28368148) was replicated using this method. The loci not replicated correspond to the lead snps: rs9271609 in HLA region and rs4424872 next to RGMA. |
| Randomization | Not relevant to the study. There wasn't any allocation to experimental groups |
| Blinding | Not relevant to the study. |

# Reporting for specific materials, systems and methods

We require information from authors about some types of materials, experimental systems and methods used in many studies. Here, indicate whether each material, system or method listed is relevant to your study. If you are not sure if a list item applies to your research, read the appropriate section before selecting a response.

## Materials & experimental systems

| n/a | Involved in the study |
|---|---|
| ☒ | ☐ Antibodies |
| ☒ | ☐ Eukaryotic cell lines |
| ☒ | ☐ Palaeontology and archaeology |
| ☒ | ☐ Animals and other organisms |
| ☐ | ☒ Human research participants |
| ☒ | ☐ Clinical data |
| ☒ | ☐ Dual use research of concern |

## Methods

| n/a | Involved in the study |
|---|---|
| ☒ | ☐ ChIP-seq |
| ☒ | ☐ Flow cytometry |
| ☒ | ☐ MRI-based neuroimaging |

# Human research participants

Policy information about studies involving human research participants

**Population characteristics**

Severe COVID-19 (n=7491). Significant comorbidities: 1,605. Died(60 days) 2154, Invasive Ventilation 4028.
mean age=60, mean BMI=29.9
European ancestry (n=5,989): males 4,062 females 1,927.
South Asian ancestry (n=788): males:586, females:202
East Asian ancestry(n=274) males:162, females:112
African ancestry(n=440) males:286, females 154
100K controls: 18,915 unaffected family members of rare diseases participants. 14,701 affected rare diseases participants, 1,005 not assessed for disease status, 12,149 cancer participants.
mean age: 51
mean BMI: 26.1
European ancestry (n=41,384): males 18,971females 22,413
South Asian ancestry (n=3698): males:1802, females:1896
East Asian ancestry(n=352) males:138, females:224
African ancestry(n=1,236) males:632, females:704
Mild COVID-19 cohort. mean age: 46.
European ancestry (n=1507): males:410, females:1,097
South Asian ancestry (n=95): males:43, females:52
East Asian ancestry(n=14) males:8, females:6
African ancestry(n=14) males:6, females:8

**Recruitment**

Critically ill patients recruited to the GenOMICC study (genomicc.org) had confirmed COVID-19 according to local clinical testing and were deemed, in the view of the treating clinician, to require continuous cardiorespiratory monitoring. In UK practice this kind of monitoring is undertaken in high dependency or intensive care units. Patients were recruited from 224 ICU across the UK.
Participants were recruited to the mild COVID-19 cohort on the basis of having experienced mild (non-hospitalised) or asymptomatic COVID-19. Participants volunteered to take part in the study via a microsite and were required to self-report the details of a positive COVID-19 test. Volunteers were prioritised for genome sequencing based on demographic matching with the critical COVID-19 cohort considering self-reported ancestry, sex, age and location within the UK.
Participants were enrolled in the 100,000 Genomes Project from families with a broad range of rare diseases, cancers and infection by 13 regional NHS Genomic Medicine Centres across England and in Northern Ireland, Scotland and Wales. For this analysis, participants for whom a positive SARS-CoV-2 test had been recorded as of March, 2021 were not included due to uncertainty in the severity of COVID-19 symptoms. Only participants for whom genome sequencing was performed from blood derived DNA were included and participants with haematological malignancies were excluded to avoid potential tumour contamination.

**Ethics oversight**

Research ethics committees (Scotland 15/SS/0110, England, Wales and Northern Ireland: 19/WM/0247. Current and previous versions of the study protocol are available at genomicc.org/protocol.
UKBiobank Study: ethical approval for the UK Biobank was previouslyobtained from the North West Centre for Research Ethics Committee(11/NW/0382). The work described herein was approved by UK Biobank under application number 26041.
GHS study: approval for DiscovEHR analyseswas provided by the Geisinger Health System Institutional Review Board under project number 2006-0258.
AncestryDNA study: all data for this research project was from subjects who provided prior informed consent to participate in AncestryDNA's Human Diversity Project, as reviewed and approved by our external institutional review board, Advarra (formerlyQuorum). All data was de-identified prior to use.
PMBBstudy: appropriate consent was obtained from each participant regarding storage of biological specimens, genetic sequencing and genotyping, andaccess to all available EHR data. This study was approved by the Institutional Review Board of the University of Pennsylvania and complied with the principles set out in the Declaration of Helsinki. Informed consent was obtained for all study participants.

Note that full information on the approval of the study protocol must also be provided in the manuscript.

