## [Peer Review File · Nature]

Manuscript Title: Whole genome sequencing identifies multiple loci for critical illness caused by COVID-19

Reviewer Comments & Author Rebuttals

Reviewer Reports on the Initial Version:

Referees' comments:

Referee #1 (Remarks to the Author):

This manuscript is an extension of work completed by this group previously evaluating COVID-19 severe outcomes in critical respiratory patients with population and mild/asymptomatic controls using sequencing. The team identifies 22 loci in Europeans and an additional 3 loci with a meta-analysis across ancestries. They also use transcriptomics and HLA classical alleles to support their findings. Overall, it is a great contribution to the field. However, there are several reservations about the approach that do deserve some attention.

1a) There are very limited details on the 100,000 Genomes project population. It's described briefly as patients broadly representing -- rare diseases, cancers and infections. More details are necessary on age, sex, etc. Especially because these are topics included in the supplementary materials but not really addressed. Is there lack of a sex or age difference because of the distribution of these traits in cases/controls?

1b) Additionally, we know that guidance for immunocompromised individuals was to take extra cautions to avoid COVID-19 infections. This means limited exposures. So there is a likely selection bias in these controls. Do they actually represent the population? You do have mild/asymptomatic controls and those should be utilized. They can help to support your findings by showing that the direction of the effects (even if p values are not significant) is the same, as is the allele frequencies. These are known exposed individuals and it will also make your inferences much stronger. Inclusion of both controls (separate findings) and their allele frequencies and results in a table should be in the main text. The supplementary table on cohort characteristics is not sufficient, and does not represent any cohort characteristics, but sequencing characteristics.

2) The tables included in the manuscript and the supplementary tables do not (unless I missed it) indicate the numbers per category. This is important as we interpret the findings. The forest plots have the direction of the effect and OR and CI, but no allele frequencies or N to capture what might be happening. A fixed effect model is also chosen for the meta-analysis which is interesting given the diversity of your population. Trans-ancestry meta analysis programs may be better suited to handle this diversity (Mantra). Although it's hard to determine if this is really a diverse study. The readers need to know who is being studied, what the age, sex, ethnicity breakdowns are and co-morbidities.

3) The paper is also set up to be the finding of 22 variants in Europeans and an additional 3 in non-Europeans + europeans via the meta-analysis. But, were there any findings outside of Europeans? How do you interpret lack of replication of variants in a sub-population. From the plots we cannot fully interpret (2nd plot not labeled) Were the cases in each sub-population all at the same severity level? The table is set up with RAF of Europeans, but you discuss trans-ancestry. If the approach is trans-ancestry then full information on all populations should be provided.

4) One novel finding is FUT2. I caution interpretation of FUT2 because its know that ABO, secretor status, and Lewis genotypes are all highly dependent on ethnicity or geographical ancestry. This should be explored more to be sure it is not a European gradient effect (either by cases or controls or both) . It could be evaluated by looking at tight clusters of European ancestry individuals from the PCA and seeing this effect remains or is supported.

5) The use of controls sequenced on a different platform can often lead to batch effects or other technical artifacts. Discussion of this concern should be included in the manuscript, along with measures on mitigation. This is a recognized issue in the field, especially for sequencing and could result in spurious outcomes.

6) The lack of rare variant gene based associations is intriguing, especially since you restricted to putative snps. Are the lead variants from the GWAS all common variation, with no rare variants underlying these associations? Or is it just that the test for gene based rare variants is not significant? I think those are two different points, but important, because this sequencing study has the opportunity to fine map these regions. The supplementary section has information on fine mapping that should be included in the main text. In my opinion, this is what moves this paper beyond the GWAS reported by this group and others. Can we identify putative causal alleles, and/or a putative causal gene with the sequencing. At the very least the Chromosome 3 region and narrowing down to a gene should be highlighted. Credible sets are included in the discussion, but there is no table or way to visualize these credible sets, and if in the conditional analysis a single SNP explains these associations.

Referee #2 (Remarks to the Author):

A. Summary of the key results

Kousathanas et al., on behalf of the GenOMICC study, describe the most extensive effort to date at delineating the human genetic determinants of life-threatening COVID-19. They present the analysis whole genome variation in 7,491 patients with critical COVID-19 recruited in >200 UK intensive care units, comparing them with 48,400 population controls. The same group previously reported (Nature volume 591, pages 92–98 (2021)) a genotyping-based GWAS performed using data from 2,244 critical COVID-19 cases, which identified 4 association signals. Now, the increase in sample size coupled with the use of whole genome sequencing and statistical fine-mapping has led to the identification of 25 independent association signals (22 of which replicated in another dataset), included 15 new associations.

B. Originality and significance

These results are timely and important, as the newly identified genes and variants are potentially informative about SARS-CoV-2 pathogenic mechanisms causing the most severe clinical presentations. The road to concrete clinical translation of the findings is obviously not straightforward, as duly acknowledged by the authors, but a large diffusion of the study results – and of the underlying genetic data – is the surest way to

ensure that the global biomedical research community quickly builds on these promising leads.

C. Data & methodology: validity of approach, quality of data, quality of presentation

The bioinformatic tools and strategies used for quality control, alignment and variant calling from the whole genome sequencing data are state-of-the-art in the field and very well described.

A remark concerning the fine mapping methods (lines 568-569): instead of selecting the worst consequence across all available transcripts, where many might not have much biological relevance, I would suggest sticking to the canonical transcript or at least transcripts known to be expressed according to GTEx.

The post-GWAS analyses (TWAS and colocalisation analysis) also follow recognized standards. The choice of focusing on lung and blood expression data makes biological sense and the addition of a pan-GTEx TWAS (metaTWAS sheet in 'TWAS results' suppl. table) is more confusing than useful.

D. Appropriate use of statistics and treatment of uncertainties

The single variant association analyses and aggregate variant testing are carefully performed. Remarkable efforts have been made to address and avoid potential batch effects. Still, the fact that cases and controls were sequenced on different platforms remains a potential issue, which is well explained in the 'Limitations' section.

E. Conclusions: robustness, validity, reliability

Most of the independent associations identified in the study (22/25) have been replicated in an independent, yet phenotypically distinct cohort (see below, suggested improvements). They are therefore robust and likely to be further validated by other ongoing genomic studies.

As a note of caution, the authors mention in the Discussion the lack of replication of the association with OAS gene variants that was reported in their previous GWAS (lines 264-270). They also didn't find convincing associations in additional regions that were identified by others, including TLR7 and several genes involved in the regulation of type I and III interferon, as reported in lines 175 to 179. A more general discussion on winner's curse and other potential reasons for lack of replication would be useful for the community.

F. Suggested improvements: experiments, data for possible revision

A better description of the study population is needed. Suppl. figures 9 and 10 show an imbalance between cases and controls in terms of age and sex. However, no numbers are given. A table summarizing the differences in basic demographic parameters between cases and controls would be useful. Also, it would be very interesting to know about risk factors / comorbidities among cases, which would allow for better dissection of the genetic signals. For example, is there any way for the authors to obtain BMI data? Combined with age, such information could help better understand the association results (e.g., through sensitivity analyses in young, non-obese patients, or in older patients with high BMI).

As replication cohort, the authors picked the Host Genetic Initiative (HGI) GWAS meta-analysis round 6 "hospitalised COVID vs. population" (B2 analysis). Another HGI association analysis - "very severe respiratory confirmed covid vs. population" (A2 analysis) - would have been much closer phenotypically to the severe COVID-19 phenotype studied here. The rationale not to use it was that "there are currently insufficient cases from other sources available to attempt replication" (line 141). It is true that GenOMICC contributed 1'825

cases to the analysis, according to the numbers shown here: <https://www.covid19hg.org/results/r6/>. Still, another 6'954 cases are available (together with about 1mio controls), which is more than enough for replication purpose. I suggest to include that analysis as an additional validation step.

G. References: appropriate credit to previous work?

References are complete and up-to-date.

H. Clarity and context: lucidity of abstract/summary, appropriateness of abstract, introduction and conclusions

The abstract, intro and conclusion are written in a concise and lucid way. In both the abstract and the conclusion, the authors suggest that the way to translating their results to clinical use is through “testing in clinical trials” (abstract) or through “large-scale randomised trials” (conclusion). While it is certainly true that clinical trials will be needed, it should also be mentioned that more fundamental research into pathogenic mechanisms and immune response is required – in light of the new findings – before anything can be tested in clinical trials.

Referee #3 (Remarks to the Author):

Summary

Kousathanas and Pairo-Castineira et al reported a whole-genome sequencing study of critical COVID-19. This is an incremental work from their previous publication on a genome-wide association study of critical illness in COVID-19 (Pairo-Castineira et al. Nature 2020). An 3-fold increase in case sample size (2,244  7,491) and deeper genotyping enables the authors to uncover 15 novel risk loci associated with critical COVID-19 (totallying of 22 risk loci). While it is crucial to highlight the importance of deep-phenotyping in the COVID-19, and in infectious diseases study design, I found the presented work has limited improvement to our current knowledge on the host genetics of COVID-19. In particular, the amount of overlapping is large - 30% of their currently enrolled severe cases were included in the COVID-19 HGI publication; and the increase in the total case and control numbers were relatively small -- 20% increase in the case numbers (6,179 → 7,491), and 30-fold decrease in control numbers (1,483,780 → 48,400) when comparing it to the recent publication from the COVID-19 Host Genetic Initiative (Nature 2021). The authors fail to present some critical information when a finding is presented or it is buried in the supplementary information. Overall, I found the current work lack of scientific vigor and could be improved in clarity. My specific comments and concerns are listed below:

Major comments:

1. Study design: the advantage of using a whole-genome sequenced cohort versus genotype + imputation is unclear/ not discussed here. Since most of the study participants are European individuals, whether one would gain more information by whole-genome sequencing versus genotyping + imputation is unclear. Since the authors had emphasised this study design in their title and in the abstract, thus it warrants some comments for the value of their current WGS study design.

2. Definition of independent regions. The authors performed a conditional analysis using summary statistics using GCTA cojo. However, given the authors have the individual-level genotype data, it would be more convincing to the readers if the reported risk loci were done using a conditional analysis with individual-level genotype data. In particular, a few of the reported “independent” regions are very close to each other (e.g. rs7528026 and rs41264915 are only ~22KB apart; rs73510898 and rs34536443 are < 50KB apart and are not complete dependent ($r^2 > 0$)). In Figure 3b, the authors highlighted such an example for two “independent” risk

loci in chromosome 21, but it would be more convincing to show the regional plot after the condition on the top association signal (rs17860115).

3. Only 1/25 listed regions share the same top SNP as their previous work in Nature 2020 (rs73064425). Is it a difference due to technology (genotyping+imputation versus WGS) or statistical method (logistic linear model versus linear mixed model). It would be useful to see the association statistics of the previously reported eight SNPs (Table 1) in the current cohort, especially their posterior probabilities for being causal if it is the same locus.

4. Likely causal variants or credible sets from fine-mapping results are not presented in the manuscript.

5. Table 1 and 2 contain duplicate yet inconsistent results? For example, rs1143011457 has different p-values in two tables. The authors should also consider to merge two tables to reduce duplicated information.

6. The authors may consider incorporating the coloc results into Figure 2 by coloring their points by the colocalization posterior probability. Or adding it to Table 1 / 2.

7. The analysis and discussion around the HLA association is incomplete. The listed SNP in the HLA region is a upstream SNP of HLA-DQA1, but the authors identified a HLA-DRB1*04:01 association instead after HLA imputation. Have the authors performed conditional analysis to identify the causal allele or amino acid positions? In addition, mentioned briefly in the method section, the authors are currently performing inference of HLA alleles directly from whole genome sequences instead of imputation. The authors indicated 96% concordance between HIBAG and HLA*LA results for all HLA alleles, but what about HLA-DRB1*04:01 specifically?

8. It is unclear to me whether the authors presented the critically-ill cases versus mild COVID-19 controls findings. Or how the mild COVID-19 controls were used.

9. In SF4C (PC3 versus PC4), there is a clear difference between cases versus controls. Please can the authors show the PC loading for the reported SNPs to make sure the reported risk variants are not due to population stratification.

Minor comments:

1. Authors should present the number of individuals included in the study broken down by ancestry (Line 122).

2. Authors put their fine-mapping analysis (Line 153-164) under the Replication section, which shouldn't be the case.

3. Information missing in Line 457 for the complex region list.

4. What is the number of variants included in the rare burden analysis shown in SF1?

Whole genome sequencing identifies multiple loci for critical
illness caused by COVID-19
Responses to reviewers

1 Referee #1 (Remarks to the Author):

This manuscript is an extension of work completed by this group previously evaluating COVID-19 severe outcomes in critical respiratory patients with population and mild/asymptomatic controls using sequencing. The team identifies 22 loci in Europeans and an additional 3 loci with a meta-analysis across ancestries. They also use transcriptomics and HLA classical alleles to support their findings. Overall, it is a great contribution to the field. However, there are several reservations about the approach that do deserve some attention.

We thank the reviewer for their generous comments and have endeavoured to respond to each point below.

- 1) (a) There are very limited details on the 100,000 Genomes project population. Its described briefly as patients broadly representing – rare diseases, cancers and infections. More details are necessary on age, sex, etc. Especially because these are topics included in the supplementary materials but not really addressed. Is there lack of a sex or age difference because of the distribution of these traits in cases/controls?

We agree about the importance of this information and have added additional description of the 100,000 Genomes Project control group in the supplementary information. Firstly, we added a figure showing a detailed phenotypic breakdown of the 100,000 Genomes Project control cohort (Supplementary figure 9). Secondly, we added and updated figures to show the numerical breakdown and distribution for age, sex and body mass index (BMI) across the different cohorts, including the 100,000 Genomes Project controls (Supplementary Figures 10 and 11, Supplementary Table 4).

With regard to differences in genetic predisposition by sex and age, we predict that these effects will be difficult to detect with high confidence due to the smaller effect sizes expected for comparisons within the extreme-susceptibility case group. The Manhattan plots in Supplementary Figure 18 suggest that we may see genome-wide significant differences in these analyses with a larger population in future. We also added a new age, sex, BMI -matched study including a subset of participants (see Supplementary Figure 20 and our reply to reviewer 2, comment F).

- 1) (b) Additionally, we know that guidance for immunocompromised individuals was to take extra cautions to avoid COVID-19 infections. This means limited exposures. So there is a likely selection bias in these controls. Do they actually represent the population? You do have mild/asymptomatic controls and those should be utilized. They can help to support your findings by showing that the direction of the effects (even if p values are not significant) is the same, as is the allele frequencies. These are known exposed individuals and it will also make your inferences much stronger. Inclusion of both controls (separate findings) and their allele frequencies and results in a table should be in the main text. The supplementary table on cohort characteristics is not sufficient, and does not represent any cohort characteristics, but sequencing characteristics.

The additional details we have added in the response above mitigate this risk. For the control population used in this study, unrelated participants were selected from the 100,000 Genomes Project cohort, which includes participants with rare disorders and their family members, and participants with a range of different cancer types. The control group contains 18,915 unaffected family members of rare disease participants, 14,701 affected rare disease participants (not related to the unaffected family members selected) and 12,149 cancer participants. Affected participants with immunological disorders account for <1% of the control group. We have now added a detailed disease breakdown (Supplementary Figure 9) and expanded the cohort description in the results to include this information (under Cohort characteristics).

However the effect of shielding is a potentially important bias that we had not considered. A fraction of our control does include otherwise healthy individuals that were recruited for this study and have had only mild symptoms. In order to address this, we have performed a new GWAS analysis using only the mild COVID-19 cohort as controls and the full severe COVID-19 cohort as cases. As the sample size of the mild cohort controls is small, there is a substantial reduction in power but results for lead loci are consistent with our main results. We have included these results in a new table (Supplementary Table 11), which includes allele frequencies of cases, controls and the mild-only cohort, and refer to them in the main manuscript.

- 2) The tables included in the manuscript and the supplementary tables do not (unless I missed it) indicate the numbers per category. This is important as we interpret the findings. The forest plots have the direction of the effect and OR and CI, but no allele frequencies or N to capture what might be happening. A fixed effect model is also chosen for the meta-analysis which is interesting given the diversity of your population. Trans-ancestry meta analysis programs may be better suited to handle this diversity (Mantra). Although it's hard to determine if this is really a diverse study. The readers need to know who is being studied, what the age, sex, ethnicity breakdowns are and co-morbidities.

We agree with the need to include a better description of the study population and have now provided additional characterisation in the Supplementary Material. We also have added allele frequency data for cases and controls in supplementary table 8.

In response to the suggestion regarding the methodology for the meta-analysis, we repeated the meta-analysis using a random effects model and MR-MEGA,^[1] which is designed to account for ancestry differences adding principal components to a fixed-effect model. In order to run MR-MEGA, the number of populations needs to be higher than the number of added pcs+2. Only variants that passed QC in all 4 populations could be tested with MR-MEGA (4M variants remain) adding 1 pc.

We did not see important changes in effects for the lead variants tested (Response to Review Figure 1). Errors and p-values for the three methods are compared for the lead variants in the meta-analysis in

Response to Review Table 1).

top variant	BETA.fixed	SE.fixed	P.fixed	BETA.mrmega	SE.mrmega	P.mrmega	BETA.random	SE.random	P.random
chr1:155066988:C:T	0.87	0.14	1.507×10^{-9}	-	-	-	0.87	0.14	6.80×10^{-10}
chr1:155175305:G:A	0.33	0.056	7.16×10^{-9}	-	-	-	0.33	0.056	3.74×10^{-9}
chr1:155197995:A:G	-0.22	0.032	3.79×10^{-12}	-0.20	0.016	6.49×10^{-12}	-0.19	0.047	3.83×10^{-5}
chr2:60480453:A:G	-0.12	0.020	9.85×10^{-10}	-0.15	0.0083	1.31×10^{-9}	-0.14	0.034	3.88×10^{-05}
chr3:45796521:G:T	0.23	0.029	2.49×10^{-15}	0.40	0.18	1.97×10^{-15}	0.30	0.15	0.049
chr3:45859597:C:T	0.92	0.035	2.18×10^{-152}	0.90	0.27	6.85×10^{-151}	0.86	0.11	1.12×10^{-14}
chr3:146517122:G:A	0.21	0.038	1.52×10^{-8}	-	-	-	0.21	0.037	7.49×10^{-9}
chr5:132441275:T:C	0.18	0.029	4.479×10^{-10}	0.19	0.02	3.15×10^{-9}	0.18	0.029	2.10×10^{-10}
chr6:32623820:T:C	-0.12	0.022	1.27×10^{-8}	-	-	-	-0.10	0.069	0.14
chr6:41515007:A:C	-0.37	0.048	7.59×10^{-15}	-0.38	0.052	6.91×10^{-14}	-0.37	0.048	5.78×10^{-15}
chr9:21206606:C:G	0.56	0.095	4.10×10^{-9}	-	-	-	0.56	0.093	1.92×10^{-9}
chr11:34482745:G:A	-0.14	0.020	1.62×10^{-11}	-0.11	0.028	6.38×10^{-11}	-0.13	0.03	3.87×10^{-5}
chr12:132489230:G:C:G	0.12	0.020	1.12×10^{-9}	-	-	-	0.12	0.019	1.25×10^{-10}
chr13:112889041:C:T	0.16	0.023	1.61×10^{-12}	0.13	0.037	2.7×10^{-12}	0.14	0.053	0.0081
chr15:93046840:T:A	-0.45	0.093	1.99×10^{-6}	-	-	-	-0.37	0.49	0.45
chr16:89196249:G:A	0.17	0.029	6.04×10^{-9}	0.23	0.045	3.47×10^{-8}	0.17	0.028	2.93×10^{-9}
chr17:46152620:T:C	-0.14	0.025	1.40×10^{-8}	-	-	-	-0.14	0.025	6.79×10^{-9}
chr17:49863260:C:A	0.35	0.054	1.46×10^{-10}	-	-	-	0.31	0.098	0.0017
chr19:4717660:A:G	0.28	0.021	6.44×10^{-39}	0.23	0.030	8.51×10^{-39}	0.27	0.063	1.81×10^{-5}
chr19:10305768:G:A	0.21	0.035	1.47×10^{-9}	-	-	-	0.068	0.18	0.70
chr19:10352442:G:C	0.41	0.049	4.22×10^{-17}	-	-	-	0.41	0.047	1×10^{-17}
chr19:48697960:C:T	0.12	0.020	3.74×10^{-10}	0.11	0.032	2.19×10^{-9}	0.097	0.036	0.0080
chr21:33230000:C:A	0.23	0.021	6.28×10^{-28}	0.23	0.025	8.64×10^{-27}	0.23	0.020	9.45×10^{-29}
chr21:33287378:C:T	0.15	0.023	4.23×10^{-12}	0.10	0.037	2.46×10^{-11}	0.16	0.022	1.55×10^{-12}
chr21:33959662:T:TAC	0.20	0.035	1.78×10^{-8}	-	-	-	0.23	0.038	1.24×10^{-9}

Response to Review Table 1: Meta-analysis results for three different approaches (fixed, random and MR-MEGA).

- 3) The paper is also set up to be the finding of 22 variants in Europeans and an additional 3 in non-Europeans + europeans via the meta-analysis. But, were there any findings outside of Europeans? How do you interpret lack of replication of variants in a sub-population. From the plots we cannot fully interpret (2nd plot not labeled) Were the cases in each sub-population all at the same severity level? The table is set up with RAF of Europeans, but you discuss trans-ancestry. If the approach is trans-ancestry then full information on all populations should be provided.

We had previously left the labels off the second plot to indicate the new findings from trans-ancestry analysis. In order to make comparison easier, we have added the labels for all peaks to this plot (Figure 1).

Cases were defined the same way across ancestry groups. We include Manhattan plots for each ancestry independently in Supplementary Figure 12, showing genome-wide significant replication of the association at 3p21.31 in the South Asian ancestry group but no new associations in non-European ancestry groups. This is consistent with the limited numbers of cases in these groups, and hence the lower statistical power to detect ancestry-specific associations. We observe no heterogeneity of effect for 23 hits (Supplementary Figure 14); there is significant heterogeneity in two loci, both highlighted in Table 1 and shown in Supplementary Table 8.

- 4) One novel finding is FUT2. I caution interpretation of FUT2 because its know that ABO, secretor status, and Lewis genotypes are all highly dependent on ethnicity or geographical ancestry. This should be explored more to be sure it is not a European gradient effect (either by cases or controls or both). It could be evaluated by looking at tight clusters of European ancestry individuals from the PCA and seeing this effect remains or is supported.

We appreciate this suggestion and have completed the analysis as proposed. We did not find any evidence of a gradient effect.

Response to Review Figure 1: Manhattan plot showing results obtained with MR-MEGA

Specifically, we defined increasingly tight clusters of European ancestry and performed a logistic regression of case/control status on the lead variant at *FUT2* (rs368565) and the same covariates as in the main analysis. Clusters were defined as follows; we projected all individuals of European ancestry on the three leading population-specific genomic principal components and computed the mean individual. We then considered subsets comprising 80%, 60%, 40% and 20% of individuals with the lowest distance to the mean individual. These are illustrated in Response to Review Figure 2

The results of analyses on these subsets are summarised in Response to Review Table 1, showing the estimated OR and its 95% confidence interval for the lead SNP for each of the subsets as well as for the whole sample as reported in the main manuscript.

Sample	OR	OR _{CI}
All EUR	1.15	1.1-1.2
80%	1.15	1.1-1.2
60%	1.14	1.1-1.2
40%	1.15	1.1-1.2
20%	1.20	1.1-1.3

There is no evidence of a systematic change in effect for increasingly tight subsets.

- 5) The use of controls sequenced on a different platform can often lead to batch effects or other technical artifacts. Discussion of this concern should be included in the manuscript, along with measures on mitigation. This is a recognized issue in the field, especially for sequencing and could result in spurious outcomes.

We fully agree with the reviewer and have taken extensive steps to mitigate the possibility of batch effects. Firstly, we masked low quality genotypes (with low GQ, low depth or aberrant ab-ratio) and then filtered stringently for low missingness to "homogenise" the different platforms. Secondly, we performed a control-control relative allele frequency filter based on samples that were aligned in both processing platforms (Illumina NSV4 vs. Genomics England Pipeline 2.0), which removes most

Response to Review Figure 2: PCA plot with different colors indicating individuals in different quintiles of the distance distribution

of the batch effect noise. Thirdly, we performed an LD-support based filter where we assessed the significance of each variant by the support of its neighbours. This meant that significant variants with no LD-partners or LD-partners with inconsistent GWAS summaries were eliminated. These filters might have been overly stringent, but we deemed them necessary to drastically reduce false-positives.

We have added these points to the discussion as suggested.

- 6) The lack of rare variant gene based associations is intriguing, especially since you restricted to putative snps. Are the lead variants from the GWAS all common variation, with no rare variants underlying these associations? Or is it just that the test for gene based rare variants is not significant? I think those are two different points, but important, because this sequencing study has the opportunity to fine map these regions. The supplementary section has information on fine mapping that should be included in the main text. In my opinion, this is what moves this paper beyond the GWAS reported by this group and others. Can we identify putative causal alleles, and/or a putative causal gene with the sequencing. At the very least the Chromosome 3 region and narrowing down to a gene should be highlighted. Credible sets are included in the discussion, but there is no table or way to visualize these credible sets, and if in the conditional analysis a single SNP explains these associations.

This suggestion inspired us to extend our fine-mapping analysis to check whether rare variants could underlie the association signals discovered. As we explain in detail in new supplementary section "Genetic fine-mapping check for rare variants", we performed a new fine-mapping analysis with susieR for our largest EUR cohort using all variants with $MAF > 0.02\%$ (reduced from 0.5%) around the 23 EUR-discovered signals. This analysis did not reveal any new potential causal variants for our main signals (shown in Supplementary Table 10).

We agree with the reviewer that the fine mapping is a major strength of the work and have moved

a larger part of this analysis into a prominent position in the main manuscript. We have also added an additional Supplementary Table 9 to show other potential candidates that we identified for each credible set and have also performed an enrichment analysis using genes identified from fine-mapping and colocalisation (Supplementary Figure 15).

2 Referee #2 (Remarks to the Author):

A. Summary of the key results

Kousathanas et al., on behalf of the GenOMICC study, describe the most extensive effort to date at delineating the human genetic determinants of life-threatening COVID-19. They present the analysis whole genome variation in 7,491 patients with critical COVID-19 recruited in >200 UK intensive care units, comparing them with 48,400 population controls. The same group previously reported (Nature volume 591, pages 92–98 (2021)) a genotyping-based GWAS performed using data from 2,244 critical COVID-19 cases, which identified 4 association signals. Now, the increase in sample size coupled with the use of whole genome sequencing and statistical fine-mapping has led to the identification of 25 independent association signals (22 of which replicated in another dataset), included 15 new associations.

B. Originality and significance

These results are timely and important, as the newly identified genes and variants are potentially informative about SARS-CoV-2 pathogenic mechanisms causing the most severe clinical presentations. The road to concrete clinical translation of the findings is obviously not straightforward, as duly acknowledged by the authors, but a large diffusion of the study results – and of the underlying genetic data – is the surest way to ensure that the global biomedical research community quickly builds on these promising leads.

We thank the reviewer for their support for the work and agree that there is considerable urgency in making these findings widely available.

C. Data & methodology: validity of approach, quality of data, quality of presentation

The bioinformatic tools and strategies used for quality control, alignment and variant calling from the whole genome sequencing data are state-of-the-art in the field and very well described. A remark concerning the fine mapping methods (lines 568-569): instead of selecting the worst consequence across all available transcripts, where many might not have much biological relevance, I would suggest sticking to the canonical transcript or at least transcripts known to be expressed according to GTEx.

As suggested, to increase the biological relevance of our variant annotation, we have modified our approach to use only transcripts included in the GENCODE basic set, which is a smaller set of transcripts with full-length protein coding transcripts [2]. We utilised the worst predicted consequence type across those transcripts as we feel it is potentially more clinically relevant. We updated Table 1. The only change compared with our previous annotation is for variant chr19:4717660:A:G impacting *DDP9*, which we now annotate as intronic rather than missense. For verification, we also checked the consequence type for the lead variants in the canonical transcript and only one difference was observed compared with the annotation using the worst consequence type: variant chr3:45859597:C:T (rs73064425) is an intronic variant in the longer transcript of *LZTFL1* (ENST00000539217.5), but approximately 18

kb upstream of the shorter, canonical transcript (ENST00000648206.1) and intronic in the canonical transcript of RNA gene AC099782.2. We have added this comparison in Supplementary Excel File GWAS.xlsx.

The post-GWAS analyses (TWAS and colocalisation analysis) also follow recognized standards. The choice of focusing on lung and blood expression data makes biological sense and the addition of a pan-GTEx TWAS (metaTWAS sheet in ‘TWAS results’ suppl. table) is more confusing than useful.

We agree that this table is difficult to interpret and have removed it from the supplementary table.

D. Appropriate use of statistics and treatment of uncertainties

The single variant association analyses and aggregate variant testing are carefully performed. Remarkable efforts have been made to address and avoid potential batch effects. Still, the fact that cases and controls were sequenced on different platforms remains a potential issue, which is well explained in the ‘Limitations’ section.

We thank the reviewer for recognising our efforts to correct for batch effects and have added to this section in response to comments from this and other reviewers.

E. Conclusions: robustness, validity, reliability

Most of the independent associations identified in the study (22/25) have been replicated in an independent, yet phenotypically distinct cohort (see below, suggested improvements). They are therefore robust and likely to be further validated by other ongoing genomic studies.

As a note of caution, the authors mention in the Discussion the lack of replication of the association with OAS gene variants that was reported in their previous GWAS (lines 264-270). They also didn’t find convincing associations in additional regions that were identified by others, including TLR7 and several genes involved in the regulation of type I and III interferon, as reported in lines 175 to 179. A more general discussion on winner’s curse and other potential reasons for lack of replication would be useful for the community.

We agree that this is an important point and have expanded the discussion to explain the winner’s curse in order to make it clearer to non-specialists.

F. Suggested improvements: experiments, data for possible revision

A better description of the study population is needed. Suppl. figures 9 and 10 show an imbalance between cases and controls in terms of age and sex. However, no numbers are given. A table summarizing the differences in basic demographic parameters between cases and controls would be useful. Also, it would be very interesting to know about risk factors / comorbidities among cases, which would allow for better dissection of the genetic signals. For example, is there any way for the authors to obtain BMI data? Combined with age, such information could help better understand the association results (e.g., through sensitivity analyses in young, non-obese patients, or in older patients with high BMI).

We have added the table of characteristics as suggested by this reviewer and reviewer 1. In response to the specific suggestion to include BMI, we have obtained linked data for a majority of cases from intensive care national audit data (ICNARC Case Mix Programme).

Response to Review Figure 3: Results for lead variants of this study comparing effect size (BETA) and P -values for EUR GWAS analyses using cases with matched and unmatched controls. Left panel shows the results of the matched study which used default covariates of age, sex, age \times sex and 20 PCs. Middle panels show results of a study using unmatched controls of the same sample size as the matched study and using covariates as the principal study gwas using only default covariates. Right panels show results of an unmatched study using default + BMI as covariate. Results for EUR-discovered loci are shown in black and with grey the trans-ancestry meta-analysis results are shown. Error bars for BETA represent standard errors of estimates.

We have repeated the GWAS analysis using a subset of the study population, matching case and controls groups for BMI and other characteristics. As expected, there is a reduction in statistical power in this analysis but the direction and magnitude of effect at genome-wide significant loci are consistent. We report these results in a new section in the Supplementary Information, "Matched case-control analysis". For convenience, we reproduce the key figure (Supplementary Figure 21) in Response to Review Figure 3.

As replication cohort, the authors picked the Host Genetic Initiative (HGI) GWAS meta-analysis round 6 “hospitalised COVID vs. population” (B2 analysis). Another HGI association analysis - “very severe respiratory confirmed covid vs. population” (A2 analysis) - would have been much closer phenotypically to the severe COVID-19 phenotype studied here. The rationale not to use it was that “there are currently insufficient cases from other sources available to attempt replication” (line 141). It is true that GenOMICC contributed 1,825 cases to the analysis, according to the numbers shown here: <https://www.covid19hg.org/results/r6/>. Still, another 6,954 cases are available (together with about 1mio controls), which is more than enough for replication purpose. I suggest to include that analysis as an additional validation step.

We have added this additional replication, using the HGI “A2” data set, in Supplementary Table 8.

We selected the "B2" analysis *a priori*, using the same approach as we took in our previous paper in 2020.[3] Although we agree that the "A2" analysis has improved during the intervening 9 months,

there are important differences between the carefully-selected, prospectively-recruited GenOMICC case group, and the broader cohort meeting the definition of "critical COVID-19" in our meta-analysis with HGI. Both our analyses, and the leave-one-out statistics from HGI, demonstrate that discovery power in the GenOMICC cohort is substantially greater than in the remaining 75% of cases included in the HGI critical COVID-19 analysis. This is likely to be due to a combination of factors, including random variation and the winner's curse, but also in study design, such as:

1. International differences in critical care practice - patients admitted to intensive care units (ICU) in the UK are generally younger, sicker and less comorbid than in other healthcare systems, creating a more extreme phenotype more suitable for population comparisons.^[4]
2. Prospective case ascertainment may have improved the quality of phenotyping in GenOMICC compared with some retrospective studies.

We discuss this issue further below in response to comments from reviewer 3.

G. References: appropriate credit to previous work?
References are complete and up-to-date.

H. Clarity and context: lucidity of abstract/summary, appropriateness of abstract, introduction and conclusions

The abstract, intro and conclusion are written in a concise and lucid way. In both the abstract and the conclusion, the authors suggest that the way to translating their results to clinical use is through "testing in clinical trials" (abstract) or through "large-scale randomised trials" (conclusion). While it is certainly true that clinical trials will be needed, it should also be mentioned that more fundamental research into pathogenic mechanisms and immune response is required – in light of the new findings – before anything can be tested in clinical trials.

We have edited both the abstract and concluding paragraph to make this clearer.

3 Referee #3 (Remarks to the Author):

Summary Kousathanas and Pairo-Castineira et al reported a whole-genome sequencing study of critical COVID-19. This is an incremental work from their previous publication on a genome-wide association study of critical illness in COVID-19 (Pairo-Castineira et al. Nature 2020). An 3-fold increase in case sample size (2,244 → 7,491) and deeper genotyping enables the authors to uncover 15 novel risk loci associated with critical COVID-19 (totallying of 22 risk loci). While it is crucial to highlight the importance of deep-phenotyping in the COVID-19, and in infectious diseases study design, I found the presented work has limited improvement to our current knowledge on the host genetics of COVID-19. In particular, the amount of overlapping is large - 30% of their currently enrolled severe cases were included in the COVID-19 HGI publication; and the increase in the total case and control numbers were relatively small – 20% increase in the case numbers (6,179 → 7,491), and 30-fold decrease in control numbers (1,483,780 → 48,400) when comparing it to the recent publication from the COVID-19 Host Genetic Initiative (Nature 2021). The authors fail to present some critical information when a finding is presented or it is buried in the supplementary information. Overall, I found the current work lack of scientific vigor and could be improved in clarity. My specific comments and concerns are listed below:

We thank the reviewer for their in-depth comments. Our objective in this work is to find new genetic

associations with severe COVID-19, which we have done with exceptional efficiency. Whilst comparing the total number of cases and controls between studies is a reasonable starting point, we would argue that relying only on the raw numbers risks overlooking the importance of careful study design and conduct. The clinical insight underlying our study design, together with careful prospective recruitment, are the key advantages of GenOMICC, and the benefits are demonstrated by our discovery of 15 new genetic associations.

More specifically, our cases were defined as COVID-19 positive patients who were assessed as critically-ill by the treating clinician. This case definition is qualitatively different from even the most severe definition of the HGI meta-analysis phenotype (*i.e.*, analysis "A" for COVID-19 positive hospitalised patients who died or required respiratory support). As discussed above, the critically ill population included in GenOMICC in the UK have been selected by clinical admission practice to be younger, and less frail, than critically ill populations in other healthcare systems [4].

Furthermore, an important consideration that is not incorporated into HGI case definitions is that some patients who die of COVID-19 do not have hypoxaemic respiratory failure: we have demonstrated this in our previous autopsy studies [5] and in the ISARIC4C study [6]. Importantly, many non-critically ill patients have distinct pathophysiology underlying their disease, indicated by the opposite effect of steroid treatment on mortality, which we discovered in the RECOVERY trial.[7] Death in this group is strongly influenced by frailty and comorbidity; hence using unselected hospital mortality as an outcome, as has been used in A2, is highly likely to add noise to a study of genetic susceptibility. In contrast, critical COVID-19 is a remarkably homogeneous - and extreme - phenotype.

We would also point out that increasing the size of the control cohort in an imbalanced study increases power only asymptotically. Using a calculation of the effective sample size as:

$$N_{eff} = \frac{2}{1/N_{cases} + 1/N_{controls}}$$

[8]

we obtain $N_{eff}=12,307$ for the cited HGI study and $N_{eff}=12,974$ for the present study. Therefore, the size of the control cohort that was used in this study did not appreciably reduce power compared with the HGI analysis, even though the number of individuals in the control cohort was much larger in the latter.

Our prediction that the GenOMICC design would be substantially more powerful is confirmed by the data. As observed by reviewer 2 above, although there are 6,179 cases in our recent meta-analysis report with HGI (and similar N_{eff}), that report only discovered one new genetic association with critical COVID-19 (rs77534576, near *TAC4*) - taking the total number of associations with critical illness to 6.

In contrast, in our present report of 22 genome-wide significant and replicated associations with critical COVID-19, only 6 have been previously reported to be associated with that phenotype, and 15 are completely new findings, which we believe is a substantial increment on previous knowledge.

Major comments:

1. Study design: the advantage of using a whole-genome sequenced cohort versus genotype + imputation is unclear/ not discussed here. Since most of the study participants are European individuals, whether one would gain more information by whole-genome sequencing versus genotyping + imputation is unclear. Since the authors had emphasised this study design in their title and in the abstract, thus it warrants some comments for the value of their current WGS study design.

We thank the reviewer for their comments. In response, we are pleased to emphasise the benefits of whole genome sequencing (WGS) in a revised manuscript.

As COVID-19 is a new disease with unknown genetic architecture, our design employed the best available sequencing strategy to investigate genetic variation across the full allele frequency spectrum underlying COVID-19 severity. More specifically, we used high-coverage WGS instead of employing a genotyping and imputation approach to accurately interrogate common, low frequency and rare variants, providing particular benefit for rarer variants which are less likely to be part of reference panels employed for imputation, and less likely to be well imputed [9].

Although our rare variant gene-level testing identified no gene-wide significant signals, our GWAS study with more common variants successfully identified many new loci. We agree with the reviewer that most of the common signals could have been found with genotype + imputation design, but WGS data provided our study with several benefits, including improved fine-mapping, and excellent coverage of indels and multi-allelic variants, many of which are represented in the credible sets for new association signals. These variants are often overlooked because they are harder to impute. As we explain in detail in new supplementary section "Genetic fine-mapping check for rare variants", we performed a new fine-mapping analysis using all variants with $MAF > 0.02\%$ (reduced from 0.5%) around the 23 EUR-discovered signals. This analysis did not reveal any new potential causal variants for our main signals (shown in Supplementary Table 10). It is thus also important to note that the absence of rare variants driving the new and known genetic signals here is in itself a valuable contribution to our understanding of the genetic architecture of COVID-19. We have amended the discussion to make this clearer.

Moreover, we have added a Mendelian randomisation analysis, which makes use of the high-resolution genotype data to maximise the number of overlapping variants with existing protein expression data, and enrichment analysis that leverages the additional missense mutations in genes identified by fine-mapping together with results from TWAS and coloc to discover relevant gene pathways for COVID-19 (Extended Data Table 1 Supplementary Table 15).

Finally, we have added a second protein-coding structure change to the main manuscript, since additional replication data obtained during the review process indicates that the association in *IFNA10* replicates robustly in a second study (Regeneron and Ancestry.com). This, together with the amino acid substitution in *PLSCR1*, provides an indication of the value of fine mapping with WGS. “

2. Definition of independent regions. The authors performed a conditional analysis using summary statistics using GCTA cojo. However, given the authors have the individual-level genotype data, it would be more convincing to the readers if the reported risk loci were done using a conditional analysis with individual-level genotype data. In particular, a few of the reported “independent” regions are very close to each other (e.g. rs7528026 and rs41264915 are only ~22KB apart; rs73510898 and rs34536443 are < 50KB apart and are not complete dependent ($r^2 > 0$)). In Figure 3b, the authors highlighted such an example for two “independent” risk loci in chromosome 21, but it would be more convincing to show the regional plot after the condition on the top association signal (rs17860115).

We are grateful to the reviewer for this suggestion, which in our view improves the robustness of our conditional analysis. We have run this confirmatory analysis using individual-level data with SAIGE. All our main loci remain significant after conditioning on the other lead signals on the same chromosome (new Supplementary Table 6). We have added a description of the additional methods and refer to the new results in the main manuscript.

3. Only 1/25 listed regions share the same top SNP as their previous work in Nature 2020 (rs73064425). Is it a difference due to technology (genotyping+imputation versus WGS) or statistical method (logistic linear model versus linear mixed model). It would be useful to see the association statistics of the previously reported eight SNPs (Table 1) in the current cohort, especially their posterior probabilities for being causal if it is the same locus.

We agree this is a useful link to our previous report and have added this as a column in supplementary table 7. Apart from differences with genotyping + imputation versus WGS, for the previous work we applied a strict threshold for QC using only SNPs with high quality imputation (score >0.9) in both UK BioBank and our imputed data, therefore not including variants that have now been tested.

4. Likely causal variants or credible sets from fine-mapping results are not presented in the manuscript.

We thank the reviewer for this suggestion, especially given that the fine-mapping is a strength of our approach, and have now added Supplementary Table 9 showing summary info about the size of the credible sets and added probable causal variants that have the worst predicted consequence type among the variants in the credible set. We provide full fine-mapping credible sets in Supplementary File GWAS.xlsx.

5. Table 1 and 2 contain duplicate yet inconsistent results? For example, rs1143011457 has different p-values in two tables. The authors should also consider to merge two tables to reduce duplicated information.

We agree this is a better way to present our results. The reason for the discrepancy is that a different set of results (from trans-ancestry meta-analysis) was presented in the second table. We think this is now much clearer in the revised version of Table 1 where we have merged important information for previous table 1 and table 2. We have added supplementary table 8 to present the full results of the replication meta-analysis with HGI.

6. The authors may consider incorporating the coloc results into Figure 2 by coloring their points by the colocalization posterior probability. Or adding it to Table 1 / 2.

We greatly appreciate this suggestion by the reviewer which helped us improve Figure 2 to add different colors for loci with strong and moderate evidence of colocalisation (PP[H4], >0.8 and >0.5, respectively), which we show below. We provide the actual numerical values for posterior probabilities for colocalisation of all variants in the supplementary excel file TWAS.xlsx.

7. The analysis and discussion around the HLA association is incomplete. The listed SNP in the HLA region is an upstream SNP of HLA-DQA1, but the authors identified a HLA-DRB1*04:01 association instead after HLA imputation. Have the authors performed conditional analysis to identify the causal allele or amino acid positions? In addition, mentioned briefly in the method section, the authors are currently performing inference of HLA alleles directly from whole genome sequences instead of imputation. The authors indicated 96% concordance between HIBAG and HLA*LA results for all HLA alleles, but what about HLA-DRB1*04:01 specifically?

We have conducted a conditional analysis on HLA-DRB1*04:01 and have observed no other significantly associated HLA alleles or SNPs. We have now edited the main text to clarify this point and moved the figure presenting the conditional analysis to main manuscript as Extended Data Figure 4.

We did not conduct analysis at the amino-acid level. Following the reviewer’s comment, we specifically looked at the concordance between HIBAG and HLA-LA results for the HLA-DRB1*04:01 calls. For the HLA-DRB1*04:01 carriers in HIBAG, we report a concordance (*i.e.* both calls agreeing at the G-group resolution between the HIBAG and HLA-LA callsets) of 87.6% across 7,927 EUR samples for which we also had HLA-LA data. The concordance is comparable across samples processed using different pipelines (86.5% for samples processed with NSV4, 87.9% for samples processed with Pipelines 2.0) and case control status (88.2% in cases, 87.5% in controls). We also re-ran our association analysis on concordant samples only using the same model as for the full analysis, and confirmed the observed association with HLA-DRB1*04:01 ($OR = 0.78, 95\%CIs : 0.75 - 0.81, P = 1.3 \times 10^{-11}$) which is stronger than for the lead variant ($OR = 0.88, 95\%CIs : 0.86 - 0.90, P = 4.4 \times 10^{-9}$), consistent with our results on the full HIBAG callset. We have also added this to the supplementary material.

Edited main text (in italics): *The lead variant in the HLA region, rs9271609, lies upstream of HLA-DQA1 and HLA-DRB1 genes.* To investigate the contribution of specific HLA alleles to the observed association in the HLA region, we imputed HLA alleles at a four digit (two-field) level using HIBAG[10]. The only allele that reached genome-wide significance was HLA-DRB1*04:01 ($OR = 0.80, 95\%CI = 0.75 - 0.86, P = 1.6 \times 10^{-10}$ in EUR), which has a stronger *P*-value than the lead SNP in the region ($OR : 0.88, 95\%CIs : 0.84 - 0.92, P = 3.3 \times 10^{-9}$ in EUR) and is a better fit to the data ($AIC_{DRB1*04:01} = 30241.34, AIC_{leadSNP} = 30252.93$). Upon conditioning on HLA-DRB1*04:01, no other HLA alleles or small variants remained genome-wide significant (Extended Data Figure 4).

8. It is unclear to me whether the authors presented the critically-ill cases versus mild COVID-19 controls findings. Or how the mild COVID-19 controls were used.

We apologise for the lack of clarity in explaining how the mild individuals were used. Mild individuals

were used in the GWAS analysis as controls together with individuals from the 100,000 Genomes Project. This is shown in the displayed diagram of the study design (Supplementary Figure 1). We have elaborated on this point in the caption of this figure to improve clarity. In light of comments received from reviewers, we have also performed an additional GWAS analysis with severe cases compared with only the individuals who have experienced mild or asymptomatic COVID-19 to further support the main results (see response comment 1B to reviewer 1).

9. In SF4C (PC3 versus PC4), there is a clear difference between cases versus controls. Please can the authors show the PC loading for the reported SNPs to make sure the reported risk variants are not due to population stratification.

We thank the reviewer for this comment. We did not observe obvious case/control imbalances from observation of the projection PCs (Supplementary Figure 4C and reproduced in Response to Review Figure 3 for convenience).

Note that in Supplementary Figure 4 (panels B-C), gray symbols represent background samples from the 1KGP3. We have amended the figure to make this more clear. We note that we primarily use this figure to check whether our cases and controls have been assigned correctly to ancestry groupings and cluster together. As we have performed population-specific GWAS, we also present population-specific PCs, which were used as covariates for the association analyses and are shown in Supplementary Figures 5,6,7,8). We did not observe case/control stratification in these population specific PCs either.

In addition to visual assessment and following the reviewers suggestion, we now also calculated SNP loadings for major axes of population-specific variation of PCs 1-5 for AFR, EAS, EUR and SAS ancestry groups. We calculated loadings for all 25 lead variants of this study and plotted them against the background of 58,925 high quality independent variants that were used for computing the PCA and controlling for stratification.

As shown in the loadings figure below, the majority of signals do not display unusual stratification, although there are four outliers. For PC1 these are rs2271616 for EUR, SAS, AFR, EAS populations,

rs73064425 for EUR and SAS populations and rs114301457 for SAS. For PC3, it is rs9271609 for the SAS population. All of these loci were discovered in the EUR-only analysis. The two loci (rs2271616 and rs73064425) that have outlier EUR loadings have been previously reported as being associated with COVID-19 phenotypes [11, 12] and we replicated these findings in this study (i.e, we do not report them as new findings here, see also main table 1).

Response to Review Figure 4: Thick dashed lines indicate 3 standard deviations away from the mean for each loading.

As a further check we also investigated whether for rs2271616 and rs73064425 population-specific stratification could be driving the signals. We performed logistic regression with plink2 using the same covariates as in the main analysis, but selecting individuals in successively tighter population centric clusters of individuals. These subsamples were computed by considering the quintiles of the distribution of distance between individuals and the mean individual of the population in PCA space, illustrated in Response to Review Figure 5.

Response to Review Figure 5: PCA for FUT2

The results for these EUR subsets are summarised in Response to Review Table 2 showing the estimated OR and 95% confidence interval for the two loci. There is no evidence for a major change in effect size for increasingly tight clusters apart from a small attenuation for rs73064425 that could also be due to

sample variance or true covariance between severe COVID-19 risk and the PC1 axis.

Variant ID	%Samples (EUR)	OR	OR_{CI}
	All	1.27	1.21-1.32
	0.8	1.24	1.18-1.3
rs2271616	0.6	1.24	1.16-1.31
	0.4	1.25	1.16-1.34
	0.2	1.21	1.09-1.34
	All	2.31	2.25-2.37
	0.8	2.28	2.21-2.35
rs73064425	0.6	2.21	2.13-2.29
	0.4	2.19	2.09-2.29
	0.2	2.08	1.94-2.22

Response to Review Table 2

Minor comments:

1. Authors should present the number of individuals included in the study broken down by ancestry (Line 122).

We thank the reviewer for pointing this omission from the main paper. We have now added the ancestry details in main results.

2. Authors put their fine-mapping analysis (Line 153-164) under the Replication section, which shouldn't be the case.

We have placed the fine-mapping analysis to the appropriate section in the results, i.e, after the GWAS analysis description and before replication.

3. Information missing in Line 457 for the complex region list.

We thank the reviewer for pointing our omission. The complex region list was taken from: [https://genome.sph.umich.edu/wiki/Regions_of_high_linkage_disequilibrium_\(LD\)](https://genome.sph.umich.edu/wiki/Regions_of_high_linkage_disequilibrium_(LD)), lifted over for GRCh38 coordinates. We have edited the manuscript accordingly.

4. What is the number of variants included in the rare burden analysis shown in SF1?

The number of variants is provided in Supplementary Table AVTsuppinfo.xlsx, sheet E and we have added this reference in the main text results section.

References

- [1] Mägi, R. *et al.* Trans-ethnic meta-regression of genome-wide association studies accounting for ancestry increases power for discovery and improves fine-mapping resolution. *Human Molecular Genetics* **26**, 3639–3650 (2017). URL <https://doi.org/10.1093/hmg/ddx280>. <https://academic.oup.com/hmg/article-pdf/26/18/3639/24646976/ddx280.pdf>.
- [2] Frankish, A. *et al.* Gencode reference annotation for the human and mouse genomes. *Nucleic Acids Research* **47**, D766–D773 (2019). URL <https://academic.oup.com/nar/article/47/D1/D766/5144133>.
- [3] Païro-Castineira, E. *et al.* Genetic mechanisms of critical illness in Covid-19. *Nature* 1–1 (2020).
- [4] S, M. & H, W. Clinical review: International comparisons in critical care - lessons learned. *Critical care (London, England)* **16** (2012). URL <https://pubmed.ncbi.nlm.nih.gov/22546146/>.
- [5] DA, D. *et al.* Tissue-specific immunopathology in fatal covid-19. *American journal of respiratory and critical care medicine* **203**, 192–201 (2021). URL <https://pubmed.ncbi.nlm.nih.gov/33217246/>.
- [6] Docherty, A. B. *et al.* Features of 20 133 UK patients in hospital with covid-19 using the ISARIC WHO Clinical Characterisation Protocol: Prospective observational cohort study. *BMJ* **369** (2020).
- [7] Horby, P. *et al.* Dexamethasone in Hospitalized Patients with Covid-19 — Preliminary Report. *New England Journal of Medicine* (2020).
- [8] Winkler, T. W. *et al.* Quality control and conduct of genome-wide association meta-analyses. *Nature Protocols* 2014 9:5 **9**, 1192–1212 (2014). URL <https://www.nature.com/articles/nprot.2014.071>.
- [9] Wu, Y., Zheng, Z., Visscher, P. M. & Yang, J. Quantifying the mapping precision of genome-wide association studies using whole-genome sequencing data. *Genome Biology* **18**, 86 (2017).
- [10] Zheng, X. *et al.* HIBAG - HLA genotype imputation with attribute bagging. *Pharmacogenomics Journal* **14**, 192–200 (2014).
- [11] Ellinghaus, D. *et al.* Genomewide association study of severe covid-19 with respiratory failure. *The New England journal of medicine* **383**, 1522–1534 (2020).
- [12] COVID-19 Host Genetics Initiative. Mapping the human genetic architecture of COVID-19. *Nature* (2021). URL <https://doi.org/10.1038/s41586-021-03767-x>.

Reviewer Reports on the First Revision:

Referees' comments:

Referee #1 (Remarks to the Author):

The authors have considered the previous comments carefully, and have addressed the comments. My biggest concern is that many of these comments are not included in the main manuscript, but instead presented as tables or figures in the extensive supplementary section. This concerns me because these are real issues that you have tried to address and by not acknowledging them in the paper even in a few sentences, you will have others question this data as well, or misinterpret the findings. In addition, you are still holding back some of the key points that global investigators will find interesting--the fine mapping, credible sets, and conditional analyses.

1) The description of 100,000 genomes is now in the manuscript, but please include comments on this population and how this will or will not affect your results. Using rare cancer patients is not representative of the general population. Even if you summarize some of the disease breakdown. Cancer patients can also be considered immunocompromised if on treatment...but at the least they were cautious on exposures during the pandemic. The additional supplementary table 11 data can be included in a few lines of text.

2) The interesting findings seem to be in the supplementary section! The credible set information and the fine mapping really should be incorporated into the main text. For example, the chromosome 3 finding--suggesting two hits under your peak is important!

You have the best data to get at what might be happening under these peaks (supplementary page 30) --and you have some diversity in your population to fine map it. Please make this more prominent in the main manuscript. Similarly, the additional conditional analyses should be addressed in the manuscript even if the figures remain in the supplementary.

It seems the authors are perhaps focused on rare variants not underlying their associations (a great point) but then not addressing how the rare variants (and LD blocks/recombination) actually help to narrow down the existing published work.

3) Age and ancestry may be confounded. In the age section please address the plots that show in some populations the age of the controls is very different than the age of the cases. The focus in the subsequent plots is on Europeans and age, but the other ancestry populations are not balanced and that may lead to some spurious results. See Supp Fig 10

4) In the individual conditional analysis, what is used in the conditional analysis? The effects seem to get stronger (more significant p-value) --can you include the region and or rsid that you are conditioning on?

5) In the comparison to 2020 paper--two regions are less significant. Have those been replicated in other studies? Or are you indicating that they were initially false positives/spurious associations? Supplementary Table 7

Referee #2 (Remarks to the Author):

The authors have carefully considered all the comments and concerns raised during the first round of reviews. They performed extensive additional analyses and provided detailed explanations that result in a much-improved manuscript.

Here are my (minor) remaining points:

- This study identifies significant associations with severe Covid for a total of 5 relatively common variants with clear roles in type I interferon signalling. Even if the gene-based association tests performed here don't replicate the previous findings of Zhang et al. (Science 2020) regarding very rare deleterious variants (with MAF <0.1%) in genes of the same pathways, I think it would be adding a line in the discussion to comment on the convergence of the findings.

- On a related note, I couldn't find the supplementary File AVTsuppinfo.xlsx, which is supposed to show the SKAT-O p-values for the 13 interferon-related genes tested by Zhang et al. Also, a precise description of TLR7 variation is missing. Considering that TLR associations have now been replicated by several groups, it would be worth getting a more precise account of the variants that were observed in this cohort. Is there absolutely no enrichment of deleterious TLR7 variants?

- Still about TLR7: the reference [11] for the following statement (lines 133-134) is wrong: "We also did not replicate the reported association for the toll-like receptor 7 (TLR7) gene." The 3 best references for TLR7 associations are the following:

o van der Made et al., JAMA. 2020 Aug 18;324(7):663-673. doi: 10.1001/jama.2020.13719.

o Fallerini et al., Elife. 2021 Mar 2;10:e67569. doi: 10.7554/eLife.67569.

o Asano et al., Sci Immunol. 2021 Aug 19;6(62):eabl4348. doi: 10.1126/sciimmunol.abl4348.

- Additional remarks regarding references:

o Ref 9, Degenhardt et al., is incomplete (no journal / preprint indicated)

o Ref 56, Kachuri et al., has now been published in a peer-reviewed journal: Genome Med 12, 93 (2020). doi: 10.1186/s13073-020-00790-x

o Ref 72, Vosa et al., has now been published in a peer-reviewed journal: Nat Genet. 2021 Sep;53(9):1300-1310. doi: 10.1038/s41588-021-00913-z

Referee #3 (Remarks to the Author):

Thank you for the authors' fast and in-depth responses. I still have a few remaining comments:

1. **Number of independent risk loci**. It is typical in GWAS to define a risk locus as a genomic region within $\pm 500\text{kb} - 1\text{Mb}$ from the lead variant. I still think the authors should use this convention in presenting their number of independent risk loci. Even if after conditional analysis there are multiple risk variants, it could be just evidence for multiple causal variants in the SAME risk locus.

2. Figure 1 and Table 1 should present *HLA-DRB1* as the lead variant, and not just HLA (which is not a single gene).

3. **Colonization in the HLA region**. Interestingly there is no coloc signal for the HLA region (Figure 2). But a search on the OpenTargets website (https://genetics.opentargets.org/variant/6_32623820_T_C) shows the lead variant (rs9271609) is in an eQTL with HLA-DRB1 in multiple tissues. Please can the authors speculate for the reason why it is the case? Could it be that the default parameters of coloc are not ideal for this gene/variants-dense region? What is a simple Pearson correlation between GWAS versus eQTL p-value in this region?

4. The concordance rate of 87.6% is somewhat disappointing for the DRB1*04:01 allele, given both methods (HLA-LA and HIBAG) claimed a much higher accuracy (>90%) than this observed value. Please can the authors clarify whether they have incorporated the imputation uncertainty in their association study (i.e., using probabilistic dosage instead of genotype).

5. **Discussion on effective sample size**. I think their response towards how carefully curated case cohorts, especially in infectious diseases, are more powerful (fewer samples needed) might be a useful addition in the discussion. It would be informative for future genetic studies in infectious diseases.

Author Rebuttals to First Revision:

Whole genome sequencing identifies multiple loci for critical
illness caused by COVID-19
Second response to reviewers

2 Referee #1 (Remarks to the Author)

The authors have considered the previous comments carefully, and have addressed the comments.

My biggest concern is that many of these comments are not included in the main manuscript, but instead presented as tables or figures in the extensive supplementary section. This concerns me because these are real issues that you have tried to address and by not acknowledging them in the paper even in a few sentences, you will have others question this data as well, or misinterpret the findings. In addition, you are still holding back some of the key points that global investigators will find interesting—the fine mapping, credible sets, and conditional analyses.

We thank their reviewer for their comment. We have moved some suggested items from the supplementary material to main text and expanded relevant sections as detailed below.

1) The description of 100,000 genomes is now in the manuscript, but please include comments on this population and how this will or will not affect your results. Using rare cancer patients is not representative of the general population. Even if you summarize some of the disease breakdown. Cancer patients can also be considered immunocompromised if on treatment...but at the least they were cautious on exposures during the pandemic. The additional supplementary table 11 data can be included in a few lines of text.

We have added a new section (highlighted in red) to the results in the main text to present more clearly the caveats of using the 100k as controls and present more clearly the two sensitivity analyses additional matched study and the comparison with mild.

2) The interesting findings seem to be in the supplementary section! The credible set information and the fine mapping really should be incorporated into the main text. For example, the chromosome 3 finding—suggesting two hits under your peak is important!

You have the best data to get at what might be happening under these peaks (supplementary page 30) –and you have some diversity in your population to fine map it. Please make this more prominent in the main manuscript.

We thank the reviewer for their comment and we agree that our fine-mapping approach should be more prominent in the main text, as it represents a strength of this work owing to the availability of WGS data. To this end, we have moved the supplementary table giving fine-mapping details to the main text as Extended Data Table 2. In addition, we have now expanded the results section with the addition of a new section (highlighted in red) on fine-mapping in which we explicitly draw attention to the fine-mapping of the association signal in the 3p21 region.

Similarly, the additional conditional analyses should be addressed in the manuscript even if the figures remain in the supplementary.

We have added an explicit mention in the results, as per the reviewer’s suggestion, highlighted in red in the resubmitted manuscript.

It seems the authors are perhaps focused on rare variants not underlying their associations (a great point) but then not addressing how the rare variants (and LD blocks/recombination) actually help to narrow down the existing published work.

We thank the reviewer for their comment. We have now expanded the fine-mapping section of the results, following the reviewer’s suggestion, to explicitly mention the narrowing down of potential causal loci for many signals and refer to the new Extended Data Table 2 where 8 independent missense or worse variants are shown.

3) Age and ancestry may be confounded. In the age section please address the plots that show in some populations the age of the controls is very different than the age of the cases. The focus in the subsequent plots is on Europeans and age, but the other ancestry populations are not balanced and that may lead to some spurious results. See Supp Fig 10

We thank the reviewer for their comment and we have performed additional analyses to address it. We focused our matched study on the EUR population as the majority of our findings (22 out of 25 associations) were found in this population. However, to additionally check any biases for the three signals (chr1:155175305:G:A,chr2:60480453:A:G,chr6:41515007:A:C) that were discovered through multi-ancestry meta-analysis, we performed matching for the other three ancestries (SAS, AFR, EAS) as well (new breakdown Supplementary Figure 19), and meta-analysed the summaries. The results were almost identical to the unmatched primary study (Supplementary Figure 22) and are also pasted below for convenience.

Response to Review Figure 1: Comparison of effect sizes between unmatched and matched control results for the three loci that were found significant in the multi-ancestry meta-analysis. For the matched study, controls were matched to cases by propensity score matching to cases for which we had BMI information (using age, sex and bmi as matching covariates and performed separately for each ancestry).

4) In the individual conditional analysis, what is used in the conditional analysis? The effects seem to get stronger (more significant p-value) –can you include the region and or rsid that you are conditioning on?

We have added a column to Supplementary table 6 with the rsids of the variants that we conditioned on.

5) In the comparison to 2020 paper–two regions are less significant. Have those been replicated in other studies? Or are you indicating that they were initially false positives/spurious associations? Supplementary Table 7

Apart from the OAS1 locus, which could theoretically indicate a change in the virus, the only significant SNPs from the Pairo-Castineira et al.(2020) paper that have declined in significance are in the HLA region. These SNPs were not replicated in Pairo-Castineira et al. using HGI and 23andme data and we did not report them as confirmed findings. In subsequent HGI freezes these SNPs were shown to have high heterogeneity between studies indicating that they may be false positives or specific to some patient populations.

3 Referee #2 (Remarks to the Author)

The authors have carefully considered all the comments and concerns raised during the first round of reviews. They performed extensive additional analyses and provided detailed explanations that result in a much-improved manuscript.

Here are my (minor) remaining points:

- This study identifies significant associations with severe Covid for a total of 5 relatively common variants with clear roles in type I interferon signalling. Even if the gene-based association tests performed here don't replicate the previous findings of Zhang et al. (Science 2020) regarding very rare deleterious variants (with MAF <0.1%) in genes of the same pathways, I think it would be adding a line in the discussion to comment on the convergence of the findings.

We have added to the discussion of this interesting convergence. (Highlighted in red in submitted manuscript.) Note also that, since our last submission, we have successfully replicated the association in IFNA10 in an external cohort, and this is added to the main results table.

- On a related note, I couldn't find the supplementary File AVTsuppinfo.xlsx, which is supposed to show the SKAT-O p-values for the 13 interferon-related genes tested by Zhang et al. Also, a precise description of TLR7 variation is missing. Considering that TLR associations have now been replicated by several groups, it would be worth getting a more precise account of the variants that were observed in this cohort. Is there absolutely no enrichment of deleterious TLR7 variants?

We apologise to have omitted the supplementary info files in our previous revised submission. We have now added those in our latest revised manuscript.

Regarding TLR7, in our largest EUR cohort, we assessed 3 predicted Loss of Function (pLoF) and 36 missense variants. Although all 3 pLoF were found in the cases, this data did not generate a significant *P*-value ($P=0.30$). For missense mutations, we also observed a small enrichment in cases (39 missense in cases versus 28 missense in controls) but was not significant ($P=0.075$). We added these *P*-values for TLR7 tests to the burden results section explicitly.

- Still about TLR7: the reference [11] for the following statement (lines 133-134) is wrong: "We also did not replicate the reported association for the toll-like receptor 7 (TLR7) gene." The 3 best references for TLR7 associations are the following:
- van der Made et al., JAMA. 2020 Aug 18;324(7):663-673. doi: 10.1001/jama.2020.13719. - Fallerini et al., Elife. 2021 Mar 2;10:e67569. doi: 10.7554/eLife.67569. - Asano et al., Sci Immunol. 2021 Aug 19;6(62):eabl4348. doi: 10.1126/sciimmunol.abl4348.

We thank the reviewer for noting these omissions and we have amended the references for reported association for TLR7 as suggested.

Additional remarks regarding references:

- Ref 9, Degenhardt et al., is incomplete (no journal / preprint indicated) - Ref 56, Kachuri et al., has now been published in a peer-reviewed journal: *Genome Med* 12, 93 (2020). doi: 10.1186/s13073-020-00790-x - Ref 72, Vosa et al., has now been published in a peer-reviewed journal: *Nat Genet.* 2021 Sep;53(9):1300-1310. doi: 10.1038/s41588-021-00913-z

We have amended these.

4 Referee #3 (Remarks to the Author)

Thank you for the authors' fast and in-depth responses. I still have a few remaining comments:

1. Number of independent risk loci. It is typical in GWAS to define a risk locus as a genomic region within $\pm 500\text{kb} - 1\text{Mb}$ from the lead variant. I still think the authors should use this convention in presenting their number of independent risk loci. Even if after conditional analysis there are multiple risk variants, it could be just evidence for multiple causal variants in the SAME risk locus.

We thank the reviewer for their comment. We agree that "risk/genomic locus" conventionally refers to a region of 500Kbp-1Mbp that can include multiple independent signals. However, the precise span of a locus is arbitrary and depends on the resolution of the GWAS study that is affected by the sample size and marker density. In our study, we identified independent signals which were also part of different LD-clumps, despite their close proximity. In order to increase clarity, we have streamlined the text and replaced mentions of "genomic loci" with genomic signals/associations.

2. Figure 1 and Table 1 should present HLA-DRB1 as the lead variant, and not just HLA (which is not a single gene).

We agree with the reviewer and have changed the HLA reference to HLA-DRB1 for Figure 1 and Table 1.

3. Colonization in the HLA region. Interestingly there is no coloc signal for the HLA region (Figure 2). But a search on the OpenTargets website (https://genetics.opentargets.org/variant/6_32623820_T_C) shows the lead variant (rs9271609) is in an eQTL with HLA-DRB1 in multiple tissues. Please can the authors speculate for the reason why it is the case? Could it be that the default parameters of coloc are not ideal for this gene/variants-dense region? What is a simple Pearson correlation between GWAS versus eQTL p-value in this region?

We thank the reviewer for this valuable suggestion, and we added the information in the results (highlighted in red) and in the supplementary material (Supplementary File: TWAS.xlsx). 2 shows genes with significant TWAS association and also indication of colocalisation between the GWAS signal and the eqtl signal. In the case of HLA-DRB1 there isn't a significant TWAS signal; the main reason for the non-significant TWAS result is that for many tissues SNP heritability of the expression of the gene is too low to create a reliable model. We calculated colocalisation between HLA-DRB1 in lung and blood and found colocalisation in both tissues. Figure 2 shows genes with significant TWAS association and also indication of colocalisation between the GWAS signal and the eQTL signal. In the case of HLA-DRB1 there isn't a significant TWAS signal, showing that gene expression can not be predicted

using the SNPs present in the TWAS. In this case, the main reason for the non-significant TWAS result is that for many tissues SNP heritability of the expression of the gene is too low to create a reliable model. We calculated colocalisation between HLA-DRB1 in lung and blood and found colocalisation in both tissues. We thank the reviewer for the excellent suggestion, and we added the information in the results and in the supplementary material (Supplementary File: TWAS.xlsx), highlighted in red in the resubmitted manuscript.

4. The concordance rate of 87.6% is somewhat disappointing for the DRB1*04:01 allele, given both methods (HLA-LA and HIBAG) claimed a much higher accuracy (>90%) than this observed value. Please can the authors clarify whether they have incorporated the imputation uncertainty in their association study (i.e., using probabilistic dosage instead of genotype).

We thank the reviewer for their comments and agree that the concordance between methods of the DRB1*04:01 allele specifically is slightly lower than all the global concordance rate of DRB1. We took further measures to verify the concordance between HIBAG and HLA*LA for DRB1*04:01 calls. The reported concordance of 87.6% for DRB1*04:01 is without filtering out alleles that are not present in the HIBAG reference panel. When excluding calls made by HLA*LA for alleles that are not included in the HIBAG reference panel, over 95% of calls are identical between the two methods. Note that there are 55 distinct alleles (at 4 digit resolution) in the HIBAG reference panel for HLA-DRB1. As previously mentioned (see Supplementary HLA subsection: HLA Association Tests), we additionally performed our association analysis including only samples which were called identically between HIBAG and HLA*LA and found the odds ratio and P-value of DRB1*04:01 to be near-identical.

To clarify, we did not use genotype dosages, instead set a Call Threshold (CT - the posterior genotype probability for each HLA type) at 0.5 - as used by the authors of HIBAG who note that increasing the CT comes with improvement in accuracy but at a cost of lower genotype call rate. We set genotypes per HLA allele as 0, 1, 2 with calls less than the CT set to missing. To verify the robustness to the choice of call threshold for posterior probability, we performed the association tests without any call threshold (CT=0), with CT=0.5, and with CT=0.7. We found that DRB1*04:01 remained the only significant locus across each of the three CT values tested with the Odds Ratios and P-values being extremely consistent across all alleles (Supplementary Figure 31, also pasted below for convenience).

Response to Review Figure 2: Robustness of HLA association results to different posterior probability call thresholds for HIBAG. Manhattan plot of HLA allele associations across the extended MHC region with COVID-19 critical illness for the EUR cohort. Each panel corresponds to association results obtained using genotypes that were called using a different call threshold for HIBAG (0, 0.5 and 0.7, respectively). Diamonds represent the HLA each allele association, coloured by locus. The lead variant from the lead HLA allele is labelled. The dashed red line is the genome-wide significance threshold for Europeans.

5. Discussion on effective sample size. I think their response towards how carefully curated case cohorts, especially in infectious diseases, are more powerful (fewer samples needed) might be a useful addition in the discussion. It would be informative for future genetic studies in infectious diseases.

We thank the reviewer for this suggestion. In response, we have included estimates of heritability (using two methods, HDL and LDSC) from the different approaches to genetics of Covid-19 susceptibility (11), and we have added a section to the discussion in the resubmitted version of the manuscript, highlighted in red.

Reviewer Reports on the Second Revision:

Referees' comments:

Referee #1 (Remarks to the Author):

I appreciate that the authors have addressed all concerns thoughtfully and comprehensively. I believe the paper is more robust and informative.

Referee #2 (Remarks to the Author):

I would like to thank the authors for their answers and congratulate them for an important contribution to Covid host genomic knowledge. I don't have any additional comments.

Referee #3 (Remarks to the Author):

The authors have addressed my main concerns in their revised manuscript. I have only two minor comments:

- Update Figure 2 (TWAS and coloc results) with HLA-DRB1 highlighted;
- Update Figure 4 conditional analysis with rsIDs for the lead SNPs;